



Atmospheric
Chemistry
and Physics

# Soot PCF CE1: pore condensation and freezing framework for soot aggregates

**Claudia Marcolli[1], Fabian Mahrt[2,3], and Bernd Kärcher[4]**

[1]Institute for Atmospheric and Climate Science, Department of Environmental Systems Science, ETH Zurich,
8092 Zurich, Switzerland
[2]Department of Chemistry, University of British Columbia, 2036 Main Mall, Vancouver, BC, V6T 1Z1, Canada
[3]Laboratory of Environmental Chemistry, Paul Scherrer Institute, 5232 Villigen, Switzerland
[4]Institut für Physik der Atmosphäre, Deutsches Zentrum für Luft- und Raumfahrt (DLR Oberpfaffenhofen),
82234 Weßling, Germany

**Correspondence:** Claudia Marcolli (claudia.marcolli@env.ethz.ch)

**Abstract.** Atmospheric ice formation in cirrus clouds is often initiated by aerosol particles that act as ice-nucleating particles. The aerosol–cloud interactions of soot and associated feedbacks remain uncertain, in part because a coherent understanding of the ice nucleation mechanism and activity of soot has not yet emerged. Here, we provide a new framework that predicts ice formation on soot particles via pore condensation and freezing (PCF) that, unlike previous approaches, considers soot particle properties, capturing their vastly different pore properties compared to other aerosol species such as mineral dust. During PCF, water is taken up into pores of the soot aggregates by capillary condensation. At cirrus temperatures, the pore water can freeze homogeneously and subsequently grow into a macroscopic ice crystal. In the soot-PCF framework presented here, the relative humidity conditions required for these steps are derived for different pore types as a function of temperature. The pore types considered here encompass $n$-membered ring pores that form between $n$ individual spheres within the same layer of primary particles as well as pores in the form of inner cavities that form between two layers of primary particles. We treat soot primary particles as perfect spheres and use the contact angle between soot and water ($\theta_{sw}$), the primary particle diameter ($D_{pp}$), and the degree of primary particle overlap (overlap coefficient, $C_{ov}$) to characterize pore properties. We find that three-membered and four-membered ring pores are of the right size for PCF, assuming primary particle sizes typical of atmospheric soot particles. For these pore types, we de-

rive equations that describe the conditions for all three steps of soot PCF, namely capillary condensation, ice nucleation, and ice growth. Since at typical cirrus conditions homogeneous ice nucleation can be considered immediate as soon as the water volume within the pore is large enough to host a critical ice embryo, soot PCF becomes limited by either capillary condensation or ice crystal growth. We use the soot-PCF framework to derive a new equation to parameterize ice formation on soot particles via PCF, based on soot properties that are routinely measured, including the primary particle size, overlap, and the fractal dimension. These properties, along with the number of primary particles making up an aggregate and the contact angle between water and soot, constrain the parameterization. Applying the new parameterization to previously reported laboratory data of ice formation on soot particles provides direct evidence that ice nucleation on soot aggregates takes place via PCF. We conclude that this new framework clarifies the ice formation mechanism on soot particles in cirrus conditions and provides a new perspective to represent ice formation on soot in climate models.

## 1 Introduction

Incomplete combustion of biomass or fossil fuel produces copious quantities of soot particles, encompassing mixtures of black carbon and organic carbon that are of particular importance in the Earth's atmosphere. Soot particles influence

atmospheric chemistry (Alcala-Jornod et al., 2000; Ammann et al., 1998; Andreae and Crutzen, 1997) and significantly contribute to air pollution, thereby negatively affecting human health (Bové et al., 2019; Janssen et al., 2012; Laumbach and Kipen, 2012). In addition, soot particles play an important role in climate directly through absorption and scattering of incoming shortwave radiation and indirectly by acting as ice-nucleating particles (INPs) (Vali et al., 2015) TS1 for clouds (Bond et al., 2013; Bond and Bergstrom, 2006; Ramanathan and Carmichael, 2008; Reddy and Boucher, 2007; Wang et al., 2014). Previous studies have found soot to be the second-most-important radiative forcing agent after carbon dioxide (Jacobson, 2001). However, owing to the complexity of the physicochemical properties of atmospheric soot particles, major gaps still exist in our ability to quantify their impact on these various processes. A key example is the large uncertainties associated with aerosol–cloud interactions of soot that affect future climate projections (Bond et al., 2013). In particular, the ice nucleation activity of soot particles and their impact on cirrus clouds remain largely unconstrained.

The ice nucleation activity of soot particles has been shown to impact cirrus cloud coverage, emissivity, and lifetime, thereby affecting the radiative balance (e.g. Lohmann, 2002; Lohmann et al., 2020; McGraw et al., 2020). Knowledge of the ice nucleation ability of soot particles is essential for estimating the anthropogenic influence on cirrus clouds (Kärcher, 2017; Penner et al., 2018; Zhao et al., 2019). For instance, the incomplete understanding of how aircraft-emitted soot particles nucleate ice crystals in cirrus clouds is one of the most uncertain components in assessing the climate impact of aviation (Lee et al., 2021). Such effects, if substantiated, are likely to become increasingly more important in view of the projected increase in global air traffic (ICAO Report, 2018).

There has been a considerable body of work aiming at understanding the ice nucleation activity of soot from different emission sources and for different atmospheric conditions. These studies have found soot generally to be a poor ice nucleus above the homogeneous nucleation temperature of supercooled liquid water $T > \mathrm{HNT}$ (homogeneous ice nucleation temperature of water; $\sim 235\,\mathrm{K}$) and below water saturation (Chou et al., 2013; DeMott, 1990; Dymarska et al., 2006; Friedman et al., 2011; Kanji and Abbatt, 2006; Mahrt et al., 2018a TS2; Möhler et al., 2005a). Above (bulk) water saturation, water vapour can condense on the soot particles, and they can act as INPs in the condensation and/or immersion freezing mode. Since previous studies have reported no or only weak ice nucleation activity for soot at $T > \mathrm{HNT}$ and relative humidities with respect to water ($\mathrm{RH_w}$) at or above 100 % (Chou et al., 2013; DeMott, 1990; Diehl and Mitra, 1998; Gorbunov et al., 1998, 2001; Kireeva et al., 2009; Schill et al., 2016), the importance of soot to nucleating ice in mixed-phase clouds (MPCs) is most likely negligible (Kanji et al., 2020; Schill et al., 2016, 2020a; Vergara-Temprado et

al., 2018), given the abundance of other, more efficient, INPs in these conditions. At $T \leq \mathrm{HNT}$, on the contrary, studies have reported uncoated soot particles nucleate ice well below relative humidities required for homogeneous freezing of solution droplets (Crawford et al., 2011; Ikhenazene et al., 2020; Koehler et al., 2009; Kulkarni et al., 2016; Mahrt et al., 2018a; Möhler et al., 2005a, b; Nichman et al., 2019; Zhang et al., 2020). A summary of studies that have investigated the ice nucleation abilities of different uncoated soot types and aggregate sizes at cirrus temperatures is given in Table F1. In Fig. 1 we summarize the ice nucleation onset conditions reported in these studies, in terms of the relative humidity with respect to ice ($\mathrm{RH_i}$) required to reach an (ice) activated fraction (AF) of 1 % at a given temperature. For instance, Nichman et al. (2019) investigated the ice nucleation ability of six different soot types, using aggregates of 800 nm electrical mobility diameter. They found the ice nucleation activity to depend on soot particle properties such as morphology and surface oxidation. The same study also reported a dependence on aggregate size for some of the investigated soot types, when comparing the ice nucleation activity of aggregates with diameters of 800 and 100 nm. A size dependency of ice nucleation was also found in the studies of Zhang et al. (2020) and Mahrt et al. (2018).

While these studies reveal significant scatter in terms of the ice nucleation activity, as evident from e.g. the range of onset $\mathrm{RH_i}$ values observed for different soot types at a given temperature, they indicate that soot particles could represent an eminent source of INPs in upper tropospheric conditions, with potentially important implications for cirrus cloud formation and climate. At the same time, field studies have reported soot to be largely absent in cirrus cloud residuals (Cziczo et al., 2013; Cziczo and Froyd, 2014). While the latter could be partially influenced by the lower size limit of $\sim 0.2\,\mu\mathrm{m}$ of the mass spectrometer deployed (Cziczo et al., 2006), these contrary conclusions reveal that the ice nucleation ability of soot is insufficiently understood to date.

The recent studies by Nichman et al. (2019) and Mahrt et al. (2018) have demonstrated that for $T < \mathrm{HNT}$, ice nucleation via pore condensation and freezing (PCF; Christenson, 2013; David et al., 2019, 2020; Fukuta, 1966; Higuchi and Fukuta, 1966; Marcolli, 2014) is the prevailing mechanism of ice formation on bare, uncoated soot particles, but it can be hampered if the pores are covered by hydrophobic material (Mahrt et al., 2020a; Zhang et al., 2020). During PCF, water uptake into pores and cavities due to capillary condensation is followed by ice nucleation of the pore water and growth of the ice out of the pores resulting in macroscopic ice crystals. While PCF is recognized to have the potential to explain the disparate ice nucleation ability of soot in mixed-phase compared to in cirrus cloud conditions, previous studies have mostly not taken the specific characteristics of soot particles into account to obtain a more coherent understanding of their ice nucleation pathway. A common property of all soot particles is their fractal morphology, with soot aggre-

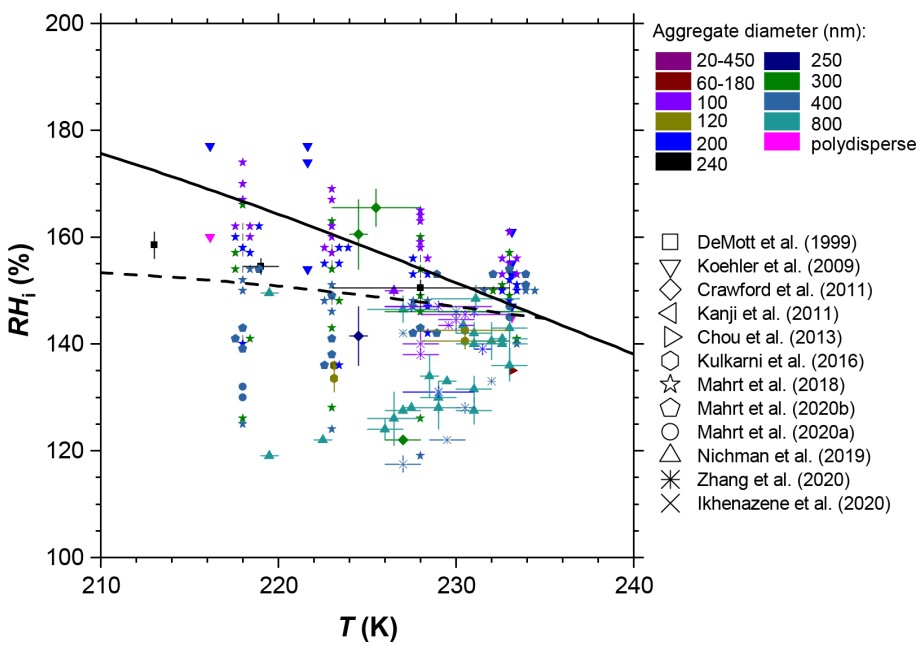

**Figure 1.** Summary of ice nucleation onset conditions for soot aggregates of different type and size. The ice onset is defined as the relative humidity with respect to ice ($RH_i$) and temperature ($T$), for which the (ice) activated fraction (AF) reaches 1 % of the total soot particle number concentration. Data are compiled from different studies, as indicated in the legend, and constrained to ice nucleation experiments on uncoated soot particles and experiments where values of $AF = 1$ % were reached and respective $RH_i$ and $T$ conditions were reported and/or directly accessible from the data presented (see Table F1 for a full list). The solid black line indicates water saturation according to the parameterization of Murphy and Koop (2005), and the black dashed line indicates conditions for homogeneous ice nucleation of solution droplets according to Koop et al. (2000). Overlapping data points are shown horizontally separated from each other with the onset conditions marked by small grey lines. Vertical error bars indicate the range of $RH_i$ values over which ice formation onset was observed in the respective studies when multiple experiments with the same soot type and size were performed at the same temperature. Conversely, horizontal error bars denote the $T$ range when multiple ice nucleation experiments covered different temperatures and the reported onset $RH_i$ was within the experimental uncertainty.

gates being composed of graphitic primary carbon spherules that cluster together (Park et al., 2004; Samson et al., 1987). Such sintered primary particles provide ample pore structures (Rockne et al., 2000), where water can condense. The equilibrium water vapour pressure over such capillary condensates, forming a concave water surface, is reduced compared to bulk water, as described by the Kelvin equation (e.g. Fisher et al., 1981; Marcolli, 2014, 2020). Therefore, water can already be taken up at $RH_w < 100$ %. Evidence of PCF being the dominant ice formation mechanism on soot is a sharp, almost step-like increase in the ice fraction at $T < $ HNT and below water saturation, as reported by Mahrt et al. (2018). Such a distinct increase cannot be explained by classical nucleation theory (CNT) and assuming ice formation to proceed via deposition nucleation, where a liquid phase is absent (David et al., 2019; Welti et al., 2014). Besides soot, PCF was recently also observed to cause ice formation on other combustion particles such as coal fly ash (Umo et al., 2019).

In general, PCF is controlled by the contact angle ($\theta$) defining the interface of soot and water (or ice) as well as the pore diameter and geometry (David et al., 2020; Fukuta,

1966; Marcolli, 2017b, 2020). The contact angle denotes the wettability of a (soot) particle surface (Lohmann et al., 2016; Pruppacher and Klett, 1997). Recent work has shown that the contact angle can play a relevant role in controlling the capillary uptake of water and thus ice formation in cylindrical pores with diameters of between 2.5 and 9.1 nm (David et al., 2020). However, given the heterogeneity in the physical (pore size and geometry) and chemical (contact angle) properties of soot particles from different combustion sources, undergoing different ageing processes, their ability to nucleate ice via PCF may vary strongly. Previous studies have focused on using the PCF framework to predict capillary condensation and subsequent homogeneous freezing of pore water in simple pore geometries, including cylindrical pores (Marcolli, 2014) and more recently conical and wedge-shaped pores (Marcolli, 2020) TS3. The focus was on geometries that are believed to be representative of cavities and pores found on mineral dust particles (e.g. Kiselev et al., 2016). As the conditions of capillary condensation depend on pore geometry and the contact angle, neglecting the pore shape and wettability of (atmospheric) soot aggregates to predict PCF is too simplistic. Here, we develop a new theoretical frame-

work to explore and constrain PCF on soot aggregates, capturing their vastly different pore properties compared to other aerosol species such as mineral dust. Specifically, we extend the previous work by Marcolli (2020) to pores formed between sintered primary soot particles and systematically investigate the effect of the contact angle, following earlier work on capillary condensation on soot aggregates by Persiantseva et al. (2004) and very recently in Jantsch and Koop (2021). Based on the physical properties of soot, we derive a new framework to predict the ice nucleation ability of soot particles resulting from PCF that fits experimental results from previous laboratory measurements (Mahrt et al., 2018a, 2020b). The PCF-based parameterizations derived from this framework are able to predict the ice activated fraction (AF) of soot aggregates, ultimately improving parameterizations (e.g. Ullrich et al., 2017) currently used in climate models to simulate their effects on cirrus clouds and associated feedbacks.

The paper is structured as follows: in Sect. 2 we summarize the soot properties that are relevant for PCF, and in Sect. 3 we define the pore structures that serve as the basis for the soot-PCF framework developed herein. In Sects. 4–6, we discuss the different steps of soot PCF, namely ice nucleation (Sect. 4), which is preceded by pore water condensation (Sect. 5) and followed by ice growth out of the pores on the soot aggregates (Sect. 6). Section 7 discusses the relevance of pre-activation for soot PCF, while Sect. 8 presents a novel parameterization of ice nucleation by soot based on the soot-PCF framework. Atmospheric implications are discussed in Sect. 9, followed by a summary and conclusions in Sect. 10.

A detailed derivation of soot-PCF equations can be found in the appendices. Pore geometries are derived in Appendix A and pore-filling and ice growth conditions in Appendix B. The derivation of the contact angle between ice and soot is given in Appendix C, while Appendix D features a compilation of different soot-PCF curves for relevant combinations of contact angle, primary particle diameter, and overlap. A detailed derivation of the soot-PCF parameterization is given in Appendix E, followed by a compilation of ice nucleation data from previous studies (Appendix F) and a list of symbols (Appendix G).

## 2 Soot properties

Soot forms aggregates that are composed of primary carbon spheres, characterized by an onion-shell structure of folded graphite sheets, which typically have diameters $D_{pp}$ of between 10 and 45 nm but that can also reach larger values (e.g. Kim et al., 2001; Liati et al., 2014; Megaridis and Dobbins, 1990; Wentzel et al., 2003). The primary particles coagulate to form soot clusters that grow further in size into soot aggregates as a result of monomer–cluster or cluster–cluster collision. Sizes of ambient soot aggregates range from a few primary particles to larger, fractal aggregates with tens

to hundreds of primary particles, depending on the combustion source and conditions (Sorensen, 2011; Wentzel et al., 2003). Diameters typically range from 0.01 to 1 µm (Ogren and Charlson, 1983; Rose et al., 2006), but they can also reach larger values depending on the source and atmospheric trajectory (e.g. Okada et al., 1992). In the past, considerable efforts have been made to characterize soot from different combustion sources and conditions and to quantify morphological parameters such as the mobility diameter, $D_m$; the primary particle diameter, $D_{pp}$; and the primary particle overlapping coefficient, $C_{ov}$. The early studies by Hess and McDonald (1983), Samson et al. (1987), and Megaridis and Dobbins (1990) set the stage to investigate the morphology of soot aggregates using transmission electron microscopy (TEM) and were followed by studies of e.g. Köylü et al. (1995, 1992) and Sorensen et al. (1992). More recently, automated algorithms have emerged that can measure morphological parameters such as the primary particle diameter from TEM images of soot (Anderson et al., 2017; Bescond et al., 2014; Dastanpour et al., 2016; Grishin et al., 2012). Given the large availability of TEM-based information on primary particle size and overlapping coefficients, we make these parameters a central part of our soot-PCF framework presented here. Relevant properties for PCF are pore shape, pore size, and the contact angle between the pore surface with water and ice. In the following, we discuss these particle properties with respect to atmospheric soot. In our soot-PCF framework we assume that pores form between spherical primary particles and that their size and shape depend on the arrangement of the primary particles, their respective size, and the overlap between primary particles. In turn, the number of pores in a soot aggregate depends on the number of primary particles in the aggregate and their compaction, i.e. fractal dimension.

### 2.1 Primary particle size

Ambient soot aggregates typically consist of primary particles with diameters of 1 nm to a few tens of nanometres that are close to a spherical shape. The size of the primary particles depends on the emission source, as well as on the fuel type and the combustion conditions. Relevant combustion conditions that influence the primary particle diameter include the fuel-to-air (oxygen) ratio during combustion, the flame temperature, and the residence time within the flame. Previous studies have investigated the primary particle diameter of soot particles from various emission sources, including car engines (e.g. Burtscher, 2005; Ferraro et al., 2016; Smekens et al., 2005; Su et al., 2004), aircraft turbines (e.g. Delhaye et al., 2017; Liati et al., 2014, 2019; Marhaba et al., 2019; Smekens et al., 2005), and commercial carbon blacks and/or soot generated from controlled flames in the laboratory (e.g. Clague et al., 1999; Cortés et al., 2018; Ferraro et al., 2016; Megaridis and Dobbins, 1990; Ouf et al., 2016, 2019). For instance, Liati et al. (2014) used TEM images to investigate the size of primary particles of a turbofan engine

(type CFM56-7B26/3), frequently used in Boeing 737 aircraft, that was operated with standard Jet A-1 fuel. They reported the primary particle diameter to increase with increasing thrust level. At 100 % thrust, typical of take-off conditions, the majority of primary particles ($\sim$ 60 %) had diameters of between 10 and 25 nm, with a mode of 24 nm and 52 % of the particles being larger than 20 nm. At cruise conditions, equivalent to a thrust level of 65 %, the average size of the primary particles was smaller, with a mode diameter of 20 nm and the majority of particles (75 %) in the range between 10 and 25 nm. At 7 % thrust, corresponding to taxiing, the mode decreased to 13 nm with most particles (99 %) being in the range between 5 and 20 nm. The trend of increasing primary particle diameters with increasing thrust levels is consistent with other studies of aircraft soot emissions (Delhaye et al., 2017; Liati et al., 2019; Marhaba et al., 2019). In their detailed TEM characterization of aircraft soot from the same engine type, Liati et al. (2019) also investigated the effect of fuel type on the soot morphology. This was achieved by comparing the primary particle size when burning standard Jet A-1 fuel and a blend of the same fuel, along with 32 % hydro-processed esters and fatty acids (HEFA) biofuel, and analysing soot aggregates collected at thrust levels of 85 % (climbing conditions) and 4 % (idle conditions). Their results reveal that the biofuel blend leads to smaller primary particles at both thrust levels compared with standard kerosene. This finding is consistent with Moore et al. (2017), who reported a reduction by 3–5 nm towards smaller primary particle diameters, for the number size distributions sampled at cruise conditions behind a turbofan engine (CFM56-2C1) fed with a 1 : 1 mixture of standard Jet A-1 fuel and HEFA biofuel, compared to when operating the engine on pure Jet A-1 fuel. Nonetheless, the results of Liati et al. (2019) revealed that the majority of particles (75 %–85 %TS4) had diameters of between 5 and 10 nm in idle conditions, and 60 % of the primary particles were found to have diameters of between 15 and 20 nm at 85 % thrust, independently of the fuel type. These values are in good agreement with those reported by Delhaye et al. (2017), who found a mean primary particle diameter of 15 nm, with extreme values of 4 and 70 nm, when testing a turbofan engine (Snecma–NPO Saturn SaM146 1S17) over different thrust levels ranging from 7 % to 100 %.

There are also many measurements published of primary particle sizes from vehicle emissions. For instance, Su et al. (2004) used TEM images to investigate the primary particle sizes of a Euro IV diesel engine. They reported a mean primary particle diameter of 13 nm, which was found to be significantly lower compared to the mean primary particle diameter of 35 nm for soot particles from older engines reported in the same study. The latter is consistent with the value of 30 nm for the mean primary particle diameter determined for diesel (VW Golf) soot particles, in idle and full speed conditions (Smekens et al., 2005), which compares well with previously reported values for diesel soot (Amann

and Siegla, 1981; Bérubé et al., 1999; Roessler, 1982). Similarly to the observations for aircraft soot, Smekens et al. (2005) reported smaller primary particles when the car was operated with biodiesel, namely 29 nm (idle) and 26 nm (full speed). More recently, Ferraro et al. (2016) sampled soot particles from a diesel car engine (Peugeot DV4) drained from the engine oil and reported the majority of particles (91.3 %) had a mean primary particle diameter of 25.4 nm while some particles (8.7 %) had primary particle diameters of 48 nm.

In the past, many laboratory sources have been developed with the goal to provide relevant analogues for atmospheric soot particles from e.g. car or aircraft engines. In these burners the combustion conditions can be varied in a controllable manner to generate soot with primary particle diameters of between approximately 5 and 50 nm (e.g. Cortés et al., 2018; Megaridis and Dobbins, 1990, and references therein), i.e. covering the size range relevant for ambient soot. Overall, soot particles emitted from different sources and using different fuel types and engine operating conditions have comparable but not identical primary particle sizes. In the following, we encompass the relevant range of primary particle sizes and focus our discussion on $10\,\text{nm} \leq D_{\text{pp}} \leq 30\,\text{nm}$.

## 2.2 Overlap between primary particles

The primary particles constituting a soot aggregate are typically not in point contact but overlap partially. Overlap between primary particles is sometimes referred to as sintering or necking and can result from various processes. For instance, adjacent monomers can merge (overlap) in high-temperature environments of the flame resulting from a lack of particle rigidity. As another example, primary particles can overlap due to strong attraction forces between them (Brasil et al., 2000). Yet, primary particle overlap can also result from an overall physical compaction of soot aggregates, e.g. during cloud processing (e.g. Bhandari et al., 2019; China et al., 2015; Mahrt et al., 2020b). The degree by which neighbouring primary particles overlap can be quantified by the overlapping parameter, $C_{\text{ov}}$, introduced by Brasil et al. (1999):

$$C_{\text{ov}} = \frac{D_{\text{pp}} - D_{\text{ij}}}{D_{\text{pp}}}, \tag{1}$$

where $D_{\text{ij}}$ denotes the distance between the centres of two overlapping/intersecting spheres with diameter $D_{\text{pp}}$. When two adjacent primary particles are in point contact, $C_{\text{ov}}$ is 0, whereas $C_{\text{ov}} = 1$ describes the other extreme for two completely overlapping primary particles. Previous studies have reported values for $C_{\text{ov}}$ to typically range within 0.05 and 0.29 (Brasil et al., 1999; Ouf et al., 2010, 2019; Wentzel et al., 2003). Wentzel et al. (2003) found values of between 0.1 and 0.29 for soot aggregates from diesel emissions. As another example, Cortés et al. (2018) reported values of $0.04 \leq C_{\text{ov}} \leq 0.18$ when using TEM to study soot aggregates

generated in co-flow diffusion flames using different fuels (ethylene, propane, and butane), which is in good agreement with the values (0.11–0.17) of soot aggregates derived from diffusion flames reported in Ouf et al. (2019). The overlap of primary particles in soot aggregates has gained attention because it impacts the radiative properties of the aggregates (e.g. Yon et al., 2015). Below, we demonstrate that $C_{ov}$ also impacts the pore geometry and hence the ability of ice to form via PCF on soot aggregates.

## 2.3 Soot aggregate size and compaction

A large body of work has been devoted to elucidating the size of soot aggregates from various emission sources, including car engines (e.g. Clague et al., 1999; Harris and Maricq, 2001, 2002; Karjalainen et al., 2014) and aircraft turbines (e.g. Delhaye et al., 2017; Lobo et al., 2015; Masiol and Harrison, 2014; Moore et al., 2015), as well as commercial and laboratory-generated soot (e.g. Han et al., 2012). For instance, Kinsey et al. (2010) observed the mean soot aggregate diameter to increase from approximately 10 to 35 nm, when increasing the thrust from 7 % to 100 %, when sampling engine exhaust plumes during the Aircraft Particle Emissions Experiment (APEX). This trend is consistent with earlier findings of Hagen and Whitefield (1996). Note that this range overlaps with the size range of primary particles rendering it likely that reported sizes refer in the lower limit to single primary particles rather than to aggregates. Similarly, Delhaye et al. (2017) reported the modal diameter of the soot number size distribution of particles emitted from a turbofan aircraft engine (SaM146 1S17) to range from 17 to 55 nm, when testing the emissions according to the ICAO landing and take-off (LTO) cycle, corresponding to thrust levels of 7 %, 30 %, 85 %, and 100 %. This range is consistent with other measurements (e.g. Liati et al., 2019; Lobo et al., 2015; Mazaheri et al., 2009; Moore et al., 2017). As another example, Mazaheri et al. (2009) investigated aircraft emissions during take-off and landing for four different aircraft engine types and found the majority of the soot aggregates to have diameters smaller than 30 nm, with a secondary mode being centred between 40 and 100 nm. This secondary mode is in line with the measurements by Herndon et al. (2008), who observed a mode of between 60 and 80 nm during take-off. Similarly to the primary particle diameter, the soot aggregate size is sensitive to the operation conditions, with larger aggregates being observed in climb conditions compared to in idle conditions. For instance, Liati et al. (2019) found 80 % of the aggregates to be smaller than 40 nm in diameter in idle conditions, while at an engine thrust of 85 %, a modal diameter of between 40 and 80 nm was observed, accounting for approximately 35 % of the total particles, while 20 % of the aggregates had diameters of between 80 and 120 nm and the remaining fraction was larger than 120 nm. Overall, diameters of soot particles (comprising individual primary particles and aggregates) emitted from aircraft engines are mostly in the range between 5 and 100 nm, even though some studies have also reported larger soot aggregates in e.g. contrail residuals (Petzold et al., 1998; Twohy and Gandrud, 1998). Comparable values have been reported for soot emitted from gasoline and diesel vehicles. Number size distributions of particulate matter from vehicle exhaust usually reveal a clear bimodality, with the maximum particle concentration observed in the nucleation mode and the secondary peak located in the accumulation mode.

For emissions from diesel engines the mean soot aggregate diameter for the accumulation mode particles is almost always within the range of 60 to 120 nm and is well represented by lognormal distributions with geometric standard deviations of 1.8–1.9 (Burtscher, 2005; Harris and Maricq, 2001). Furthermore, diesel soot aggregate sizes have been reported to be largely independent of the size and type of the engine, as well as of the operation conditions (Burtscher, 2005). Soot aggregates from gasoline vehicles are generally smaller, and number size distributions tend to be more asymmetric (Harris and Maricq, 2001). Karjalainen et al. (2014) observed a bimodal number size distribution for particles emitted from gasoline vehicles, with the modes corresponding to mean diameters below 30 and 70 nm, respectively, including small almost-spherical soot particles and larger agglomerated soot particles. The latter value falls well within the range of 40 to 80 nm reported previously (Harris and Maricq, 2001). Some studies have also reported diesel soot aggregates significantly larger than 200 nm in diameter, even reaching supermicron sizes (Hoard et al., 2013). Such particles likely result from flaking particle deposits that build up within the combustion chamber or the exhaust pipe. However, given that such larger particles should be more easily captured by exhaust filters, their atmospheric relevance is questionable.

In the discussion above, the reported sizes are mostly given as electrical mobility diameters, $D_m$, of soot aggregates but also comprise in the limit to small sizes $D_m$ values of individual primary particles that have not undergone any aggregation. Only a few studies have investigated the relation between the aggregate size and the primary particle size. For instance, Dastanpour and Rogak (2014) reported a positive correlation between primary particle and aggregate size, with larger aggregates containing larger primary particles, based on a TEM analysis of soot from various combustion sources, including particles sampled from aircraft jet engines. The same study revealed that within an individual aggregate, the primary particle size is more uniform compared to an ensemble of aggregates. A similar trend between primary particle size and aggregate size was suggested by Olfert et al. (2017), which can have important implications for the pore number, size, and type that can form on individual soot aggregates. Based on these results, Olfert and Rogak (2019) have demonstrated that the primary particle size increases with aggregate size, following a universal scaling law ($D_{pp} \propto D_m^{0.35}$ TS5), that is valid for a wide range of soot particles/sources.

The number of equally sized primary particles contained in a soot aggregate of a given mobility diameter depends on the degree of compaction. Freshly emitted particles are typically fractal-like chain agglomerates that can become more compact via various ageing processes during transport through the atmosphere (Bhandari et al., 2019; China et al., 2015) such as coagulation, condensation of semi-volatile material, heterogeneous reactions, relative humidity changes, and cloud processing (China et al., 2015; Ding et al., 2019; Yuan et al., 2019). To quantify the degree of compaction, TEM or SEM (scanning electron microscopy) images of soot particles can be evaluated with respect to morphological descriptors such as aspect ratio, roundness, convexity, or fractal dimension (e.g. Bhandari et al., 2019; China et al., 2015; Mahrt et al., 2020b). Such analyses evidence compaction at high RH and collapse after cloud cycling (Colbeck et al., 1990; Huang et al., 2014; Zuberi et al., 2005). Collapse of fractal-like aggregates due to capillary condensation of water has also been reported in HTDMA (humidified tandem differential mobility analyser) measurements, where a decreasing mobility diameter with increasing RH has been observed in the case of carbon particles (Weingartner et al., 1997, 1995). Conversely, fresh diesel and jet engine combustion particles exhibit much less restructuring together with hygroscopic growth that becomes more pronounced with increasing sulfur content (Gysel et al., 2003; Weingartner et al., 1997). Compaction has also been observed in field measurements (Bhandari et al., 2019; Ding et al., 2019). For instance, soot particles sampled from evaporated cloud droplets have been found to be significantly more compact than freshly emitted and interstitial soot (Bhandari et al., 2019).

While circularity or the aspect ratio of soot aggregates are two-dimensional morphological parameters that provide descriptive measures of compaction, classification via the (three-dimensional) fractal dimension has deeper implications as soot is assumed to be a fractal object. Yet, soot aggregates are not truly fractals because they are not completely scale invariant but exhibit self-similarity only over a finite range of length scales (Huang et al., 1994; Mandelbrot, 1977). Nonetheless, the concepts of fractal geometry have successfully been used to quantitatively describe their morphology during aggregate growth by agglomeration (Sorensen, 2011). To describe soot aggregates as fractals, the primary particles are assumed to all be of the same size with point contacts between each other (Sorensen, 2011). As a consequence of self-similarity, the number of primary particles scales as a power law with the radius, implying a fractal dimension of 1 for a chain agglomerate and a fractal dimension of 3 for primary particles ordered as a sphere. Assuming diffusion-limited cluster–cluster aggregation, a characteristic fractal dimension for soot aggregates of 1.78 should result, while in the case of reaction-limited cluster–cluster aggregation the expected fractal dimension is 2.1 (Sorensen, 2011). The (two-dimensional) fractal dimension from TEM images of soot aggregates has been derived by relating the maximum length of soot particles with the number of particles contained in them. While fractal dimensions of loose chain agglomerates can be well determined with this method, the fractal dimension tends to be underestimated in the case of compacted soot aggregates. Therefore, China et al. (2015) resorted to a two-dimensional fractal dimension such that a sphere would be assigned a fractal dimension of 2. Alternatively, the fractal dimension of soot aggregates can be derived by relating the particle mass measured by an aerosol particle mass analyser (APM) to the electrical mobility diameter from a DMA (differential mobility analyser) through a power law of the form $m \propto D_{\mathrm{m}}{}^{D_{\mathrm{f}}}$, where $m$ is the mass of a (size-selected) soot aggregate (e.g. Schmidt-Ott et al., 1990). Using both methods, Park et al. (2004) obtained fractal dimensions of 1.75 from the analysis of TEM images while relating the soot aggregate mass with its mobility diameter yielded a fractal dimension of 2.35. While the (two-dimensional) TEM analysis probably underestimated the fractal dimension, Sorensen (2011) argued that the interpretation of the exponent as the fractal dimension in the relation of aggregate electrical mobility diameter with aggregate mass is misleading in the limit of aggregates containing only a few primary particles and results in too-high fractal dimensions. Shortcomings in the analysis of two-dimensional projections have been confirmed by electron tomography yielding higher fractal dimensions (Adachi et al., 2007; Baldelli et al., 2019), yet this method is laborious and not suited to screening a large number of aggregates for a statistical analysis.

## 2.4 Contact angle

The contact angle of soot particles with a water droplet is a measure to quantify the hydrophilicity/wettability of the particle surface. The maximum contact angle of $\theta_{\mathrm{sw}} = 180°$ (non-wetting case) represents the situation of a free spherical water droplet in point contact with the soot substrate. In contrast, if the soot surface were completely covered by a water layer (complete wetting case), the contact angle between the soot surface and the water droplet would become $\theta_{\mathrm{sw}} = 0°$ (Lohmann et al., 2016). For capillary condensation taking place below bulk water saturation, soot–water contact angles below 90° are required, because only if $\theta_{\mathrm{sw}} < 90°$ TS6 does the capillary water exhibit a concave curvature, implying an equilibrium vapour pressure that is below that of bulk water. Early work by Fowkes and Harkins (1940) reported values of 85.3–85.9° for $\theta_{\mathrm{sw}}$, which are in good agreement with the value of $\theta_{\mathrm{sw}} = 83.9°$, measured by Morcos (1972). However, soot derived from different combustion sources and having undergone different kinds and degrees of atmospheric processing can span a wide range of contact angles. For instance, highly ordered pyrolytic graphite has been reported to have soot–water contact angles in the range of 75–91° (Shin et al., 2010; Westreich et al., 2007). Persiantseva et al. (2004) investigated the soot–water contact angle for different surrogates of aviation soot, using the sessile drop technique.

They tested aircraft engine combustor soot (AEC) by operating a gas turbine in an engine test stand (Popovitcheva et al., 2000). When simulating cruise conditions and using TS1 kerosene as fuel, they found $\theta_{sw} = 63°$ for freshly emit-
ted soot. Persiantseva et al. (2004) also investigated soot derived from burning different types of aviation fuels (German kerosene, TC1 and T6) in an oil lamp and reported the soot–water contact angles to range between 70 and 80°, demonstrating that the fuel composition can strongly impact the
contact angle of the resulting combustion particles. Similar values of 59 and 69° for soot derived from burning TC1 and oxidized (aged) TC1 kerosene, respectively, were reported by Popovicheva et al. (2008a), who also investigated contact angles of other soot types. A lower contact angle than that from
aviation fuel of $\theta_{sw} = 59°$ was found for lamp black soot, derived from burning lantern fuel in an oil lamp (Persiantseva et al., 2004). Interestingly, Persiantseva et al. (2004) found $\theta_{sw}$ to decrease by 17° for TC1 soot, after heating the sample at 300 °C for 30 min. An even more pronounced decrease
of 35° from the original value was found when outgassing the kerosene soot for 18 h at 200 °C, leading to a value of $\theta_{sw} = 43°$, which was attributed to a loss of semi-volatile organics and a change in surface composition. This is in good agreement with earlier work demonstrating contact angles on
graphite to be as low as $35 \pm 4°$, after evacuating the sample to $10^{-10}$ Torr ($\sim 1.33 \times 10^{-8}$ Pa), compared to values of between 50 and 80° found for the untreated (0001) graphite surface (Schrader, 1975). These results are supported by more recent findings demonstrating that contamination, e.g. in the
form of hydrocarbon adsorption on the soot particle surface, can increase the soot–water contact angle (e.g. Kozbial et al., 2016; Martinez-Martin et al., 2013). Proxies for soot particles from aviation emission were also studied by Kireeva et al. (2009). When burning TS1 kerosene in an oil lamp (dif-
fusion flame), a contact angle of $\theta_{sw} = 59°$ was found. In the same study, Kireeva et al. (2009) also investigated different commercial carbon blacks (lamb black, furnace black, channel black, thermal black, acetylene soot). Using the sessile drop technique, they found contact angles to range from 32 to
47°. Han et al. (2012) investigated the influence of the combustion condition (fuel-to-oxygen ratio) on the hydrophilicity of laboratory soot surrogates, encompassing particles derived from n-hexane, decane, and toluene diffusion flames. In fuel-rich burning conditions values of approximately $\theta_{sw} = 120°$
were found, largely independently of the fuel type. While the hydrophilicity of toluene was found to be almost insensitive to the fuel-to-oxygen ratio, the contact angles of n-hexane and decane flame soot significantly decreased under fuel-lean conditions, reaching values of 83°. This value is com-
parable to that observed by Zelenay et al. (2011), reporting $\theta_{sw} = 80 \pm 3°$ for untreated soot produced with a combustion aerosol standard (CAST; under fuel-lean conditions). Interestingly, Zelenay et al. (2011) reported an increase in the soot–water contact angle for the CAST soot after ex-
posure to ozone ($\theta_{sw} = 100 \pm 4°$) and ozone and UV light

($\theta_{sw} = 120 \pm 3°$). These results are in stark contrast to the findings of Wei et al. (2017), who reported a decrease in the contact angle due to ozone ageing. In their study, they took soot particles from a wood-burning fireplace, different bus engines, and a kerosene lantern and aged the pelletized soot by exposure to vapours of ozone, sulfuric acid, and nitric acid. They reported the contact angles of fresh, untreated soot to be between 65 and 110°. After 10 d of exposure to 40 ppb ozone all samples revealed a decrease in the contact angle of $\sim 15°$. Ageing with nitric acid and sulfuric acid was reported to be less effective.

This survey reveals that contact angles between atmospheric soot particles and water cover a wide range with the highest values for fresh soot from fuel-rich flames, which are clearly hydrophobic ($\theta_{sw} > 90°$), and the lowest values after strong evacuation ($\theta_{sw} \approx 35°$). Given this wide range, the soot–water contact angle is a highly relevant parameter that strongly affects the efficiency of soot PCF or even renders it irrelevant (if $\theta_{sw} > 90°$). For simplicity, we assume in this study that the surface of an individual soot aggregate has a uniform contact angle with water and ice.

## 3  Morphology and pore structure of model soot aggregates

To characterize and model the features of soot aggregate porosity, we treat the primary particles as perfect spheres of diameters $D_{pp}$ and allow them to partially overlap as quantified by the overlap coefficient $C_{ov}$. In order to derive pore structures relevant for PCF, we assume that the primary particles enclosing one specific pore are of uniform size and equal overlap, i.e. one $D_{pp}$ and $C_{ov}$. Using the concept of grain packing, the primary particles can be stacked in different packing arrangements such as tetrahedral and cubic as shown in Fig. 2. In this study, we use these two packing arrangements and investigate the ability of the associated pore types to form ice via PCF.

Figure 2 illustrates the pores arising from tetrahedral and cubic packing arrangements. Different types of pores form within one lattice layer or plane of spheres and between two overlaying planes of spheres. Within a single plane, pores form as the annular (ring-like) opening between the three spheres arranged on the corners of the equilateral triangle ABC in the case of the tetrahedral packing arrangement (Fig. 2a). This pore has the shape of a *concave triangle* (Fig. 2b), and we refer to it as a *three-membered ring pore*. The dimensions of the three-membered ring pore relevant for PCF are the circumcircle radius, $r_{c,tr}$, and the radius of the circle describing the opening of the concave triangle, $r_{o,tr}$. In the case of the cubic packing arrangement (Fig. 2d) the pore within a single plane is formed as the opening between the four spheres arranged on the corners of the square ABCD and therefore has the shape of a *concave square* (Fig. 2e). We refer to these voids as *four-membered ring pores*. The di-

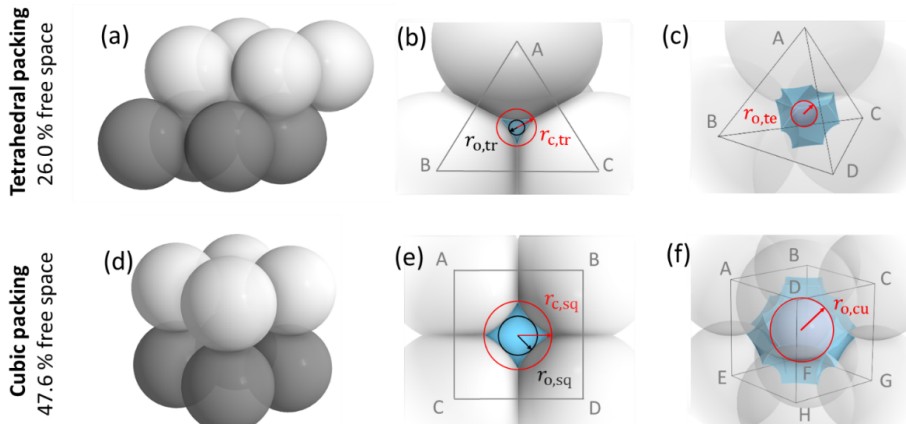

**Figure 2.** Overview of tetrahedral **(a)** and cubic **(d)** packing arrangements along with theoretical porosities when ordering uniformly sized spheres into a cubic unit cell, defined as the volume of the voids relative to the total volume of the sample unit cell. The porosity refers to zero overlap between rigid spheres and is independent of sphere size. Differently coloured spheres indicate spheres in different lattice planes of the arrangements. Panels **(b)** and **(e)** present an enlargement of the three-membered and four-membered ring pores along with key pore dimensions relevant for PCF as discussed in the text. Panels **(c)** and **(f)** reveal the inner cavities forming a concave octahedron (tetrahedral packing) and concave cube (cubic packing). See Figs. A1 and A2 for a more detailed view of panels **(c)** and **(f)**.

mensions relevant for PCF are the circumcircle radius, $r_{c,sq}$, and the radius of the circle describing the opening of the concave square, $r_{o,sq}$. Ring pores are narrowest at the midplane, formed by the centres of the spheres making up the ring, and widen with increasing distance from the midplane. In the following, we refer to the narrowest part of the pore as the pore neck.

Larger pores form between two overlaying (neighbouring) planes of primary particles. In the case of tetrahedral packing, a pore of the shape of a dented *concave octahedron* forms between four primary particles arranged on the corners ABCD of a regular tetrahedron (Figs. 2c, A1b). The four openings of the cavity within the tetrahedral cell form the three-membered ring pores described above. Its free space can be quantified by the radius of the maximum sphere within it, denoted by $r_{o,te}$. In the case of a cubic arrangement, the inner pore between eight spheres (A–H) across two lattice layers of primary particles has the form of a dented *concave cube*. Its free space can be quantified by the radius of the maximum sphere within it, denoted by $r_{o,cu}$. The cube formed between the eight primary particles has six openings on the faces of the cube, forming four-membered ring pores (Figs. 2f, A2b). A detailed description of the pore geometry along with a derivation of the relevant pore dimensions is given in Appendix A. In Fig. 3, we show the relevant pore dimensions for pores formed between primary particles depending on primary particle diameter, $D_{pp}$, and overlap, $C_{ov}$. In general, pore radii increase in size with increasing diameters of the primary particles for both the tetrahedral and the cubic packing arrangement. In the absence of overlap, pores are very open and become more constrained (narrow) with increasing overlap, thus improving their ability to host water on the one hand and reducing the enclosed water vol-

ume on the other hand. The overlap at which pores vanish is independent of primary particle diameter ($D_{pp}$) and just depends on the pore geometry. The opening of the three-membered ring pore vanishes, i.e. $r_{o,tr} = 0$, for $C_{ov} = 0.134$ (dashed lines in Fig. 3a). Conversely, the concave octahedron within the tetrahedral cell persists to $C_{ov} = 0.184$ (solid lines in Fig. 3a), although without connection to the surrounding for $C_{ov} > 0.134$, since at this overlap the three-membered ring pores forming the openings of the concave octahedron vanish. For the cubic arrangement, larger overlaps are required for both pore types to disappear compared to the tetrahedral packing arrangement. Specifically, the opening of the four-membered ring pore vanishes for $C_{ov} = 0.293$ ($r_{o,sq} = 0$; dashed lines in Fig. 3b) and the concave cube within the cubic cell vanishes only for $C_{ov} = 0.423$ ($r_{o,cu} = 0$; solid lines in Fig. 3b), as detailed in Appendix A.

In order to contribute to ice formation via PCF, pores need to be small enough to allow water uptake below water saturation and large enough to host a critical embryo along with pore openings that are sufficiently large to allow ice growth out of the pore so that macroscopic ice crystals can form. In the following, we first derive the size of a critical ice cluster and set it in relation to the pore dimensions shown in Fig. 3. This forms the baseline constraint for any soot pore to be ice active via PCF. We then go on and derive conditions for pore filling and discuss the growth of ice out of the pore.

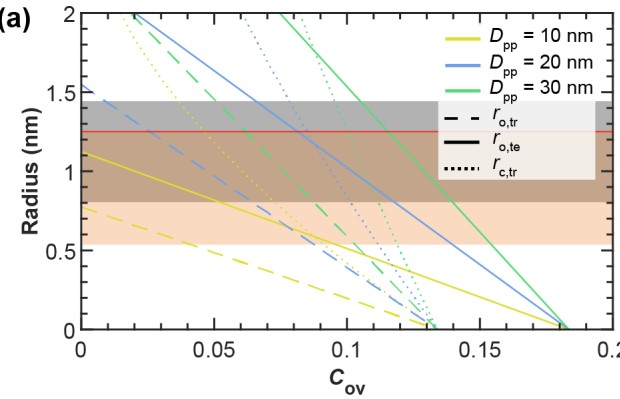

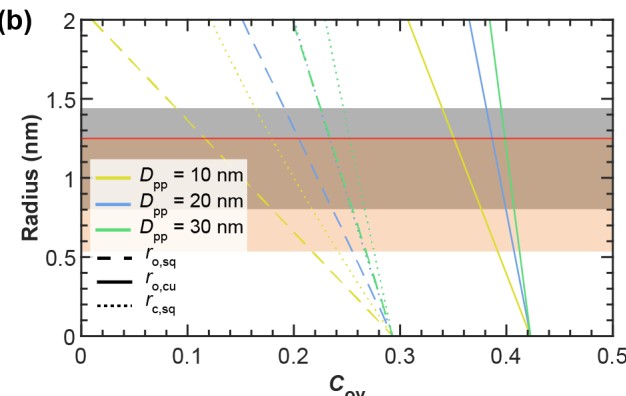

**Figure 3.** Overview of the radii of key pore dimensions (see Figs. 2, A1, and A2) formed in pores of **(a)** tetrahedral and **(b)** cubic packing arrangements between equally sized spherical primary particles of indicated diameter and as a function of primary particle overlap, with $C_{ov}$ as defined in Eq. (1). A derivation of the indicated pore dimensions can be found in Appendix A. For comparison, the grey- and orange-shaded regions mark the variation in the critical ice embryo radius, $r_{crit}$, for $200\,K \le T \le 235\,K$ TS7, as calculated based on Eq. (3) using the parameterizations of Murray et al. (2010) and Ickes et al. (2015), respectively. The horizontal red line denotes a pore radius of $r = 1.25\,nm$, corresponding to the smallest investigated pore size, for which ice formation was experimentally observed by David et al. (2020).

## 4 Ice nucleation in soot pores

### 4.1 Critical pore size for PCF

The nucleation of the thermodynamically stable phase (ice) within a metastable phase (supercooled liquid water) can be described by classical nucleation theory (Farkas, 1927; Fletcher, 1960; Gibbs, 1875). CNT assumes a spherical shape of the critical ice cluster (ice embryo), because a sphere minimizes the surface area for a fixed volume (Fukuta, 1966; Zaragoza et al., 2015). CNT describes the change in Gibbs free energy, $\Delta G_{w,i}(T, P)$, for the formation of a critical embryo as the sum of the volume energy released by passing from the thermodynamically metastable (supercooled) liquid

to the stable ice phase and the surface energy consumed by forming a new interface (Lohmann et al., 2016; Pruppacher and Klett, 1997) TS8:

$$\Delta G_{w,i}(TP) = \underbrace{4\pi r^2 \gamma_{iw}(T, P)}_{\text{Surface energy}} + \underbrace{\frac{4\pi r^3}{3v_i(T, P)} \Delta \mu_{iw}}_{\text{Volume energy}}. \quad (2)$$

We TS9 include here the absolute pressure dependence, as large negative pressures can be reached in narrow pores below water saturation with strong impact on homogeneous ice nucleation rates (Marcolli, 2020). Here, $\gamma_{iw}(T, P)$ denotes the interfacial tension between the ice and the water phase, $r$ is the radius of the evolving ice cluster, $v_i(T, P)$ TS10 is the molecular volume of water in the ice phase, and $\Delta \mu_{iw} = \mu_i(T, P) - \mu_w(T, P)$ TS11 represents the difference between the chemical potentials of ice and liquid water. Subcritical ice clusters randomly form within the supercooled liquid and grow continuously once they have passed the critical size (Lohmann et al., 2016). The critical radius $r_{crit}(T, P)$ of an ice embryo is reached when growth and shrinkage both lead to a decrease in the Gibbs free energy and can be determined by setting $\delta \Delta G_{w,i}(T, P)/\delta r$ to 0:

$$r_{crit}(T, P) = \frac{2\gamma_{iw}(T, P)v_i(T, P)}{-\Delta \mu_{iw}}. \quad (3)$$

The TS12 critical embryo radius decreases with decreasing temperature and pressure. Moreover, values depend on the CNT parameterization applied. It ranges between $r_{crit} = 0.53\,nm$ at ice saturation ($P = -55\,MPa$) at $200\,K$ to $r_{crit} = 1.25\,nm$ at water saturation ($P = 0.1\,MPa$) at $235\,K$, using the pressure-dependent parameterization from Ickes et al. (2015) as described in Marcolli (2020). Using the pressure-dependent CNT parameterization by Murray et al. (2010) with $n_{it} = 0.97$ instead, the range changes to $r_{crit} = 0.8\,nm$ at ice saturation at $200\,K$ and $r_{crit} = 1.43\,nm$ at water saturation at $235\,K$ (Marcolli, 2020). In Fig. 3, we show these ranges of critical ice embryo radii as shaded regions.

For a soot pore to be suitable for PCF, it must be large enough to host a spherical ice embryo of critical size. Comparison of the critical embryo size with the radii of the different pore types shown in Fig. 3 demonstrates that the pores of soot aggregates with atmospherically realistic primary particle diameters ranging from 10 to 50 nm are sufficiently large to accommodate a critical ice embryo. In particular, it becomes evident that for pores formed in the tetrahedral arrangement, low overlap is required to be relevant for PCF. As the overlap coefficient increases, the pores quickly become too narrow (falling outside the shaded regions in Fig. 3) to host an ice embryo. We note that even though the inner cavity within four tetrahedrally arranged spheres is large enough to host a critical embryo, its openings given by the three-membered ring pores may be too small to allow macroscopic ice growth out of the pore. Moreover, a quasi-liquid

layer has been shown to form between ice and the pore surface, which further increases the pore size required to host a critical ice embryo. This surface layer thickness has been quantified as about 0.7 nm in narrow (3.2–4.8 nm diameter) hydrophobic cylindrical pores at 200 K (Morishige, 2018), while molecular dynamics simulations have yielded a half width of 0.37 nm for the water layer adjacent to the hydrophobic surface of cylindrical pores at $T = 190$ K (Moore et al., 2012). Yet, the ability of very narrow pores to host ice has been supported by David et al. (2020), who observed ice growth out of methylated silica particles with average cylindrical pore diameters of 2.4 nm for temperatures of 223 and 228 K.

Another important aspect to be recognized from Fig. 3 is that the three-membered and four-membered ring pores characterized by $r_{o,tr}$ and $r_{o,sq}$ are sufficient by themselves to host a critical ice embryo. In other words, the relatively larger inner cavities denoted as concave octahedron CE2 (tetrahedral packing) and concave cubes (cubic packing) do not need to fill in order for a soot aggregate to be ice active. We therefore concentrate in the following on $n$-membered ring pore structures formed by soot aggregates with the main focus on three-membered and four-membered ring pores.

## 4.2 Nucleation rates in cirrus conditions

Even if a ring pore is wide enough to host ice, the pore water remains liquid when the ice nucleation rate is too low. As soot is a poor ice nucleator in the immersion freezing mode (Kanji et al., 2020), we exclude here the possibility of heterogeneous freezing and assume that the pore water needs to freeze homogeneously to form ice via soot PCF. Numerous experimental studies on micrometre-sized droplets have constrained homogeneous ice nucleation rates in the temperature range from 234 to 238 K to values below $10^{11}$ cm$^{-3}$ s$^{-1}$ (e.g. Ickes et al., 2015, and references therein). A few experimental studies have explored homogeneous ice nucleation rates at lower temperatures by investigating the freezing of water clusters with radii of a few nanometres (Amaya and Wyslouzil, 2018; Bartell and Chushak, 2003; Manka et al., 2012). These studies have reported values of above $10^{20}$ cm$^{-3}$ s$^{-1}$ for temperatures between 190 and 225 K. Conversely, Laksmono et al. (2015) only reached homogeneous ice nucleation rates of $10^{11}$ and $10^{13}$ cm$^{-3}$ s$^{-1}$ between 226 and 231 K investigating larger volumes by increasing the cooling rate to $10^6$–$10^7$ K s$^{-1}$. Since the water volume within soot pores hardly exceeds the critical size to produce an ice embryo, the nucleation rates need to be in the range of $10^{20}$ cm$^{-3}$ s$^{-1}$ to freeze water homogeneously within 1 s. Thus, the rates reported for $T = 234$–238 K by numerous studies and for $T = 226$–231 K by Laksmono et al. (2015) are too low for pore water to freeze homogeneously within atmospherically relevant timescales. Yet, water within the cylindrical pores ($D_p \approx 2$–10 nm) of mesoporous silica particles was able to grow into ice crystals within a few seconds ($\sim 10$ s) at 228 K but not at 233 K (David et al., 2019, 2020), implying that homogeneous ice nucleation rates were high enough at 228 K but not at 233 K. Also, soot particles proved to freeze according to a PCF mechanism at 228 K but not at 233 K (Mahrt et al., 2018a). In both cases, nucleation rates have been enhanced due to the negative pressure within the pores that develops below water saturation (Marcolli, 2017a, 2020). At 228 K, the negative pressure of $P = -43$ MPa at ice saturation increases the homogeneous ice nucleation rate by 5 orders of magnitude from $\sim 10^{15}$ cm$^{-3}$ s$^{-1}$ to $\sim 10^{20}$ cm$^{-3}$ s$^{-1}$ K, using the pressure-dependent parameterization by Murray et al. (2010) for $n_{it} = 0.97$.

Based on these findings, we can estimate a temperature range where freezing is immediate once the water volume is large enough to host a critical ice embryo, namely for $T \leq 228$ K, an intermediate temperature range where freezing is not immediate and strongly depends on RH ($228 < T < 233$ K) and a range where homogeneous ice nucleation rates are too low to trigger ice formation via PCF ($T > 233$ K). However, more dedicated experiments are needed to better constrain these ranges. In the following, we concentrate on the cirrus temperature range below $T = 228$ K, where it is safe to assume that freezing of pore water is immediate once its volume is large enough to host a critical ice embryo.

## 5 Capillary condensation within soot aggregates

Water vapour can condense in capillaries below bulk water saturation, leading to the formation of water surfaces with concave curvature. The radius of the curvature is related to the equilibrium vapour pressure above the curved surface through the Kelvin equation (Kelvin, 1904; Thomson, 1871). The Kelvin equation in turn relies on the Young–Laplace equation that quantifies the pressure in liquids with curved surfaces (see Appendix B). In soot particles, as in any porous medium, capillary condensation of water starts within the narrowest pore segments and gradually fills widening pores as the ambient RH increases. Capillary condensation of water within soot aggregates has been discussed in previous studies (e.g. Persiantseva et al., 2004). The narrowest and simplest cavity consists of the slit between two adjacent primary particles (see Fig. B3), where liquid water surrounds the contact point of the two spheres forming a pendular ring (see Fig. B3b), referred to as the pendular state (e.g. Afrassiabian et al., 2016; Urso et al., 1999). In the case of wetting surfaces (negligible contact angle with water), pendular rings form concave water surfaces with curvatures that are symmetric about the line connecting the centres of the two particles (Huang et al., 2015). Note that the curvature of the water surface switches from concave to convex with an increasing pore-filling angle for $\theta_{sw} > 0$ TS13. Pendular rings can be described by two principal radii of curvature related

to the height and the width of the water (pendular ring) above the plane connecting the centres of the two particles (filling level). Considering these two principal radii, the Kelvin equation describing the equilibrium vapour pressure over the curved surface formed by a pendular water ring (see Appendix B2 for details) can be expressed as

$$\frac{p}{p_{\mathrm{w}}} = \exp\left(\frac{\gamma_{\mathrm{vw}}(T)\left(\frac{2}{D_{\mathrm{pp}}\sin\varepsilon} + \frac{2\cos(\theta_{\mathrm{sw}}+\varepsilon)}{-D_{\mathrm{ij}}+D_{\mathrm{pp}}\cos\varepsilon}\right)v_{\mathrm{w}}(T)}{kT}\right). \quad (4)$$

Here, $p$ and $p_{\mathrm{w}}$ describe the water vapour pressure and the equilibrium water vapour pressure with respect to water, respectively, and their ratio $p/p_{\mathrm{w}} = S_{\mathrm{w}}$ denotes the saturation ratio with respect to water. Moreover, $k$ is the Boltzmann constant, $v_{\mathrm{w}}(T)$ is the molecular volume of liquid water, and $\gamma_{\mathrm{vw}}(T)$ is the interfacial tension between liquid water and water vapour. Finally, the angles $\theta_{\mathrm{sw}}$ and $\varepsilon$ correspond to the soot–water contact angle and the slit-filling angle, respectively.

The pendular rings of water between neighbouring particles may also be the starting point for pore filling in ring pores. As the pendular rings of water grow in height with increasing RH, the rings on neighbouring particles will meet, eventually leading to full coalescence of the individual water rings. We refer to this pathway as pore filling via pendular rings of water (Fig. 4). For the three-membered ring pore, gathering of the three water rings takes place at a slit-filling angle of $\varepsilon = 30°$ CE3. For four-membered ring pores, $\varepsilon = 45°$ is required as illustrated in Fig. 4. These threshold values are independent of the size of the spheres in between which the pores are formed (Afrassiabian et al., 2016). Further increase in the RH leads to a continuous growth of the water phase, ultimately causing complete filling of the ring pore. The state when a sufficient amount of liquid is available to fill the interparticle space (pore neck) is considered the capillary state (Urso et al., 1999).

In Fig. 5 we show the $\mathrm{RH_w}$ conditions for filling of three-membered and four-membered ring pores via pendular rings as a function of the soot–water contact angle as dashed lines. Filling conditions were calculated with Eq. (4), using fixed values of $\varepsilon = 30°$ (three-membered ring pore) and $\varepsilon = 45°$ (four-membered ring pore), for different primary particle diameters, overlap coefficients, and a typical cirrus temperature of $T = 220\,\mathrm{K}$. Relatively high $\mathrm{RH_w}$ values are required for the filling of a three-membered ring pore via pendular water rings. The $\mathrm{RH_w}$ required for pore filling decreases for decreasing contact angles and for increasing overlap coefficients. For instance, a perfectly wettable ($\theta_{\mathrm{sw}} = 0°$) three-membered ring pore formed between three particles of $D_{\mathrm{pp}} = 20\,\mathrm{nm}$ is filled at 66 % $\mathrm{RH_w}$ (dashed rose line) for a minimal overlap of $C_{\mathrm{ov}} = 0.01$ but already at 14 % $\mathrm{RH_w}$ for $C_{\mathrm{ov}} = 0.1$ (dashed purple line). Hence, in this case, an overlap of $C_{\mathrm{ov}} = 0.1$ can decrease the $\mathrm{RH_w}$ required for pore filling by $\sim 52$ percentage points. However, the $\mathrm{RH_w}$ for pore filling quickly rises with increasing contact angle. Three-

membered ring pores can only fill below bulk water saturation for soot–water contact angles below 56°, as indicated by the intercept of the dashed lines and the horizontal black line for $p/p_{\mathrm{w}} = 1$ (see also Fig. B4). The $\mathrm{RH_w}$ conditions for filling of a four-membered ring pore are shown in Fig. 5d–f in a similar fashion as for the three-membered ring pore. The criterion of $\varepsilon = 45°$ for complete filling limits filling of four-membered ring pores to particles with soot–water contact angles below 37° (intercept of the dashed lines and the horizontal black line for $p/p_{\mathrm{w}} = 1$ in Fig. 5d–f; see also Fig. B4).

In conclusion, the filling of ring pores (e.g. three-membered or four-membered) via pendular rings of water is strongly constrained by the contact angle, i.e. wettability of the pore material. These results suggest that ring pore filling via (coalescence of) pendular water rings is likely not an important mechanism for atmospheric soot particles, which tend to have higher contact angles (see Sect. 2.4). Yet, if the soot–water contact angles are too high for $n$-membered pore ring structures to become completely filled via coalescence of pendular water rings, complete pore filling can still take place through *direct pore filling*. This filling mechanism is illustrated in Fig. 4b and d for the three-membered and four-membered ring pores, respectively. Direct filling starts from random fluctuations of adsorbed water patches resulting in spontaneous condensation of water at the pore neck (narrowest part of a ring pore, which is located at the midplane formed by the centres of the spheres making up the ring) as illustrated in Fig. 4b and d. The Kelvin equation describing the equilibrium vapour pressure over the curved surface for direct filling of three-membered ring pores with liquid water (Appendix B4) is given by

$$\frac{p}{p_{\mathrm{w}}} = \exp\left(\frac{\gamma_{\mathrm{vw}}(T)\left(\frac{4\sqrt{3}\cos(\theta_{\mathrm{sw}}+\varepsilon)}{-2D_{\mathrm{ij}}+\sqrt{3}D_{\mathrm{pp}}\cos\varepsilon}\right)v_{\mathrm{w}}(T)}{kT}\right). \quad (5)$$

In Eq. (5) the shape of the curvature of the water surface is described by only one principal radius. This is because non-orthogonal, unequal radii of curvature will equalize and minimize the surface energy at equilibrium so that the hydrostatic pressure within the capillary condensate is always isotropic. This is consistent with the results of previous studies addressing the shape of capillary surfaces in non-circular geometries. For instance, Feng and Rothstein (2011) studied the equilibrium shape of curved surfaces in capillaries with $n$-sided, polygonal, cross-sectional areas, finding the shape of the curved surface to be spherical. A change in cross-sectional geometry is essentially equivalent to a change in the contact angle between the pore wall and the fluid. This is in line with more recent results from Son et al. (2016), who modelled the menisci formed in triangular and square tubes, equivalent to our three-membered and four-membered ring pores, and reported the liquid surface to form a hemisphere at equilibrium.

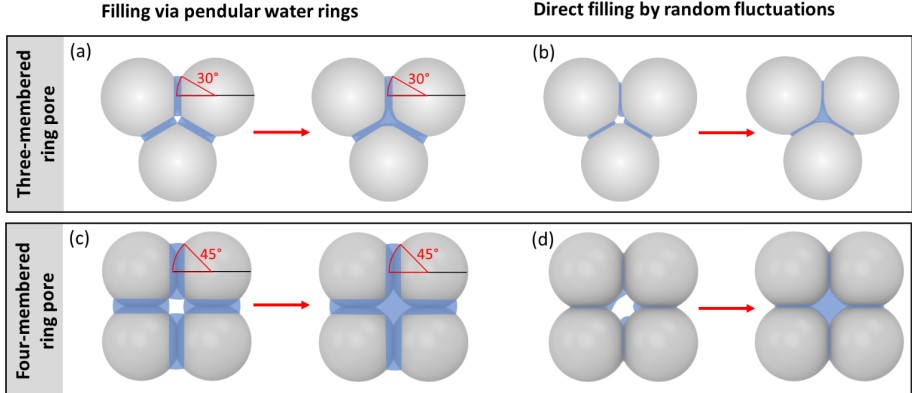

**Figure 4.** Illustration of the filling scenarios for **(a, b)** three-membered and **(c, d)** four-membered ring pores formed between spheres in a tetrahedral and cubic packing arrangement assuming $\theta_{sw}$ is 60°. In the three-membered and four-membered ring pore case, uptake of liquid water is considered either by *filling via the pendular rings of water* **(a, c)** or by *direct filling* **(b, d)**. Direct filling starts at the pore neck, defined as the narrowest point of the ring pore opening located at the midplane between the centres of the spheres. In the case of filling via the pendular water rings slit-filling angles of $\varepsilon = 30°$ and $\varepsilon = 45°$ (red marking) need to be reached for pendular rings to gather together and coalesce. Note that for $\theta_{sw} = 60°$, the water surface has a convex curvature at a filling angle of $\varepsilon = 45°$. If the RH is further increased, further filling of the ring pore as well as of the inner cavity (concave octahedron and concave cube) takes place.

In the case of four-membered ring pores, the Kelvin equation describing the equilibrium water vapour for direct pore filling (Appendix B5) is given by

$$\frac{p}{p_w} = \exp\left( \frac{\gamma_{vw}(T) \left( \frac{4\cos(\theta_{sw}+\varepsilon)}{-\sqrt{2}D_{ij}+D_{pp}\cos\varepsilon} \right) v_w(T)}{kT} \right). \quad (6)$$

The $RH_w$ conditions required for direct filling of three-membered and four-membered ring pores, as calculated with Eqs. (5) and (6), are shown in Fig. 5 as solid lines. Obviously, direct pore filling requires significantly lower $RH_w$ compared to pore filling via pendular water rings for both types of ring pores considered here (compare dashed and solid lines in each panel of Fig. 5). As an example, at $T = 220$ K, the $RH_w$ required for filling of a three-membered ring pore with $D_{pp} = 20$ nm and $C_{ov} = 0.01$, assuming a contact angle of $\theta_{sw} = 0°$ (60°), is $\sim 66\%$ (119 %) for pore filling via pendular rings of water and $\sim 31\%$ (55 %) for direct filling (dashed rose and solid rose lines, respectively). If the primary particles with $D_{pp} = 20$ nm forming the three-membered ring pore overlap with $C_{ov} = 0.1$, capillary condensation within the three-membered ring pore via direct filling can already take place at $RH_w < 5\%$ for $\theta_{sw} = 0°$ and at $RH_w \approx 12\%$ for $\theta_{sw} = 60°$. Thus, soot aggregates with appropriate pore properties can take up water not only in clouds but potentially throughout the troposphere, where $RH_w$ frequently cycles from $\sim 20\%$ to 80 % (Wallace and Hobbs, 2011). Comparing the conditions for direct filling of a three-membered ring pore (Fig. 5a–c) to those of a four-membered ring pore (Fig. 5d–f), it is evident that direct filling of four-membered ring pores requires higher $RH_w$. This is due to the larger size of the opening formed between four compared to three spheres, as

discussed in Sect. 3 (see also Fig. 3). In contrast to pore filling via the pendular water rings, direct pore filling exhibits a weaker dependence on the soot–water contact angle and remains effective below water saturation for $0 \leq \theta_{sw} \leq 90°$ (see Fig. 5). For contact angles of $\theta_{sw} > 90°$ the condensed liquid exhibits a convex curvature, which goes along with CE4 an equilibrium vapour pressure that is higher than that of bulk liquid water.

The solid lines in Fig. 5 denote the conditions for the formation of a continuous liquid water phase at the pore neck. However, in order to contribute to ice formation via PCF, the water volume within a pore must be sufficiently large to host a critical ice germ, as discussed in Sect. 4.1. We take this into account by calculating the $RH_w$ required for direct filling of the ring pores with water up to a filling level of $h_{fr}$, denoting the height that the water needs to extend vertically in both directions from the midplane (pore neck) for homogenous freezing of the capillary condensate. A value of $h_{fr} = 1.25$ nm is an approximate value that was chosen based on the experimental observations of David et al. (2020) and theoretical considerations taking calculated critical embryo size and quasi-liquid layer width into account as discussed in Sect. 4.1. The $RH_w$ values required to reach a filling level of $h_{fr}$ are given by the dotted lines in Fig. 5. For each set of dotted and solid lines, the $RH_w$ required to just fill the neck of the ring pore is lower compared to filling the ring pore up to a level of $h_{fr}$. This difference becomes larger with an increasing soot–water contact angle, as can be seen from the diverging solid and dotted lines in Fig. 5 for each set of curves. Moreover, the difference in $RH_w$ required for direct filling and direct filling up to $h_{fr}$ becomes more distinct for increasing overlap coefficients and decreasing primary particle diameters. It should be noted that in the absence of over-

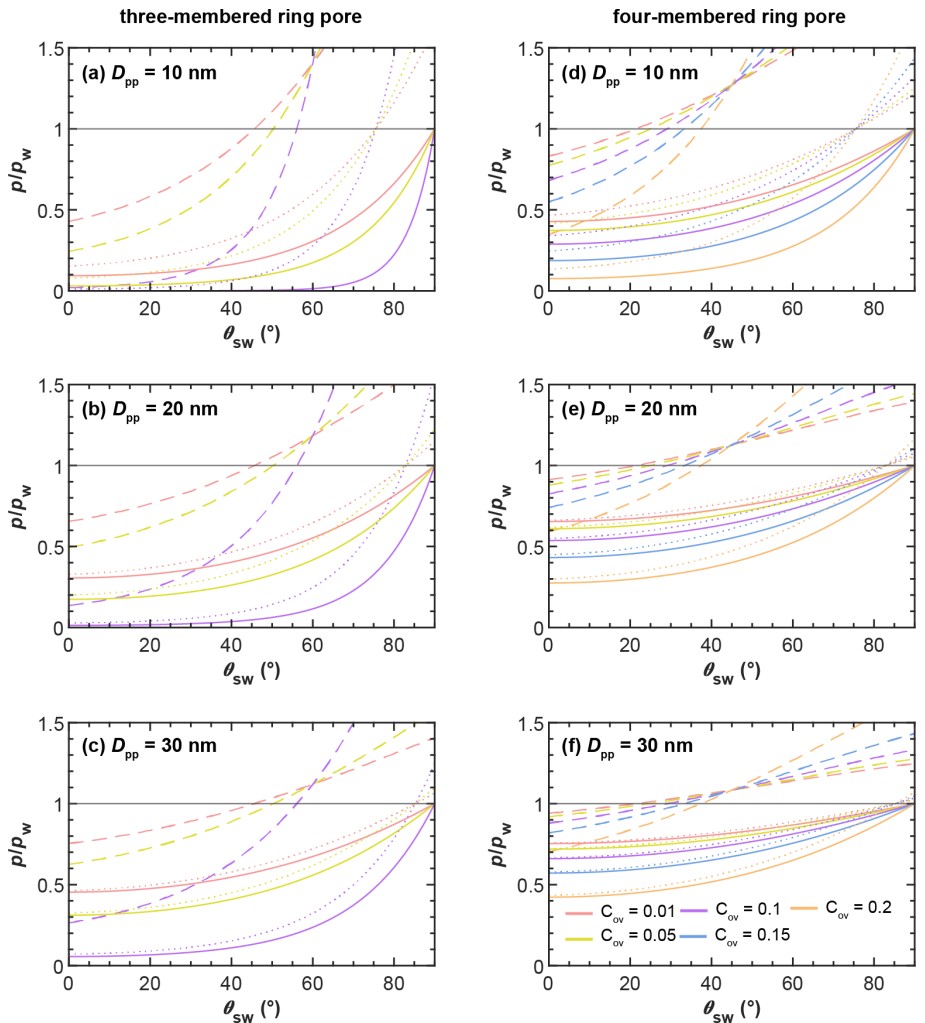

**Figure 5.** Pore-filling conditions for three-membered ring pores (tetrahedral packing, **a**, **b**, **c**) and four-membered ring pores (cubic packing, **d**, **e**, **f**) showing the saturation ratio with respect to water ($p/p_w = S_w$) required for filling of ring pores as a function of the soot–water contact angle ($\theta_{sw}$) at $T = 220$ K. The different rows correspond to ring pores formed for spheres with different diameters ($D_{pp}$). In each panel the $S_w$ needed for capillary condensation of water in the ring pore is compared for different filling mechanisms: slit filling up to $\varepsilon = 30°$ (three-membered ring pore) and $\varepsilon = 45°$ (four-membered ring pore) via pendular rings of water, followed by coalescence of water rings and complete filling of the ring pore is shown by the dashed lines. Direct filling of the ring pore neck is shown by the solid lines. Direct filling of the ring pore up to a height of $h_{fr} = 1.25$ nm, required for freezing, is shown by the dotted lines. The set of lines within each panel corresponds to different values for the overlap coefficient, $C_{ov}$, as indicated in the legend. The expressions for the Kelvin equations describing the conditions for pore filling for the various filling mechanisms are given in Appendix B.

lap, $h_{fr}$ cannot be reached. On the whole, we conclude that direct filling gives a realistic pathway of pore condensation for ring pores on soot aggregates that are characterized by considerable contact angles, while the pathway via pendular rings of water strongly overestimates the RH required for pore filling.

## 6   Ice growth out of the soot ring pores

In Sect. 5 we have shown that direct filling of ring pores with liquid water up to the freezing level ($h_{fr}$) may take place at RH$_w$ well below bulk water saturation. In this section, we determine the ice supersaturation required for the growth of the pore ice out of the ring pores into a macroscopic ice crystal.

In the following, we assume rigid soot pore structures that cannot facilitate ice growth out of narrow pore openings by expanding upon ice growth. We describe ice growth out of the pore by the same governing equations as for filling with water (see Appendix B), treating the ice growth as continuous filling of the pore throat with ice. The RH$_i$ of water vapour in equilibrium with the growing ice phase out of a three-membered ring pore can then be obtained by evaluat-

ing Eq. (B15) with respect to ice:

$$
\frac{p}{p_i} = \exp\left(\frac{\gamma_{vi}(T)\left(\frac{4\sqrt{3}\cos(\theta_{si}+\varepsilon)}{-2D_{ij}+\sqrt{3}D_{pp}\cos\varepsilon}\right)v_i(T)}{kT}\right). \tag{7}
$$

Here, $p$ and $p_i$ are the water vapour pressure and the equilibrium water vapour pressure with respect to bulk ice, respectively, and their ratio yields the saturation ratio with respect to ice ($S_i$). Moreover, $\gamma_{vi}(T)$ is the interfacial tension between ice and water vapour, $v_i(T)$ is the molecular volume of water in ice, and $\theta_{si}$ is the contact angle between ice and water vapour. Similarly, evaluating Eq. (B19) with respect to ice, the Kelvin equation describing the equilibrium vapour pressure for ice growth out of a four-membered ring pore is given by

$$
\frac{p}{p_i} = \exp\left(\frac{\gamma_{vi}(T)\left(\frac{4\cos(\theta_{si}+\varepsilon)}{-\sqrt{2}D_{ij}+D_{pp}\cos\varepsilon}\right)v_i(T)}{kT}\right). \tag{8}
$$

The interfacial tension between ice and vapour is not well known. Following previous studies, we assume a quasi-liquid layer to form on the ice surface (Gelman Constantin et al., 2018; Wettlaufer, 1999). In this case, an upper limit of the interfacial tension of ice and water vapour is given by the sum of the interfacial tensions of water and vapour and ice and water: $\gamma_{vi}(T) = \gamma_{vw}(T) + \gamma_{iw}(T)$, following David et al. (2019). Their molecular dynamics simulation showed an ordered arrangement of an ice phase growing out of a silica pore that can be interpreted as a contact angle between the two solid phases of the ice and the silica particle. This approach allows the substitution of $\gamma_{vi}(T)$ in Eqs. (7) and (8) by the sum of $\gamma_{vw}(T)$ and $\gamma_{iw}(T)$. Similarly, the contact angle between ice and soot is not well constrained by measurements and an upper limit can be estimated by assuming a quasi-liquid layer between pore ice and the soot surface, as described in Appendix C. The presence of a quasi-liquid layer between pore ice and the (hydrophobic) pore wall material has been confirmed by measurements (Morishige, 2018) and molecular dynamics simulations (Moore et al., 2012).

Figure 6a and d show the equilibrium vapour pressure with respect to ice, $p/p_i = S_i$, above the condensing water and growing ice phases during PCF in three-membered and four-membered ring pores as a function of growth angle $\varepsilon$ as illustrated in panels c and f and evaluated using Eqs. (5)–(8). In panels b and e, soot PCF is exemplified for ring pores with $D_{pp} = 20$ nm and contact angles of $\theta_{sw}(\theta_{si}) = 60°$ (78°) with $C_{ov} = 0.05$ for the three-membered ring pore and $C_{ov} = 0.1$ for the four-membered ring pore. The curves for capillary condensation and pore ice growth for these pores are shown as solid rose lines in panels a and d (Fig. 6). Curves for other contact angle values are given as dashed lines for comparison. The curves start at the $p/p_i$ required for direct pore filling by water followed by an increase in $p/p_i$ up to the point

where a filling level of $h_{fl} = h_{fr} = 1.25$ nm (given as stage A in Fig. 6) is reached. Once $h_{fr}$ is reached ice nucleation of the pore water takes place and the governing (growth) equations have to be evaluated with respect to ice to describe further growth. The switch from water to ice growth is marked by the kink in the curves and also indicated by the horizontal black and green arrows on top of panel a in Fig. 6. During this first step of water condensation and condensational growth, the slit-filling angle increases from $\varepsilon = 0°$ to $\varepsilon = \varepsilon_{fr}$. The second step of ice growth by water vapour deposition is described by a further increase in the growth angle from $\varepsilon_{fr}$ to $\varepsilon = 180°$. Using $\varepsilon$ as the growth coordinate assumes that the contact angle between ice and soot is maintained throughout the ice growth, determining the ice surface curvature. This growth coordinate is able to capture the specific characteristics of the ice growth although it neglects deviations from the prescribed curvature that occur when the ice caps growing out of the two opposite openings merge as is the case for the three-membered ring pores (see panel b, frame D). During ice growth, $p/p_i$ again increases and passes through a maximum before it transforms into depositional growth of an ice crystal. Inspection of Fig. 6 shows that the maximum in the ice growth curves is for $\varepsilon < 30°$, i.e. for ice growth within the pore, implying that it is the local pore geometry that determines the supersaturation required for PCF on soot and not the overall aggregate structure. Along each curve, the filling or growth step that is associated with the highest saturation ratio (global maximum) is the limiting step and determines the critical supersaturation for soot PCF. For the cases shown in Fig. 6, PCF for the three-membered ring pore is limited by ice growth while for the four-membered pore it is limited by water condensation up to $h_{fr} = 1.25$ nm. More specifically, the maximum RH$_i$ of the filling and growth steps of the three-membered ring pore are 86 % and 137 %, respectively (solid rose line in Fig. 6a). Conversely, the filling-limited four-membered ring pore has maxima of 129 % RH$_i$ and 124 % RH$_i$, for the filling and ice growth steps, respectively (solid rose line in Fig. 6d).

Figures D1 and D2 show additional PCF curves for other primary particle sizes and overlaps. This overview shows that soot PCF is sensitive to contact angles, primary particle size, and overlap. Overall, the narrower three-membered ring pores tend to be ice growth limited, while the wider four-membered ring pores are rather limited by water condensation. Both increasing overlap and decreasing primary particle size shift the limiting step of soot PCF from capillary condensation to pore ice growth. With an increasing contact angle, the RH required for both pore condensation and ice growth shifts to higher values. For instance, for a given contact angle of $\theta_{sw}(\theta_{si}) = 45°$ (68°), the onset of soot PCF is expected already at $S_i \approx 1.15$ (for four-membered ring pores with $D_{pp} = 30$ nm and $C_{ov} = 0.2$). This onset shifts to $S_i \approx 1.21$ for $\theta_{sw}(\theta_{si}) = 60°$ (78°), to $S_i \approx 1.27$ for $\theta_{sw}(\theta_{si}) = 70°$ (85.5°), and to $S_i \approx 1.35$ for $\theta_{sw}(\theta_{si}) = 78°$ (92°).

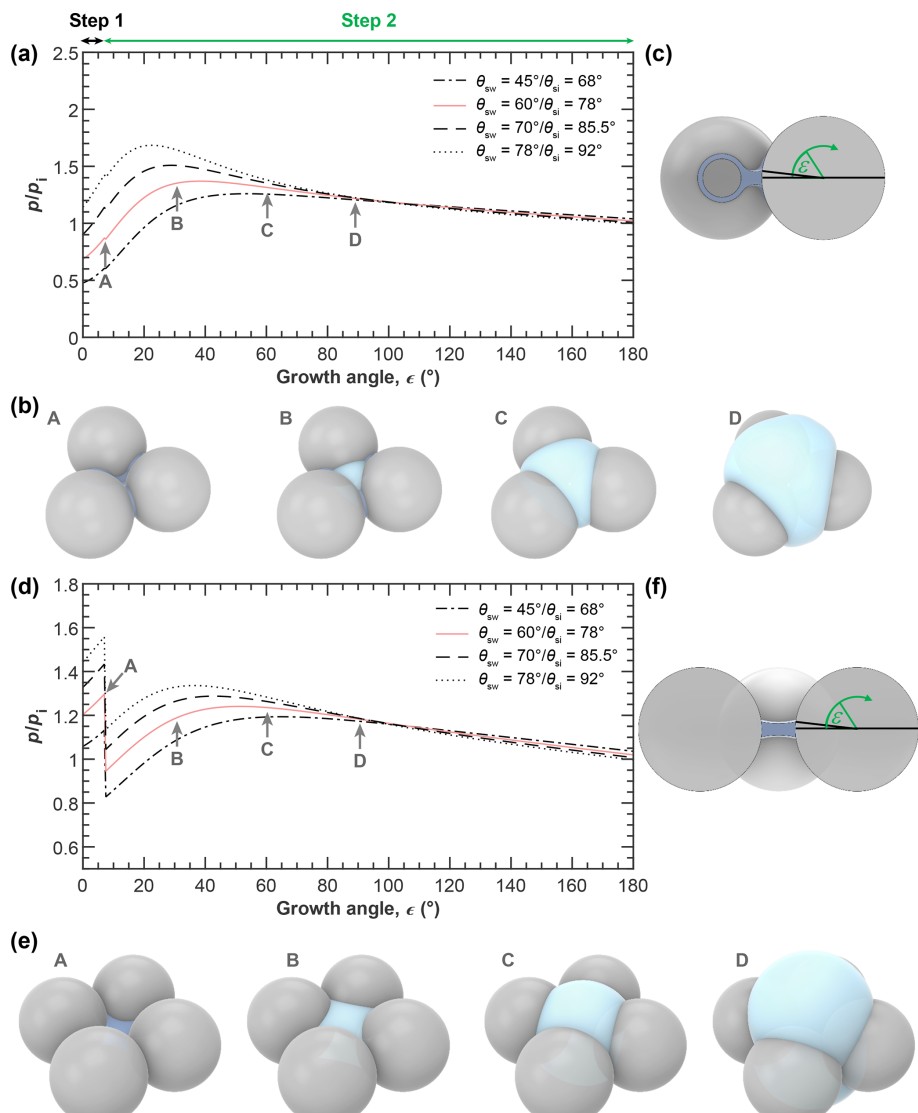

**Figure 6.** PCF by three-membered (**a–f**) and four-membered (**d–e**) ring pores. (**a, d**) Saturation ratio with respect to ice ($p/p_i = S_i$) required for (step 1) direct filling of the ring pore up to a filling level $h_{fl} = h_{fr} = 1.25$ nm, followed by (step 2) growth of pore ice as a function of the growth angle $\varepsilon$, as illustrated in the side views (panels **c** and **f**). The 3D renderings illustrate the water growth up to $h_{fr} = 1.25$ nm (A: darker blue; $\varepsilon$ is 7.18°) and ice growth (light blue; $\varepsilon$ is 30° (B), 60° (C), and 90° (D)) for a three-membered ring pore with $D_{pp} = 20$ nm, $C_{ov} = 0.05$, and $\theta_{sw}(\theta_{si}) = 60°$ (78°, panel **b**) and a four-membered ring pore with $D_{pp} = 20$ nm, $C_{ov} = 0.1$, and $\theta_{sw}(\theta_{si}) = 60°$ (78°, panel **e**). All soot-PCF curves were calculated for $T = 220$ K, $D_{pp} = 20$ nm, and different contact angles as indicated in the legend. See Figs. D1 and D2 for soot-PCF curves encompassing different primary particle sizes and overlaps.

## 7 Constraints on stability of pore ice and pre-activation

The ability of aerosol particles to form macroscopic ice crystals at lower relative humidities and/or higher temperatures compared to their intrinsic ice nucleation efficiency after previously having experienced an ice nucleation event or low temperatures is called pre-activation (Fukuta, 1966; Mason, 1950; Mossop, 1956; Pruppacher and Klett, 1997). Many or even most observations of pre-activation can be explained by pore ice that has nucleated in a preceding activation step/cloud cycle (Marcolli, 2017b), allowing ice crys-

tal formation by pure depositional growth when the particles re-encounter ice supersaturation. Yet, some observations of pre-activation may also be explained by the contact of soot aggregates with ice or liquid water changing their (pore) morphology or modifying surface functionalization (Bhandari et al., 2019; China et al., 2015; Colbeck et al., 1990; Ma et al., 2013; Miljevic et al., 2012; Zuberi et al., 2005) and thus enhancing their ice nucleation activity in subsequent cloud cycles, even in the absence of ice pockets preserved in pores (Mahrt et al., 2020b). Here, we focus

on pre-activation through ice preserved in pores. This pre-activation mechanism is most effective when the ice nucleation rate is low and the presence of pore ice enables the bypassing of the nucleation step. This is the case when pore ice that formed through homogeneous ice nucleation at low temperatures persists to temperatures above the HNT, enabling growth to macroscopic ice when (re-)encountering ice supersaturated conditions. Soot proved to be susceptible to this pre-activation mechanism in a study by Wagner et al. (2016). Specifically, they found soot, which was not ice active at temperatures above the HNT, to produce a small ice fraction even up to $T = 250$ K, after having undergone an initial cloud processing cycle at 228 K, i.e. well below the HNT.

Ring pores have the potential to show pre-activation as ice may persist in the pore necks at RH well below ice saturation and, when ice supersaturation is reached, grow out of the pore. Inspection of the soot-PCF curves displayed in Figs. 6, D1, and D2 shows that the ability of a pore to show pre-activation depends on the pore type (three-membered or four-membered) and primary particle size and overlap, as well as on the contact angle of soot with water and ice. For ice preserved in narrow pores with high contact angles, growth is only expected at RH above water saturation, implying that cloud droplet activation should precede ice crystal growth. This was indeed observed by Wagner et al. (2016), where the majority of the pre-activated soot particles that produced ice at $T = 250$ K froze in condensation mode. Note that the critical embryo size increases with increasing temperature. Therefore, the soot-PCF curves shown in Figs. 6, D1, and D2 must be re-calculated at the ice growth temperature to judge whether the ice is able to persist in the pore. Indeed, pre-activation to temperatures well above the HNT relevant for glaciation of MPCs requires large cavities with small openings that need to open up to release the ice contained in them during cloud droplet activation. Thus, the larger inner cavities between four tetrahedrally or eight cubically arranged primary particles (see Fig. 2 for cavity shapes and Fig. 3 for their radii) are the ones that are likely best suited to preserving ice up to higher temperatures.

In cirrus conditions, when ice nucleation rates are high, pre-activation is not expected to increase the fraction of ice-active soot aggregates, but its main effect is to reduce the supersaturation required for soot PCF to result in macroscopic ice crystals. When homogeneous ice nucleation rates are high, PCF is limited by either water condensation or ice crystal growth. In the case of pre-activation, since the pores are already filled with ice, the supersaturation required for pore condensation is irrelevant and ice growth becomes the limiting step. Thus, in cases where soot PCF is ice growth limited, pre-activation has no effect. Conversely, in cases where capillary condensation of pore water constitutes the limiting step of soot PCF, pre-activation lowers the supersaturation required for soot PCF to the value for ice growth. Moreover, for pre-activation, pore ice needs to remain thermodynamically stable at $RH_i < 100$ % so that it can grow by vapour depo-

sition upon (re-)encountering ice supersaturation. Whenever $n$-membered ring pores in soot aggregates fulfil both conditions, i.e. soot PCF is capillary condensation limited and ice remains in pores to $RH_i < 100$ %, pre-activation should be relevant. Figures 6, D1, and D2 exemplify for which combinations of primary particle diameters, overlap coefficients, and contact angles these conditions are met. It can be seen that pre-activation has a limit for high contact angles, as for $\theta_{si} > 85.5°$ pore ice is not preserved at $RH_i < 100$ % for any three-membered or four-membered ring pore considered here. Since three-membered ring pores tend to be narrow, PCF is for most of them constrained by ice growth, and pre-activation has only a marginal effect or no effect at all. The only exceptions are combinations of large primary particles ($D_{pp} > 30$ nm) and small overlap ($C_{ov} \leq 0.05$). Pre-activation in four-membered ring pores is more prevalent and may occur when $D_{pp} > 20$ nm and/or $C_{ov} \leq 0.2$ and is able to reduce the supersaturation required for macroscopic ice growth by up to 40 % $RH_i$ at $T = 220$ K. Yet, in ring pores for which soot PCF is limited by water condensation, ice is only preserved to 80 %–90 % $RH_i$. To keep pore ice stable to a lower RH, larger pores need to fill with ice, such as the larger inner cavities between four tetrahedrally or eight cubically arranged primary particles. However, these cavities are rarer and require larger soot aggregate sizes. Nonetheless, we note that by taking into account the complex pore morphology of fractal soot aggregates, the framework presented herein is able to capture the hysteresis of soot particles between water uptake and release as previously reported (e.g. Dubinin, 1980; Mahrt et al., 2018a; Popovicheva et al., 2008b), which cannot be described by approximating soot pores as cylindrical cavities that lack hysteresis.

Overall, pre-activation is most important when aerosol particles that were activated under cirrus conditions reach MPC conditions and the ice preserved in pores can grow to macroscopic ice crystals. Moreover, our results suggest that pre-activation in cirrus conditions is less relevant than previously estimated (e.g. Zhou and Penner, 2014), as it does not increase the number of ice-nucleating soot particles but just reduces the supersaturation required for activation for a minor fraction of the atmospheric soot particles. This confirms that the increased ice nucleation activity in cirrus conditions after cloud processing as observed by e.g. Mahrt et al. (2020b) is due to changes in the surface functionalization and morphology of soot aggregates (compaction) and not caused by pre-activation through ice preserved in pores.

## 8 Parameterizing soot PCF for ice nucleation on soot aggregates

In the previous sections we have shown that $n$-membered ring pores form cavities that are able to endow soot aggregates with ice nucleation activity. We now use these findings to explain ice nucleation experiments performed with

different soot types. In Fig. 7 we show activated fractions as a function of RH of size-selected soot particles as determined in a continuous-flow diffusion chamber (CFDC; Mahrt et al., 2018a, 2020b). The AF describes the cumulative fraction of total soot particles of a given size that formed macroscopic ice crystals larger than 1 μm in (optical) diameter within the ice chamber in given RH and $T$ conditions, as counted with an optical particle counter (see Mahrt et al., 2018a, for details). We constrain the discussion to data obtained from two soot types with vastly different properties, namely FW200, taken from Mahrt et al. (2018), and miniCAST black soot, taken from Mahrt et al. (2020b). FW200 is a commercially available soot (Orion Engineered Carbons GmbH, Frankfurt, Germany) with an average primary particle diameter of $D_{pp} = 22 \pm 3.9$ nm and fractal dimension of $D_f = 2.35$, as measured by Mahrt et al. (2018). The miniCAST black soot is a propane flame soot, generated in a diffusion flame under fuel-lean conditions with average $D_{pp}$ equal to $31 \pm 5.9$ nm and $D_f$ equal to 1.86 (Mahrt et al., 2018a). Furthermore, the water sorption measurements by Mahrt et al. (2018) indicate that the water uptake capacity of the miniCAST black is significantly lower compared to that of the FW200 soot. Thus, miniCAST black should represent a soot with a higher soot–water contact angle compared to FW200 (Mahrt et al., 2018a). Both soot types taken together cover a wide range of properties relevant for ambient soot particles (see Sect. 2.1). In Fig. 7 we show the AF as a function of both $RH_i$ and $RH_w$, for aggregates with electrical mobility diameters of between $D_m = 400$ nm and $D_m = 100$ nm and for typical cirrus temperatures of $T = 223$ K (crosses) and $T = 218$ K (circles). Interestingly, for miniCAST black, the AF curves for $T = 218$ K and $T = 223$ K collapse onto the same line when plotted as a function of $RH_w$ (see Fig. 7b) but are clearly offset when plotted as a function of $RH_i$. This behaviour suggests that the filling of the ring pore structures with water is the limiting step for miniCAST black particles for soot PCF (see Sect. 6). This is consistent with our prediction that critical supersaturations for three-membered and four-membered ring pores with a comparable primary particle size of $D_{pp} = 30$ nm tend to be determined by capillary condensation (see bottom row in Figs. D1 and D2). Conversely, the AF curves of the 400, 300, and 100 nm aggregates of the FW200 soot at $T = 218$ K and $T = 223$ K do not overlap when plotted as a function of $RH_w$, while the curves clearly overlap when plotting them as a function of $RH_i$. The only exception is the 200 nm aggregates, which show deviations on both scales. This demonstrates that soot PCF by the ring pores of the FW200 soot is not limited by filling with liquid water but by ice growth out of the pores instead. Considering our predictions for pore filling and ice growth in three-membered and four-membered ring pores formed between particles with $D_{pp} = 20$ nm suggests that the PCF activity of FW200 soot results primarily from three-membered ring pores, since four-membered ring pores tend to be limited by water uptake into the pores (i.e.

the filling step; see Figs. 6, D1, and D2). The onset of freezing for both soot types is clearly above ice saturation, as is expected for pore surfaces with rather high contact angles to water and ice. Since the detection limit for this experimental setup is reached at $AF = 10^{-4}$–$10^{-3}$ as evidenced by the increased scatter, we consider the RH when values above this AF range are reached as onset. In a recent study, David et al. (2020) observed a shift in freezing onsets from about 115 % $RH_i$ to 120 %–135 % $RH_i$ when the surface of hydroxylated mesoporous silica particles ($\theta_{sw} = 41$–$45°$) was methylated ($\theta_{sw} = 75$–$80°$). Inspection of Figs. D1 and D2 shows that the freezing onset of 400 nm miniCAST black aggregates of about 125 % $RH_i$ is in accordance with approximately $\theta_{sw}(\theta_{si})$ of 70° (85.5°), while a freezing onset of about 116 % $RH_i$ as observed for 400 nm FW200 aggregates conforms with contact angles $\theta_{sw}(\theta_{si})$ of approximately 60° (78°). Overall, the theoretical framework developed here allows the description of ice formation on soot aggregates and helps to explain differences in the ice nucleation ability of soot particles reported in the literature.

In order to use the new framework to predict soot PCF via $n$-membered ring pores, the abundance of such pore structures within soot aggregates needs to be considered. For this reason, we have developed an equation to quantify the ice activated fraction as a function of RH that reflects aggregate and pore properties:

$$AF(RH) = 1 - (1 - P_N(RH))^{\left((N_p - n_m)^{D_f}\right)}. \tag{9}$$

Here, $P_N(RH)$ is the probability of a primary particle being part of a ring pore that induces ice nucleation and growth at a given RH in a deterministic fashion. $N_p$ denotes the number of primary particles within the soot aggregate, and $D_f$ is the fractal dimension. The exponent $\left(N_p - n_m\right)^{D_f}$ takes into account the number of primary particles available within an aggregate to form a ring pore, $N_p$, as well as their spatial/morphological arrangement, via the fractal dimension, $D_f$. The derivation of Eq. (9) is elaborated in more detail in Appendix E.

The active-site probability function $P_N(RH)$ can be derived from first principles to reflect the properties (contact angle, primary particle size and overlap) of a specific soot type (e.g. aviation or diesel car emissions), or it can be treated as a fit function to bring the AF parameterization into agreement with measured datasets. Depending on the soot properties and the ice nucleation data, $P_N(RH)$ can be parameterized in terms of relative humidity with respect to ice ($P_N(RH_i)$) or water ($P_N(RH_w)$). We note that even if $P_N(RH)$ is determined through fitting to an existing dataset, it can be used along with Eq. (9) to predict the ice nucleation activity of soot particles with different properties (fractal dimension, number and size of primary particles).

To obtain the fraction of soot aggregates with suitable ring pores, the probability of a primary particle being part of a ring pore is potentiated by the number of primary particles

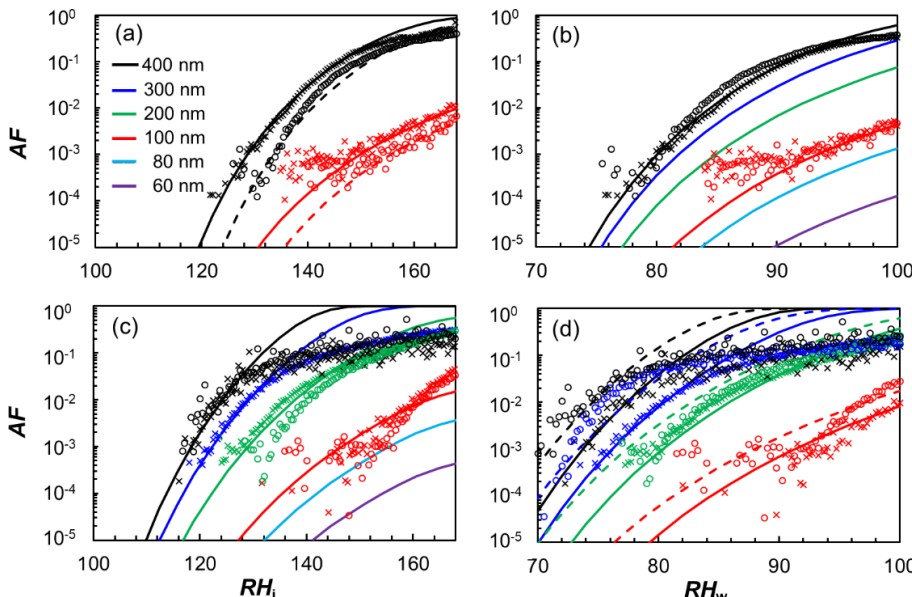

**Figure 7.** Fraction of soot particles that nucleate ice, termed activated fraction (AF) CE5 (at temperatures of $T = 218$ K (circles) and $T = 223$ K (crosses)), as a function of relative humidity with respect to ice (RH$_i$; **a, c**) and water (RH$_w$; **b, d** TS14) for **(a, b)** miniCAST black soot and **(c, d)** FW200 soot. The different colours denote soot aggregates of different electrical mobility diameters $D_m$ as indicated in the legend. Data are taken from Mahrt et al. (2018) for FW200 and from Mahrt et al. (2020b) for miniCAST black. The solid and dashed lines show the parameterization for 223 and 218 K, respectively. Note that in the case of miniCAST black the curves at 218 and 223 K coincide when plotted as a function of RH$_w$, while in the case of FW200 the curves coincide when plotted as a function of RH$_i$. See Appendix E for details.

contained within the soot aggregate ($N_p - n_m$). $N_p$ in turn can be calculated from defining relations of fractal aggregates based on the primary particle size and the fractal dimension (see Appendix E). Furthermore, the number of neighbouring spheres surrounding a primary particle is considered in terms of the fractal dimension $D_f$. To account for the minimum number of primary particles required to form a ring pore, $n_m$ is subtracted from the total number of primary particles within the aggregate, $N_p$. In the case of three-membered and four-membered ring pores, the value of $n_m$ is 2 and 3, respectively. The value of $n_m$ is relevant to correctly represent the smallest soot aggregates that may still contain a pore and ensures AF to be nonzero even for a small number of primary particles within an aggregate, as long as ring pores can be formed. The inclusion of primary particle number and compaction in Eq. (9) allows the parameterization of the ice nucleation activity of differently sized soot aggregates using the same active-site probability function.

Because soot properties such as the primary particle size and the fractal dimension can be readily measured, this soot-PCF framework can be directly compared to and constrained by experimental data. The lines shown in Fig. 7 are based on Eq. (9) using the experimental soot characterization and the ice nucleation data as detailed in Appendix E. The numbers of primary particles in soot aggregates as a function of their aggregate size are detailed in Table E1, while Table E2 lists the parameter values that were used in Eq. (9) to model the AF data depicted in Fig. 7. Since for FW200, activated fractions measured at 218 and 223 K collapse on one line when plotted as a function of RH$_i$, we parameterized the active-site probability function in terms of RH$_i$. Since the conversion from RH$_i$ to RH$_w$ is temperature dependent, AF splits into two separate curves, one for 218 K and one for 223 K, when plotted as a function of RH$_w$. Conversely, since the AFs at 218 and 223 K of miniCAST black coincide when plotted as a function of RH$_w$, we parameterized the active-site probability as a function of RH$_w$.

The differently coloured curves shown in Fig. 7 denote activated fractions of soot aggregates with different mobility sizes. Since the activated fraction of aggregates from a given soot type relies on the same active-site probability function ($P_N$(RH)), AF curves for mobility diameters that have not been measured can easily be derived from Eq. (9). Thus, activated fractions for lower supersaturation ratios and smaller soot aggregate sizes than have been measured, e.g. because they were below the detection limit, can be derived as long as data for larger aggregates and/or higher supersaturations are available. Moreover, a parameterization can be established even in the absence of ice nucleation data, when a soot sample has been characterized with respect to the aggregate size and compaction, the primary particle diameter and overlap, and the contact angle with water and ice (see Appendix E). The active-site probability (fit) function depends on the soot particle properties, and measurements thereof are often associated with some degree of uncertainty. For instance, Anderson et al. (2017) quantified the uncertainty in a single mea-

surement of the mean soot primary particle diameter derived from TEM images to be $\pm 14\%$. We note that Eq. (9) can easily be adapted to consider different primary particle diameters to describe the same AF curves of the different soot aggregate sizes shown in Fig. 7, again using a single fit function ($P_N$(RH)). For example, assuming a lower mean primary particle diameter increases the number of primary particles within a soot aggregate and allows smaller soot aggregates to be active via soot PCF. In this case, a fit to the data is achieved by lowering the probability of a primary particle being part of a ring pore that induces PCF at a specific RH. Conversely, the spacing between the soot-PCF curves of differently sized aggregates in Fig. 7 is not a free parameter but determined by the fractal dimension of the soot aggregates. Therefore, complying with the AF curve measured for 100 nm FW200 particles implies full activation (AF = 1) for $D_\mathrm{m} = 300$ nm at $\mathrm{RH_i} \approx 155\%$ for $T = 223$ K or at lower saturation ratios for larger soot aggregates. This suggests that the levelling off observed for the experimental data points at AF $\approx 0.2$ results from instrumental limitations. Specifically, for the data depicted in Fig. 7, which were sampled with a CFDC, it could result from divergence of particles out of the region of highest RH conditions within the CFDC, the so-called aerosol lamina, as has been demonstrated in previous studies (e.g. DeMott et al., 2015; Garimella et al., 2017). Overall, the good agreement of the parameterized curves with the measured data shown in Fig. 7 is compelling evidence that the active-site probability function provides an excellent means to describe PCF on soot aggregates of different sizes when combined with relevant aggregate and pore properties. The physical basis of the soot-PCF parameterization permits easy and fast adaptation to different types of atmospherically relevant soot, while parameterizations based on ice nucleation active surface site (INAS) densities are mostly restricted to the data underlying their parameterization. Consequently, INAS-based parameterizations cannot easily account for the wide diversity of soot types and their ice nucleation activities. Moreover, INAS-based parameterizations, such as the one by Ullrich et al. (2017), assume a scaling of ice nucleation sites with the particle (aggregate) surface area. Yet, the size-resolved measurements displayed in Fig. 7 exemplify that the ice-active particle fraction declines much more strongly with particle diameter than the surface area does. This is even more important considering that laboratory studies often concentrate on soot aggregates with mobility diameters much larger than typically found in the atmosphere (see Sect. 2.3), likely resulting in an overestimation of the effect soot has on cirrus formation. Conversely, the soot-PCF parameterization has a physically constrained particle size dependence and more importantly assumes pores, not the aggregate surface area, determine the ice nucleation activity of soot aerosols. This is an advantage over previously proposed parameterizations that often rely on datasets that were specific to an aggregate size and/or a single emission source.

Soot-PCF parameterizations in the form of Eq. (9) can be derived from first principles using activation curves such as the ones shown in Appendix D, presuming a contact angle combined with characteristic distributions of primary particle sizes and overlap coefficients. Weighting the ice activation RH for each combination of primary particle size and overlap coefficient with its occurrence probability leads to an ice activation probability distribution as a function of RH. If PCF is water condensation limited, the parameterization should be formulated based on $\mathrm{RH_w}$; if it is ice growth limited, it should be formulated with respect to $\mathrm{RH_i}$. Moreover, the temperature below which PCF becomes active needs to be defined. Assuming that soot is not ice active in condensation and/or immersion freezing mode, this threshold temperature depends on the increase in the homogeneous ice nucleation rate with decreasing temperature and should be around 230 K, given the small water volumes involved in forming ice via soot PCF. Yet, more ice nucleation experiments are needed in this temperature range with ice nucleation active soot samples to better constrain this threshold temperature.

## 9 Atmospheric implications

The framework presented here links the ice nucleation ability of soot with its physicochemical properties, namely primary particle size and overlap, aggregate size, and contact angle with water and ice, and in doing so provides a physical basis to describe the ice nucleation mechanism by pore condensation and freezing for soot.

Given that the ice nucleation activity of soot particles is linked to pores that form between primary particles, the framework presented here allows us to CE6 better constrain the properties required for atmospheric soot particles to form ice via PCF. For instance, at least three primary particles are required to form the smallest-conceivable CE7 pore (the three-membered ring pore). Hence, the minimum soot particle (aggregate) size to nucleate ice via PCF should exceed the average primary particle size of a soot sample. Moreover, soot spherules need to be arranged in a closed loop to form a ring pore or even be more closely packed to form an inner cavity. Thus, having three primary particles in a soot aggregate is not a sufficient condition for PCF. The soot aggregate needs some degree of compaction to generate closed loops along a chain of soot spherules. This likely explains the role that cloud processing plays in increasing the ice nucleation activity of soot as observed by e.g. Mahrt et al. (2020b), who found the ice nucleation ability of processed, compacted soot particles to be significantly higher compared to unprocessed, more fractal soot. This is further supported by recent results of Zhang et al. (2020), who investigated different laboratory analogues of soot and found the largest ice nucleation activity to be associated with the most spherical, i.e. compacted, soot type. Since evaporating water or ice exerts a tension on the surrounding soot spherules (e.g. China et al., 2015; Ma et al.,

2013), these spherules are attracted to each other and open loops can close, giving rise to ring pores and larger cavities.

We focused our analysis on three-membered and four-membered ring pores as these should be the pores most relevant for soot PCF on soot particles with $10 \leq D_{pp} \leq 30$ nm. The results presented in Sect. 8 corroborate that this is a valid assumption and provide guidance for assessing the ice nucleation activity of atmospheric soot particles with similar properties. For PCF on soot aggregates characterized by smaller primary particles, five-membered or six-membered ring pores or even larger and more complex pore structures and cavities may become relevant. Indeed, Möhler et al. (2005a) showed that GSG soot with particle diameters of 70–140 nm and primary particle diameters as small as 4–8 nm proved to be efficient INPs around 210–220 K with freezing onsets as low as 110 % RH$_i$ (see Fig. 1 and Table F1).

In general, coatings of soot aggregates with semi-volatile material are expected to inhibit PCF. Yet, thin coatings with hydrophilic material should only have a minor effect. In general, they (i) will shift condensation of pore water to lower RH, (ii) will lead to a freezing point depression, and (iii) may facilitate ice growth if diluted aqueous coatings wet the soot surface. Thus, the inhibiting effect of thin coatings should be most pronounced in the temperature range around 230 K where homogeneous ice nucleation rates are critical. Hydrophobic coatings, on the contrary, in general increase the contact angle between the coated soot surface and water (and ice) or completely fill the pores and make them unavailable for PCF, as has been demonstrated recently for soot coated with different types of secondary organic aerosol material (Zhang et al., 2020). Similarly, ice nucleation by soot generated from propane flames was found to shift to higher ice supersaturation and eventually vanish with increasing organic carbon content (Crawford et al., 2011; Mahrt et al., 2018a; Möhler et al., 2005b). More hydrophilic coatings such as strongly oxidized secondary organic aerosol material (Zhang et al., 2020) and sulfuric acid (Möhler et al., 2005b, a) had a less prohibiting effect on the ice nucleation activity but nevertheless increased freezing onsets to higher relative humidity. At the same time, sulfuric acid ageing can oxidize and/or remove organic material residing within the soot pores and thus enhance their susceptibility to taking up water by capillary condensation and soot PCF (Mahrt et al., 2020a). When sulfuric acid condensation leads to thick coatings, the freezing mode will shift to immersion or homogeneous freezing of solution droplets, depending on the ice activity of the specific soot in immersion mode.

Whether a coating inhibits PCF depends on the amount of condensed material and on its distribution on the soot aggregate. Semi-volatile species may preferentially condense within the pores and block them for water condensation. Hydrophilic coatings do not spread on soot particles as an even coating but will form patches or droplets on soot aggregates, depending on the contact angle. A reasonable value to discriminate thin from thick coatings might be 20 % of the soot mass as chosen by Ullrich et al. (2017).

While the soot-PCF framework points to the relevance of the mixing state and coatings for the ice nucleation ability of soot, our present knowledge about these effects on soot PCF is insufficient to draw final conclusions. The delicate dependency of soot PCF on soot (pore) properties may indeed be the reason for the large uncertainties associated with the ice nucleation ability of soot and explain seemingly contradictory results between laboratory studies, when ice nucleation experiments with soot from various sources are compared. Additional studies are crucial to better constrain the effect of coatings depending on their thickness and hydrophobicity.

With its physical basis, the proposed soot-PCF framework has the potential to predict ice nucleation by soot based on physicochemical particle properties. For soot particles that are emitted from a variety of combustion sources, these properties indeed strongly depend on the fuel type and the combustion conditions (e.g. Atiku et al., 2016). Highly refined fuels used in road traffic (gasoline, diesel) and aircraft jet engines (kerosene) mostly contain hydrogen and carbon. When freshly emitted particles from these sources can be regarded as uncoated soot agglomerates, soot PCF can be applied without restriction. However, soot particles from road traffic are emitted at ground level and need to be lifted to cirrus altitudes, throughout which condensation of semi-volatile organic material can occur, limiting or even preventing the applicability of soot PCF. In the case of thick coatings, immersion freezing should be the prevailing ice nucleation pathway, which has been shown to be ineffective in the case of soot (Kanji et al., 2020; Schill et al., 2016). Conversely, aviation emissions constitute an in situ CE8 source of high number concentrations of soot particles in the upper troposphere where cirrus conditions prevail. Persistent contrails and the contrail cirrus clouds evolving from them, forming from water activation of aircraft-emitted soot particles and subsequent homogeneous freezing, are responsible for a major part of the radiative forcing from aviation (Kärcher, 2018). While their direct radiative forcing is relatively well represented in global climate models, little is known about the indirect effect, i.e. the potential of aircraft-emitted soot particles to trigger or alter cirrus formation (Lee et al., 2021). After the contrail dissipates in dry ambient conditions, processed and compacted soot aggregates are left behind, with an increased susceptibility to soot PCF. Such soot particles may impact ice crystal number concentration and crystal size of cirrus clouds, in turn affecting cloud coverage, lifetime, and radiative forcing. A key uncertainty in previous estimates of aircraft–soot cirrus interactions is the ice nucleation efficiency of soot particles (Kärcher et al., 2007), as a function of both soot type and its dependence on aggregate size. In combination with a detailed cirrus cloud model, the soot-PCF framework presented here provides a physically based description of the ice nucleation ability and mechanism of soot with improved predictive capabilities and as such opens

up new avenues to address and reduce these uncertainties (Kärcher et al., 2021)TS15.

Because biomass is a chemically complex fuel, it produces particles of higher heterogeneity in terms of particle morphology and chemical composition compared to highly refined fuels. For instance, biomass burning usually emits soot together with large shares of organic combustion products, so immediate internal mixing is likely, resulting in particles that are frequently internally mixed with inorganic species, mostly potassium salts (Li et al., 2003; Liu et al., 2017; Posfai et al., 2003), resulting in complex chemical composition (e.g. Liang et al., 2021). Nonetheless, under flaming conditions biomass burning has been found to emit relatively pure soot particles (Li et al., 2003; Posfai et al., 2003) that may form ice via soot PCF. Given that combustion of biomass globally denotes an important source of carbonaceous particles, which are furthermore frequently found at cirrus level in the upper troposphere (Schill et al., 2020b), more studies are required to establish the share of sufficiently pure soot particles for soot PCF to be effective.

The way current parameterizations in global climate models predict the ice nucleation activity of soot particles are hampered by a number of factors. Most importantly, they often do not include a size dependence but simplistically assume a fixed percentage of ice-active soot particles ranging from 0.1 % to 100 % with ice activation occurring at one distinct $RH_i$ level (Gettelman et al., 2012; Gettelman and Chen, 2013; e.g. Hendricks et al., 2005; Penner et al., 2009; Wang and Penner, 2010; Zhou and Penner, 2014). Detailed cirrus models with spectral aerosol representations are capable of accommodating size- and supersaturation-dependent INP activity (Jensen et al., 2013, 2018; Kärcher, 2020; Kienast-Sjögren et al., 2015), leading to more realistic simulations of indirect aerosol effects on cirrus. While accounting for soot aggregate size is indispensable to correctly estimating the number of primary particles and pores present that ultimately determine the ice nucleation activity, explicitly measuring the size- and relative humidity-dependent ice nucleation activity has only been the focus of a limited number of studies to date. Recently, Lohmann et al. (2020) used a more realistic parameterization derived from the ice nucleation activity of 400 nm miniCAST black soot aggregates, measured at 233 and 218 K, with the AF being a function of $RH_i$ but still independent of aggregate size. Moreover, Zhou and Penner (2014) and Lohmann et al. (2020) assumed that soot aggregates are rendered inactive over time upon acquiring a coating of three monolayers or one monolayer, respectively. Compared to these parameterizations soot PCF denotes a key step forward as it comprehensively reflects the strong size and relative humidity dependence of ice nucleation on soot aggregates and can be adapted to specific soot properties.

## 10 Summary, conclusions, and outlook

In this work, we have developed a new theoretical framework to describe ice nucleation by atmospherically relevant soot particles through PCF at cirrus conditions. Soot PCF is a three-step process, where water is first taken up into the pores of soot aggregates, followed by homogeneous nucleation of the supercooled pore water and subsequent growth of the pore ice into a macroscopic ice crystal, which can ultimately impact cloud microphysics and radiative properties.

While previous work has inferred the susceptibility of soot to PCF from the dependence of freezing efficiency on temperature and relative humidity, the soot-PCF framework presented here goes a step further as it derives the ice nucleation activity of soot from particle properties, namely the aggregate size and compaction and primary particle diameter and overlap coefficient, as well as the contact angle with water and ice (pore wettability). We find that $n$-membered ring pores are the dominant pore structures for soot PCF, as they are common features of soot aggregates and have a suitable geometry for both filling with water and growing ice below water saturation. We focused our analysis on three-membered and four-membered ring pores as they are of the right size for PCF assuming primary particle sizes typical of atmospherically relevant soot. For these pores, we have derived equations that describe the conditions for all three steps of soot PCF, namely pore filling, ice nucleation, and ice growth. We demonstrated that pore filling occurs via spontaneous gathering of pore water in the pore neck, which we referred to as direct filling. For cirrus conditions, we assume that ice nucleation occurs immediately after the condensed water volume is large enough to host a critical ice embryo, which has been shown to be the case when $T \leq 228$ K (David et al., 2019; Mahrt et al., 2018a, 2020b). Ice growth out of the pore sets a second thermodynamic barrier to PCF. While the critical supersaturation for soot PCF in narrow pores is limited by ice growth, the limiting step switches to capillary condensation for wider pores. The width of ring pores increases with increasing primary particle size and decreases with increasing overlap. Overall, the narrower three-membered ring pores tend to be ice growth limited, while the wider four-membered ring pores tend to be limited by capillary condensation of water. With increasing hydrophobicity, both capillary condensation and ice growth shift to higher ice saturation ratios, making the contact angle between soot and water/ice a key parameter for soot PCF.

Based on the characteristics of three-membered and four-membered ring pores, we have developed a framework to parameterize the fraction of ice-active soot particles. A key feature of the new parameterization is its capability to reliably extrapolate activated fractions measured for one mobility diameter to other aggregate sizes and to predict ice activity of a soot sample based on its physicochemical characterization. As such, we believe that it is of great advantage over previously reported parameterizations, as it significantly reduces

the complexity of the parameters and processes that need to be represented and incorporated into climate models, relying on just one fit function to represent the ice nucleation activity of differently sized soot aggregates from the same emission source, i.e. with the same contact angle. The applicability of this framework is supported by the excellent agreement of the derived parameterization and experiments when simulating previously measured laboratory data of ice formation by soot at cirrus temperatures. Our analysis further revealed that pre-activation by ice preserved in soot pores is of limited relevance under cirrus conditions, because, when ice nucleation rates are high, its only effect is to decrease the ice supersaturation required to activate those soot particles whose limiting step for PCF is capillary condensation. However, when soot particles that have been pre-activated below the HNT encounter ice supersaturated conditions above the HNT, the presence of pore ice bypasses the nucleation step enabling ice crystal formation from soot particles with no intrinsic ice nucleation activity in MPC conditions. Ring pores have limited potential to exhibit pre-activation, because the pore ice is not preserved at low relative humidity and high temperatures. In contrast, the inner cavities between four tetrahedrally or eight cubically arranged primary particles, also described herein, are able to preserve ice at higher temperatures and lower relative humidity because the large volume of the inner cavity prevents the ice from melting and the narrow pore opening protects it from evaporation.

Overall, the soot-PCF framework provides strong constraints on the extent to which atmospheric soot particles contribute to ice nucleation in cirrus clouds depending on their physicochemical properties. Consideration of this framework when designing future experiments and incorporation of the new parameterization into future modelling studies may vastly improve attempts to quantify the impact of soot PCF on cirrus microphysical properties and provide guidance to predict the radiative impact of soot via soot–cirrus interactions. Although we only studied a number of idealized pore types formed between equally sized primary particles, the presented framework is able to link previously reported ice nucleation experiments with the pore characteristics of the soot samples. Nonetheless, we acknowledge that there remain a number of challenges for future studies to address. Arguably, uncertainties arise from the soot–ice contact angle and its relation to the soot–water contact angle. Future studies measuring the contact angles on soot surfaces are required but should aim at taking into account surface structure, chemical composition, and contact angle hysteresis. Further uncertainties arise from the critical volume required for ice nucleation depending on temperature and relative humidity and from the abundance of ring pores in soot aggregates in relation to their size and compaction. Moreover, we focused our investigation on three-membered and four-membered ring pores made of equally sized primary particles. Analysis of other pore types including pores between unequally sized primary particles might improve the

assessment of ice activity of soot aggregates depending on their size and compaction. While we have used available literature data to constrain our framework, further experimental exploration is required, in particular on ambient soot particles. Such studies should take great care in characterizing the physical and chemical properties of the soot particles, in particular those discussed herein, which will be essential to further validate the soot-PCF framework. We reiterate that there is a clear need for dedicated ice nucleation studies on size-selected soot particles and also for aggregate sizes with (mobility) diameters below 100 nm, as well as for studies exploring the impact of semi-volatile coatings on the ice nucleation ability of these particles. Lastly, we underscore the importance of reporting full ice activation spectra, rather than only ice nucleation onset conditions, to further improve our understanding of the physical principles determining the ice nucleation by aerosol particles and thereby better predict their effects on clouds and climate.

## Appendix A: Pore geometry

### A1 Pore geometry in tetrahedral packing arrangement

We TS16 start by analysing the pores formed between four primary soot particles out of a tetrahedral packing arrangement as illustrated in Fig. A1. The centres A, B, C, and D of the four primary particles form the corners of an equilateral tetrahedron. This arrangement gives rise to two types of pores, namely the three-membered ring pore enclosed by the spheres making up the triangle (ABC), whose dimensions we denote with the subscript tr, and the concave octahedron, the void in between the tetrahedron enclosed by the spheres A, B, C, and D, whose dimensions we denote by the subscript te. In the following we only consider pores formed between spheres of equal diameter.

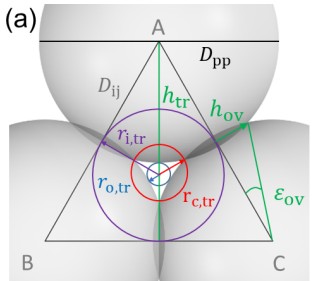 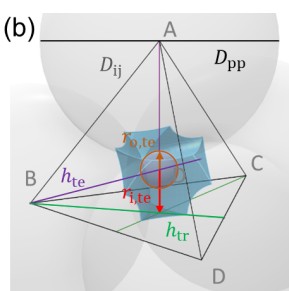

**Figure A1.** Dimensions of the different pore types formed in the case of a tetrahedral packing arrangement, where four overlapping spheres of diameter $D_{pp}$ are arranged on the corners of a regular tetrahedron, indicated by A, B, C, and D. **(a)** Top view of the three-membered ring pore that forms between the three spheres with centres at the corners of the triangle ABC. **(b)** The inner cavity formed within the tetrahedral cell with apices at the centres of the spheres A, B, C, and D has the form of a concave octahedron.

### A1.1 Three-membered ring pore

The three-membered ring pore has the shape of a concave triangle (see Fig. A1). The radii of its circumcircle, $r_{c,tr}$, and its incircle, $r_{i,tr}$, can be determined from the diameter of the primary particles, $D_{pp}$, and the distance, $D_{ij}$, between the centres of two (overlapping) spheres. We start out by calculating the height, $h_{tr}$, of the equilateral triangle ABC:

$$h_{tr} = \frac{\sqrt{3}}{2} D_{ij} \, . \tag{A1}$$

The radius of the circumcircle of the concave triangle, $r_{c,tr}$, can be expressed as a function of the incircle of the triangle ABC ($r_{i,tr}$) and the height of the overlap between the primary particles, $h_{ov}$, as illustrated in Fig. A1:

$$r_{c,tr} = r_{i,tr} - h_{ov} \, . \tag{A2}$$

The radius of the incircle is given by

$$r_{i,tr} = \frac{1}{2\sqrt{3}} D_{ij} \, . \tag{A3}$$

The height of the overlap, $h_{ov}$, can be expressed as a function of the angle $\varepsilon_{ov}$:

$$h_{ov} = \frac{\sin \varepsilon_{ov} D_{pp}}{2} . \tag{A4}$$

Making use of the "trigonometric Pythagoras", $\sin \varepsilon_{ov}$ can be expressed as

$$\sin \varepsilon_{ov} = \sqrt{1 - \left( \frac{D_{ij}}{D_{pp}} \right)^2} . \tag{A5}$$

Inserting Eqs. (A3)–(A5) into Eq. (A2) yields

$$r_{c,tr} = \frac{1}{2\sqrt{3}} D_{ij} - \frac{D_{pp}}{2} \sqrt{1 - \left( \frac{D_{ij}}{D_{pp}} \right)^2} . \tag{A6}$$

The radius of the circumcircle $r_{c,tr}$ coincides with the radius of the incircle in the absence of overlap of the primary particles. The radius of the opening of the concave triangle, $r_{o,tr}$, can be derived from the height of the triangle ABC, the radius of the circumcircle of the concave triangle, and the primary particle diameter as

$$r_{o,tr} = h_{tr} - r_{i,tr} - \frac{D_{pp}}{2} = \frac{\sqrt{3}}{2} D_{ij}$$
$$- \frac{1}{2\sqrt{3}} D_{ij} - \frac{1}{2} D_{pp} = \frac{1}{\sqrt{3}} D_{ij} - \frac{1}{2} D_{pp} \, . \tag{A7}$$

Setting Eq. (A7) to zero yields $D_{ij} = \frac{\sqrt{3}}{2} D_{pp}$, which can be inserted into Eq. (1) to derive the overlap coefficient, for which the opening of the three-membered ring pore vanishes:

$$C_{ov} = \frac{D_{pp} - D_{ij}}{D_{pp}} = 1 - \frac{\sqrt{3}}{2} = 0.134. \tag{A8}$$

### A1.2 Concave octahedron

The pore enclosed by the four primary particles located on the corners of the tetrahedron has the shape of a concave octahedron as depicted in Fig. A1. The height of the regular tetrahedron, $h_{te}$, with apices given by the centres of the spheres A, B, C, and D is given by

$$h_{te} = \sqrt{\frac{2}{3}} D_{ij} \, . \tag{A9}$$

The sphere with its origin at the centre of the tetrahedral cell reaches its maximum size when it comes in contact with the bodies of the primary particles. This maximum radius can be expressed as

$$r_{o,te} = h_{te} - r_{i,te} - \frac{D_{pp}}{2}, \tag{A10}$$

where $r_{i,te}$ is the radius of the insphere that is tangent to the faces of the regular tetrahedron given as

$$r_{i,te} = \frac{D_{ij}}{\sqrt{24}} . \tag{A11}$$

Inserting Eqs. (A9) and (A11) into Eq. (A10) yields

$$r_{o,te} = h_{te} - r_{i,te} - \frac{D_{pp}}{2} = \sqrt{\frac{2}{3}} D_{ij} - \frac{1}{\sqrt{24}} D_{ij}$$

$$- \frac{D_{pp}}{2} = \frac{3 D_{ij}}{\sqrt{24}} - \frac{D_{pp}}{2} . \tag{A12}$$

Again, setting Eq. (A12) to zero yields $D_{ij} = \sqrt{\frac{2}{3}} D_{pp}$, which can be inserted into Eq. (1) to yield the overlap coefficient for which the pore within the concave octahedron disappears:

$$C_{ov} = \frac{D_{pp} - D_{ij}}{D_{pp}} = 1 - \sqrt{\frac{2}{3}} = 0.184 . \tag{A13}$$

An overview of the pore dimensions formed for the case of a tetrahedral packing arrangement is given in Fig. 3a, showing $r_{o,tr}$, $r_{c,tr}$, and $r_{o,te}$ for different sphere (primary particle) sizes as a function of overlap.

## A2    Pore geometry in cubic packing arrangement

The case of a cubic packing arrangement of overlapping spherical particles is depicted in Fig. A2, showing in panel (a) the top view of four overlapping particles, forming a square with corners A, B, C, and D, that denote the centres of neighbouring spheres within the same lattice layer (plane). In this packing arrangement the centres of spheres among different lattice layers are aligned with each other across the different layers. The cubic packing gives rise to two types of pores. Four-membered ring pores arise as the cavity formed between the spheres making up the square ABCD. We use the subscript sq to refer to the dimensions within this pore type. Another type of pore originates as the concave cube, i.e. the void in between eight spheres formed between two sets of four neighbouring spheres in adjacent lattice layers, and we denote the dimensions of this pore by the subscript cu.

### A2.1    Four-membered ring pore

The four-membered ring pore has the shape of a concave square. The radii of its circumcircle, $r_{c,sq}$, and its incircle, $r_{i,sq}$, can be calculated from the diameter of the primary particles, $D_{pp}$, and the distance between the centres of the overlapping spheres, $D_{ij}$, similarly to the case of the three-membered ring pore. The radius of the circumcircle of the concave square can be expressed as a function of the incircle of the square ABCD and the height of the overlap between two adjacent spheres, as shown in Fig. A2:

$$r_{c,sq} = r_{i,sq} - h_{ov} . \tag{A14}$$

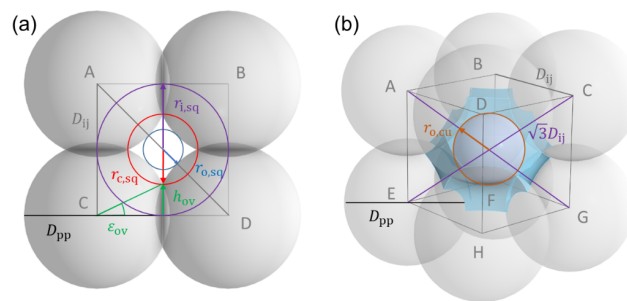

**Figure A2.** Dimensions of the different pore types formed in the case of a cubic packing arrangement of overlapping spheres with diameter $D_{pp}$. **(a)** Top view of the four-membered ring pore that forms between the four spheres with centres at the corners of the square ABCD. **(b)** The inner cavity formed within the cubic cell between eight neighbouring spheres of two lattice layers has the form of a concave cube (blue shading).

Here, the radius of the incircle of the square ABCD, $r_{i,sq}$, is directly given by the half width of the distance along the centres of two overlapping spheres as

$$r_{i,sq} = \frac{1}{2} D_{ij} . \tag{A15}$$

The height of the overlap can be calculated from the angle $\varepsilon_{ov}$, analogously to the case of the three-membered ring pore, and is given by Eq. (A4). Based on this, the radius of the circumcircle can be expressed as

$$r_{c,sq} = \frac{1}{2} D_{ij} - h_{ov} = \frac{1}{2} D_{ij}$$

$$- \frac{\sin \varepsilon_{ov} D_{pp}}{2} = \frac{1}{2} D_{ij} - \frac{D_{pp}}{2} \sqrt{1 - \left( \frac{D_{ij}}{D_{pp}} \right)^2} . \tag{A16}$$

The radius of the circumcircle of the concave square, $r_{c,sq}$, coincides with the radius of the incircle of the square ABCD, $r_{i,sq}$, in the absence of overlap; i.e. $h_{ov} = 0$. The radius of the incircle of the concave square, $r_{o,sq}$, describing the radius of the pore opening, is given by

$$r_{o,sq} = \frac{\sqrt{2}}{2} D_{ij} - \frac{1}{2} D_{pp} . \tag{A17}$$

Here, $\sqrt{2}/2 D_{ij}$ is the half length of the diagonal of the square ABCD (see Fig. A2). Setting Eq. (A17) to zero yields $\sqrt{2} D_{ij} = \frac{1}{\sqrt{2}} D_{pp}$ TS17, which can be inserted into Eq. (1) to obtain the value of the overlap coefficient for which the pore opening of the four-membered ring pore vanishes:

$$C_{ov} = \frac{D_{pp} - D_{ij}}{D_{pp}} = 1 - \frac{1}{\sqrt{2}} = 0.293 . \tag{A18}$$

### A2.2    Concave cube

Considering two adjacent layers of the cubic packing arrangement, a cavity is formed between eight primary particles arranged on the corners of a cube (A–H; see Fig. A2b).

The enclosed pore has the shape of a concave cube. The centre of a sphere that can be placed within this concave cubic cell is located along the space diagonal of the cube spanned by the eight spheres, given by $\sqrt{3}D_{ij}$. Accordingly, the radius of the sphere that is in point contact with the bodies of the primary particles is given by

$$r_{o,cu} = \sqrt{3}D_{ij} - D_{pp} . \qquad (A19)$$

Setting Eq. (A19) to zero yields $D_{ij} = \frac{1}{\sqrt{3}}D_{pp}$, and consequently the overlap for which the pore in the dented concave cube disappears becomes

$$C_{ov} = \frac{D_{pp} - D_{ij}}{D_{pp}} = 1 - \frac{1}{\sqrt{3}} = 0.423. \qquad (A20)$$

An overview of the various pore dimensions formed for the case of a cubic packing arrangement is given in Fig. 3b, showing $r_{o,sq}$, $r_{c,sq}$, and $r_{o,cu}$ for different sphere (primary particle) sizes and as a function of overlap.

## Appendix B: Pore filling with liquid water and ice

### B1   Kelvin equation for conical and cylindrical pores

Water in capillaries exhibits a curved surface, a so-called meniscus, that stabilizes it with respect to bulk water. Similarly, the stabilization of ice in narrow pores can be described by its surface curvature. The radius of the curvature is related to the equilibrium vapour pressure above the curved surface through the Kelvin equation (Kelvin, 1904; Thomson, 1871). The Kelvin equation relies on the Young–Laplace equation (Laplace et al., 1829; Young, 1805), which describes the pressure difference, $\Delta P$, across the interface between the curved surface and the surrounding water vapour. In its general form the Young–Laplace equation can be written as

$$\Delta P = P - P_0 = \gamma(T)C = \gamma(T)\left(\frac{1}{r_{c1}} + \frac{1}{r_{c2}}\right), \qquad (B1)$$

where $\gamma(T)$ is the interfacial tension between the condensed phase (water or ice) and water vapour (in the case of liquid water, $\gamma_{vw}(T)$; in the case of ice, $\gamma_{vi}(T)$), $P$ is the absolute pressure within the condensed phase; $P_0$ is the standard pressure, and $\Delta P$ is the difference in pressure. $C$ denotes the curvature of the interface, which can be expressed by $r_{c1}$ and $r_{c2}$ describing the principal radii of curvature of the condensate. Concave curvatures of the condensed surface (negative radii of curvature) are indicative of a negative absolute pressure within the condensed phase. In contrast, in the case of spherical droplets (or minuscule ice crystals), the convex surface (positive radius of curvature) is linked to positive (increased) absolute pressure within the droplet (ice crystal).

The Kelvin equation in its general form (see e.g. Appendix A2 in Marcolli, 2020, for a detailed derivation) is

given as

$$\frac{p}{p_{w/i}}(T) = \exp\left(\frac{\gamma_{vw/i}(T)\left(\frac{1}{r_{c1}} + \frac{1}{r_{c2}}\right)v_{w/i}(T)}{kT}\right). \qquad (B2)$$

Here, the subscript $w/i$ can refer to either $w$ (liquid water) or $i$ (ice), to indicate the Kelvin equation with respect to liquid water or ice, respectively. Hence, $p/p_{w/i}(T) = S_{w/i}$, with $p$ and $p_{w/i}$ describing the water vapour pressure and the equilibrium water/ice vapour pressure; $k$ is the Boltzmann constant, and $v_{w/i}(T)$ is the molecular volume of liquid water/ice. If the two principal radii of curvature are identical, i.e. $r_{c1} = r_{c2} = r_c$, the term in the nominator describing the curvature of the interface becomes $2/r_c$.

### B1.1   Cylindrical pores

In the case of cylindrical and conical pores, there is only one principal axis of curvature describing the concave water or ice surface of capillary condensates (Fig. B1). The shape of the water/ice surfaces formed in such pore geometries is a classical problem (e.g. de Gennes, 1985). In addition to the pore geometry, capillary condensation also depends on the wettability of the material making up the pore given by the contact angle $\theta_{sw/i}$, describing the angle between the water/ice surface and the pore wall (i.e. substrate ($s$) TS18 surface). For cylindrical pores with a constant pore diameter $D_p$, as shown in Fig. B1a, pore filling occurs at a concave water/ice surface with radius

$$r_c = \frac{-D_p}{2\cos\theta_{sw/i}}, \qquad (B3)$$

and the RH of pore filling can be calculated as

$$\frac{p}{p_{w/i}} = \exp\left(-\frac{\gamma_{vw/i}(T)\left(\frac{4\cos\theta_{sw/i}}{D_p}\right)v_{w/i}(T)}{kT}\right). \qquad (B4)$$

The dependence of pore filling on the pore diameter and the contact angle is illustrated in Fig. B2a and b, for the case of water condensation, showing the saturation ratio with respect to water, $S_w$, calculated from Eq. (B4). For a given pore diameter, cylindrical pores with a lower contact angle fill at lower $RH_w$ compared to pores with a higher contact angle. We reiterate that for $\theta_{sw/i} > 90°$ the pore water or ice exhibits a convex curvature, associated with an equilibrium vapour pressure that is larger than the bulk vapour pressure.

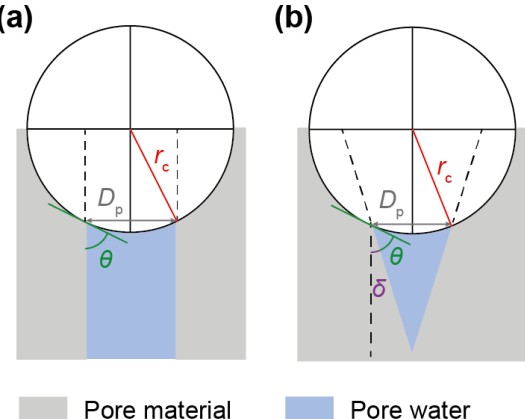

**Figure B1.** Illustration of radius of curvature $r_c$ of **(a)** cylindrical and **(b)** conical pores with pore diameter $D_p$, contact angle $\theta_{sw}$, and pore-opening angle $\delta$. For the cylindrical pore, $\delta$ is $0°$.

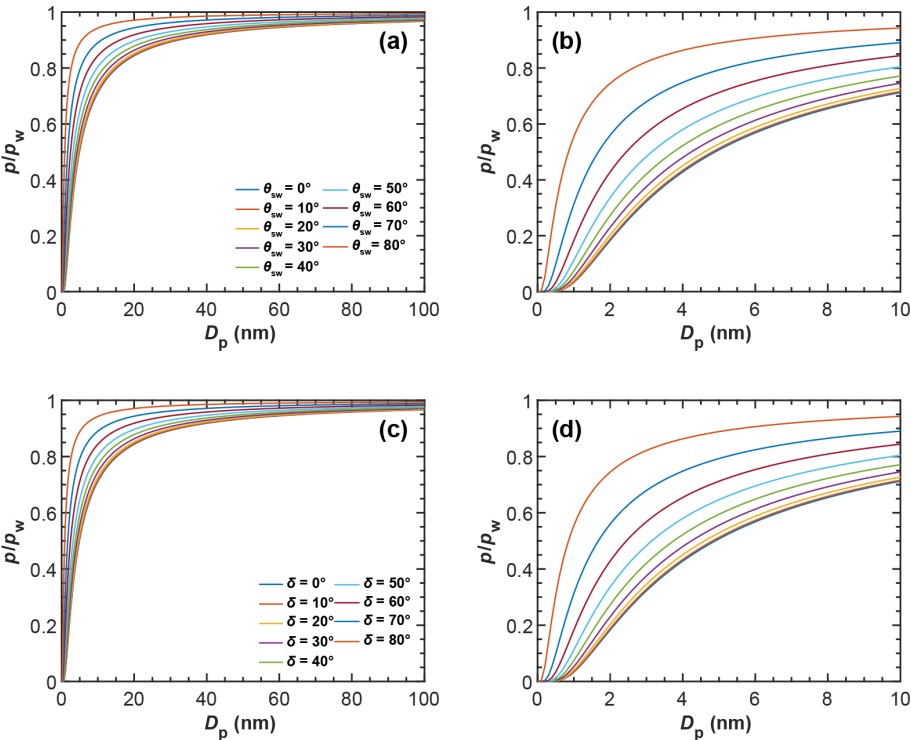

**Figure B2.** Influence of contact angle and pore opening on pore filling for cylindrical and conical pores. **(a, b)** Saturation ratio with respect to water ($p/p_w$) required to fill cylindrical pores with different water–soot contact angles ($\theta_{sw}$), as a function of the pore diameter ($D_p$). **(c, d)** Saturation ratio with respect to water required to fill a conical pore with different opening angles ($\delta$) to the level $D_p$, for a fixed value of $\theta_{sw} = 0°$. Panels **(b)** and **(d)** are enlargements of panels **(a)** and **(c)**, respectively. All lines are calculated for $T = 220\,\mathrm{K}$. The relation between the water–soot and ice–soot contact angle is given in Appendix C.

## B1.2 Conical pores

While cylindrical pores fill completely when the RH of pore filling is reached, conical pores fill gradually. Their filling level depends on the pore diameter at the level of the water/ice surface; the pore-opening angle, $\delta$, which is constant for conical pores (see Fig. B1b); and the contact angle:

$$r_{\mathrm{c}} = \frac{-D_{\mathrm{p}}}{2\cos(\theta_{\mathrm{sw/i}} + \delta)}. \tag{B5}$$

Thus, pore filling takes place up to the pore diameter $D_{\mathrm{p}} = -2r_{\mathrm{c}}\cos(\theta_{\mathrm{sw/i}} + \delta)$. Accordingly, the Kelvin equation for conical pores is given by

$$\frac{p}{p_{\mathrm{w/i}}} = \exp\left(-\frac{\gamma_{\mathrm{vw/i}}(T)\left(4\cos\left(\theta_{\mathrm{sw/i}} + \delta\right)\right)v_{\mathrm{w/i}}(T)}{D_{\mathrm{p}}kT}\right). \tag{B6}$$

Figure B2 illustrates the effect of the pore-opening angle on the water saturation ratio required for pore filling. For $\delta = 0°$, the geometries of cylindrical and conical pores are the same, and consequently the pore-filling conditions are the same. For a given contact angle, increasing the pore-opening angle causes the $\mathrm{RH_w}$ for pore filling to increase, as the radius of the concave meniscus increases. Similarly to the case of the cylindrical pore, the $\mathrm{RH_w}$ required for pore filling decreases with a decreasing contact angle. Inspection of Eq. (B5) reveals that an increase in the pore-opening angle has the same effect on $r_{\mathrm{c}}$ as an increase in the contact angle by the same amount, since the two angles add up within the cosine.

## B2 Slit filling between two adjacent primary particles with liquid water

The slit forming between two neighbouring, overlapping primary particles is shown in Fig. B3. The water meniscus takes the form of a pendular ring with two principle radii of curvature that can be related to the height and the width of the filling level of the slit, $h_{\mathrm{fl}}$ and $d_{\mathrm{fl}}$, respectively (e.g. Gladkikh and Bryant, 2003; Huang et al., 2015; Persiantseva et al., 2004), which are independent of each other. The height of slit filling, $h_{\mathrm{fl}}$, can be expressed as a function of the primary particle diameter $D_{\mathrm{pp}}$ and the slit-filling angle $\varepsilon$ as depicted in Fig. B3a:

$$h_{\mathrm{fl}} = \frac{D_{\mathrm{pp}}}{2}\sin\varepsilon. \tag{B7}$$

The height of the filling level, $h_{\mathrm{fl}}$, corresponds to the radius of the pendular ring of water condensed within the slit between the two particles and consequently yields a convex (positive) curvature:

$$r_{\mathrm{c1}} = \frac{D_{\mathrm{pp}}\sin\varepsilon}{2}. \tag{B8}$$

As the pendular ring is closed in itself, there is no contact angle to be considered.

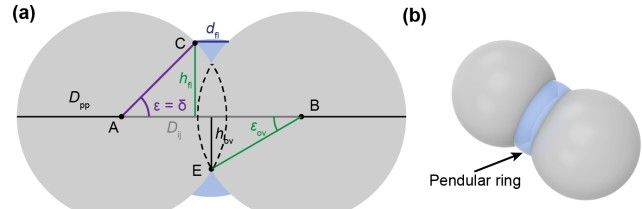

**Figure B3. (a)** Two-dimensional illustration of the slit-shaped pore formed between two overlapping spherical particles of the diameter $D_{\mathrm{pp}}$. The liquid water phase forms a pendular ring of water. The two principal radii of curvature of the pendular water ring are given by $r_{\mathrm{c1}}$ and $r_{\mathrm{c2}}$, which are related to $h_{\mathrm{fl}}$ and $d_{\mathrm{fl}}$, respectively (see text for details). **(b)** The three-dimensional shape of the pendular ring can be obtained by revolving the slit-shaped water phase (blue) about the axis described by the line between points A and B. Represented in panel **(b)** is a ring corresponding with a contact angle of $\theta_{\mathrm{sw}} = 60°$.

The expression for the second principal radius of curvature can be derived from evaluating the width of the pendular ring where it contacts the surface of the spherical particles given as $d_{\mathrm{fl}} = D_{\mathrm{ij}} - D_{\mathrm{pp}}\cos\varepsilon$. Since the water surface along this axis spans a "bridge" between the two primary particles, the radius of curvature depends on the contact angle and the pore-opening angle $\delta$, which coincides with the slit-filling angle $\varepsilon$ (i.e. $\delta = \varepsilon$):

$$r_{\mathrm{c2}} = \frac{-D_{\mathrm{ij}} + D_{\mathrm{pp}}\cos\varepsilon}{2\cos(\theta_{\mathrm{sw}} + \varepsilon)}. \tag{B9}$$

Thus, unlike the pore-opening angle of conical pores, the slit-opening angle is not constant but increases with increasing slit filling. As a consequence, the radius of curvature $r_{\mathrm{c2}}$ switches from concave to convex during pore filling when $(\theta_{\mathrm{sw}} + \varepsilon)$ becomes $>90°$. Inserting Eqs. (B8) and (B9) into the general expression of the Kelvin equation given in Eq. (B2) finally yields the Kelvin equation describing the filling of the slit-shaped pore formed between two neighbouring, spherical particles via a pendular water ring:

$$\frac{p}{p_{\mathrm{w}}} = \exp\left(\frac{\gamma_{\mathrm{vw}}(T)\left(\frac{2}{D_{\mathrm{pp}}\sin\varepsilon} + \frac{2\cos(\theta_{\mathrm{sw}} + \varepsilon)}{-D_{\mathrm{ij}} + D_{\mathrm{pp}}\cos\varepsilon}\right)v_{\mathrm{w}}(T)}{kT}\right). \tag{B10}$$

## B3 Pore filling via pendular rings for three-membered and four-membered ring pores

Ring pores form between circles of primary particles that are in contact with each other. Figure 2 in the main text depicts the pore opening of three-membered and four-membered ring pores. At each of the contacts between the primary particles, pendular rings of water form as RH increases. As discussed above, each pendular ring is described by the two principal radii of curvature given in Eqs. (B8) and (B9), which are related to the height and the width of the filling level, respectively. As the RH increases, the pendular water ring increases in volume, continuously rising from the

narrowest part of the pore slit (see Fig. B3) according to an increase in the slit-filling angle, $\varepsilon$. In the case of three-membered ring pores, coalescence of the three pendular rings and complete filling of the ring pore is assumed to take place
when $\varepsilon = 30°$ is reached, while water filling up to $\varepsilon = 45°$ is assumed for four-membered ring pores to fill (see also Fig. 4).

Figure B4 shows the filling levels of the slits as a function of the slit-filling angle, $\varepsilon$, for an increasing saturation ratio
with respect to water ($p/p_w$) for primary particle diameters of $D_{pp} = 10$, $D_{pp} = 20$, and $D_{pp} = 30$ nm and different soot–water contact angles, $\theta_{sw}$. The lines in Fig. B4 are calculated using Eq. (B10) assuming overlaps of $C_{ov} = 0.01$, $C_{ov} = 0.05$, and $C_{ov} = 0.1$. We stress that water uptake only starts
for $\varepsilon > \varepsilon_{ov}$. Comparison of the different panels in Fig. B4 reveals that the curves of the water uptake become steeper with increasing overlap. Figure B4 clearly shows that in order to reach the conditions for three-membered ring pore filling ($\varepsilon = 30°$) and four-membered ring pore filling ($\varepsilon = 45°$)
below water saturation, relatively low soot–water contact angles are required. Specifically, soot–water contact angles of $\theta_{sw} < 50°$ for three-membered ring pores and $\theta_{sw} < 30°$ for four-membered ring pores are needed for pore filling via pendular water rings. This makes the filling of ring pores via
pendular rings an ineffective process for porous aerosol particles with (rather) high contact angles such as ambient soot aggregates.

## B4 Direct filling of three-membered ring pore (tetrahedral arrangement)

The side view in Fig. B5b shows that the three-membered ring pore is constrained towards all sides up to the end of the overlap given by the height $h_{ov}$. It widens with increasing distance from the midplane as the radii of the spherical layers (i.e. $D_{fl}$) decrease; hence the size of the concave triangle in-
creases. Capillary condensation by direct filling is expected to start at the narrowest part, which we refer to as the pore neck. First pore filling originates from randomly fluctuating slit water and water patches that coalesce spontaneously when the RH for pore filling is reached. As the ambient RH
further increases, the filling level ($h_{fl}$) gradually rises. Once the water has frozen, ice growth occurs along the same co-ordinate, so the filling of the three-membered ring pore with either liquid water or ice can be described in the same way. We track the filling level as a function of $\varepsilon$ as depicted in
Fig. B5b. As the pore has a 3-fold symmetry, there is only one independent radius of curvature describing the interface between the condensed phase (either liquid or ice) and the vapour phase. We assume that the radius $r_{fl,tr}$ and the contact angle between soot and the condensed phase determine the
filling level as a function of RH. From the side view shown in Fig. B5b the radius of the pore opening at height $h_{fl}$ above

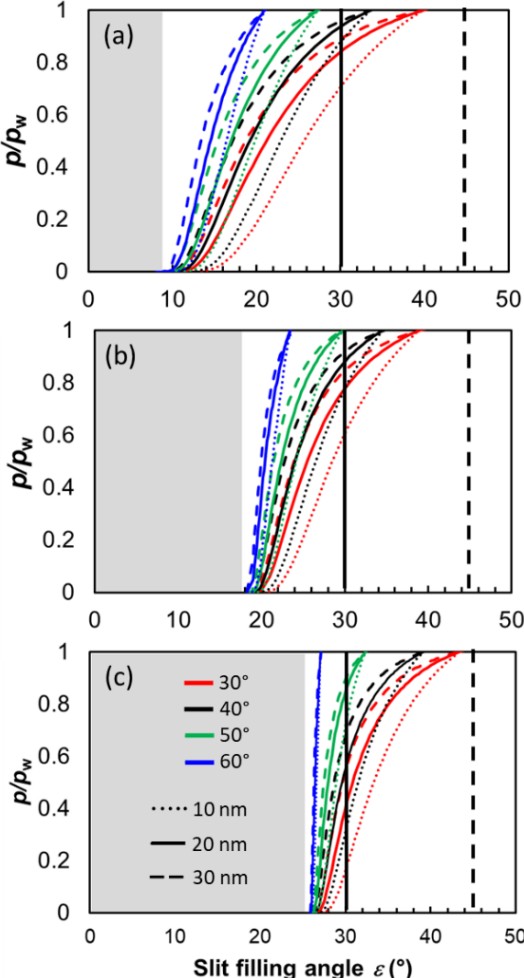

**Figure B4.** Saturation ratio with respect to water ($S_w = p/p_w$) required for filling of three-membered and four-membered ring pores as a function of slit-filling angle $\varepsilon$ at $T = 220$ K for **(a)** $C_{ov} = 0.01$, **(b)** $C_{ov} = 0.05$, and **(c)** $C_{ov} = 0.1$. Primary particle diameters $D_{pp} = 10$–30 nm and soot–water contact angles $\theta_{sw} = 30$–60° are shown as indicated in the legend. For a slit filling of $\varepsilon = 30°$ (solid black vertical line) and $\varepsilon = 45°$ (dashed black vertical line), the pendular water rings coalesce resulting in complete filling of the pore opening for the three-membered and four-membered ring pores, respectively. The grey-shaded area indicates the range of overlap between primary particles which extends to $\varepsilon_{ov} = 8.11°$ for $C_{ov} = 0.01$, $\varepsilon_{ov} = 18.19°$ for $C_{ov} = 0.05$, and $\varepsilon_{ov} = 25.84°$ for $C_{ov} = 0.1$.

the midplane can be written as

$$r_{fl,tr} = h_{tr} - r_{i,tr} - \frac{D_{fl}}{2}. \qquad (B11)$$

$D_{fl}$ can be expressed as a function of $\varepsilon$:

$$D_{fl} = D_{pp} \cos \varepsilon. \qquad (B12)$$

Note that the slit-filling angle $\varepsilon$ and the filling level $h_{fl}$, as given in Eq. (B7), are related via $\varepsilon = \arcsin 2 h_{fl}/D_{pp}$. The

expressions for $h_{tr}$ and $r_{i,tr}$ are given in Eqs. (A1) and (A3), respectively, such that

$$r_{fl,tr} = \frac{\sqrt{3}D_{ij}}{2} - \frac{D_{ij}}{2\sqrt{3}} - \frac{D_{pp}\cos\varepsilon}{2}$$
$$= \frac{2D_{ij} - \sqrt{3}D_{pp}\cos\varepsilon}{2\sqrt{3}}. \tag{B13}$$

To obtain the radius of curvature of the water/ice surface of three-membered ring pores, the contact angle and the pore-opening angle need to be considered additionally, leading to the principal radius of curvature, $r_{c,fl,tr}$:

$$r_{c,fl,tr} = \frac{-2D_{ij} + \sqrt{3}D_{pp}\cos\varepsilon}{2\sqrt{3}\cos(\theta_{sw/i}+\varepsilon)}, \tag{B14}$$

Finally, inserting Eq. (B14) into Eq. (B2), the general form of the Kelvin equation describing the equilibrium vapour pressure over the surface of the filled three-membered ring pore yields

$$\frac{p}{p_{w/i}} = \exp\left(\frac{\gamma_{vw/i}(T)\left(\frac{4\sqrt{3}\cos(\theta_{sw/i}+\varepsilon)}{-2D_{ij}+\sqrt{3}D_{pp}\cos\varepsilon}\right)v_{w/i}(T)}{kT}\right). \tag{B15}$$

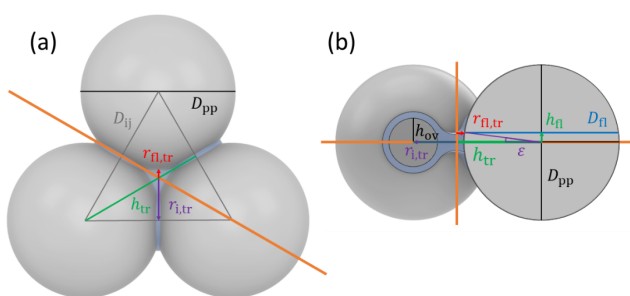

**Figure B5.** Illustration of filling of a three-membered ring pore in **(a)** top view and **(b)** side view. The orange line in panel **(a)** indicates the cross section shown in panel **(b)**. The pore filling is illustrated for a liquid water filling level to $h_{fr} = 1.25$ nm, but the same geometry also applies for filling the pore with ice.

## B5 Direct filling of four-membered ring pore (cubic arrangement)

In analogy to Appendix B4, in this section we derive expressions for the conditions of a four-membered ring pore to fill with liquid water or ice. Similarly to the three-membered ring pore, the filling level will gradually rise with increasing RH. Again, the ambient RH and the contact angle of soot and the condensed phase (water or ice) determine the filling level and hence the radius of curvature of the interface. The four-membered ring pore shown in Fig. B6a has a 4-fold symmetry. Since non-orthogonal radii of curvature equalize,

a surface of constant curvature will form. This surface can be described by one principal radius of curvature analogously to the case of the three-membered ring pore. Considering the side view shown in Fig. B6b, an expression for the radius of the pore opening at height $h_{fl}$ above the midplane is given by

$$r_{fl,sq} = \frac{\sqrt{2}}{2}D_{ij} - \frac{D_{fl}}{2}, \tag{B16}$$

with $\sqrt{2}D_{ij}$ describing the diagonal of the square between the centres ABCD and $D_{fl}$ denoting the diameter of a cross section of a spherical particle at the filling height $h_{fl}$ above the midplane. Using Eq. (B12) $r_{fl,sq}$ is given by

$$r_{fl,sq} = \frac{\sqrt{2}}{2}D_{ij} - \frac{D_{fl}}{2} = \frac{\sqrt{2}D_{ij}}{2} - \frac{D_{pp}\cos\varepsilon}{2}. \tag{B17}$$

With Eq. (B3) the principal radius of curvature is given by

$$r_{c,fl,sq} = \frac{-\sqrt{2}D_{ij} + D_{pp}\cos\varepsilon}{2\cos(\theta_{sw/i}+\varepsilon)}. \tag{B18}$$

Finally, inserting the expression for $r_{c,fl,sq}$ into Eq. (B2) we obtain the general form of the Kelvin equation that can be used to calculate the RH of filling of the four-membered ring pore given by

$$\frac{p}{p_{w/i}} =$$
$$\exp\left(\frac{\gamma_{vw/i}(T)\left(\frac{4\cos(\theta_{sw/i}+\varepsilon)}{-\sqrt{2}D_{ij}+D_{pp}\cos\varepsilon}\right)v_{w/i}(T)}{kT}\right). \tag{B19}$$

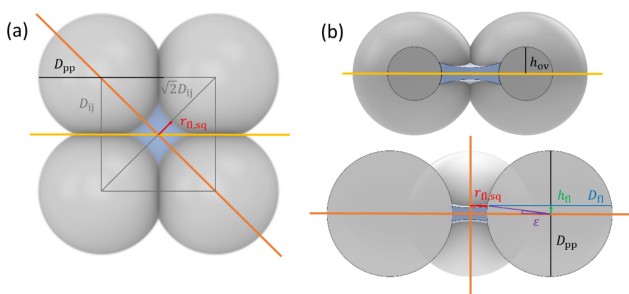

**Figure B6.** Illustration of filling of a four-membered ring pore in **(a)** top view and **(b)** side views. Bright and dark orange lines in panel **(a)** indicate the cross sections shown in the side views depicted in panel **(b)**. The pore filling is illustrated for a liquid water filling level to $h_{fr} = 1.25$ nm, but the same geometry also applies for filling the pore with ice.

## Appendix C: Water soot and ice soot contact angles

The contact angle between soot and water, $\theta_{sw}$, is illustrated in Fig. C1a and can be written as

$$\cos\theta_{sw}(T) = \frac{\gamma_{vs}(T) - \gamma_{sw}(T)}{\gamma_{vw}(T)}, \tag{C1}$$

where $\gamma_{vs}(T)$, $\gamma_{sw}(T)$, and $\gamma_{vw}(T)$ denote the interfacial tensions between water vapour and soot, soot and water, and water vapour and water, respectively. The contact angle between ice and water, $\theta_{iw}$ (Fig. C1a), can be written as

$$\cos\theta_{iw}(T) = \frac{\gamma_{vw}(T) - \gamma_{iw}(T)}{\gamma_{vi}(T)}, \tag{C2}$$

where $\gamma_{iw}(T)$ and $\gamma_{vi}(T)$ are the interfacial tensions between ice and water and water vapour and ice, respectively. In a similar manner, the contact angle between soot and ice (Fig. C1a) can be expressed as

$$\cos\theta_{si}(T) = \frac{\gamma_{vs}(T) - \gamma_{si}(T)}{\gamma_{vi}(T)}, \tag{C3}$$

where $\gamma_{si}(T)$ and $\gamma_{vi}(T)$ are the interfacial tensions between soot and ice and water vapour and ice, respectively. While the surface tension of water can be readily measured (e.g. Wei et al., 2017), the surface tension of ice is not well known. Here, we follow the approach of David et al. (2019) and assume a quasi-liquid layer forms on the ice surface (Nenow and Trayanov, 1986). In this case, an upper limit for the interfacial tension between water vapour and ice (David et al., 2019) can be approximated by

$$\gamma_{vi}(T) = \gamma_{vw}(T) + \gamma_{iw}(T). \tag{C4}$$

Similarly, we assume an upper limit for the surface tension of the soot–ice interface to be given by

$$\gamma_{si}(T) = \gamma_{sw}(T) + \gamma_{iw}(T). \tag{C5}$$

Making use of Eqs. (C4) and (C5) we can express the ice–soot contact angle in Eq. (C3) as a function of the soot–water contact angle:

$$
\begin{aligned}
\cos\theta_{si}(T) &= \frac{\gamma_{vs}(T) - \gamma_{sw}(T) - \gamma_{iw}(T)}{\gamma_{vi}(T)} \\
&= \frac{\gamma_{vw}(T)\cos\theta_{sw}(T) - \gamma_{iw}(T)}{\gamma_{vi}(T)}.
\end{aligned} \tag{C6}
$$

In Fig. C1b the expression in Eq. (C6) is evaluated, for a range of cirrus-relevant temperatures. Here, $\gamma_{vw}(T)$ was calculated using the parameterization given in Hrubý et al. (2014), which is based on the work of Vargaftik et al. (1983), and $\gamma_{iw}(T)$ was calculated using the parameterization given in Ickes et al. (2015).

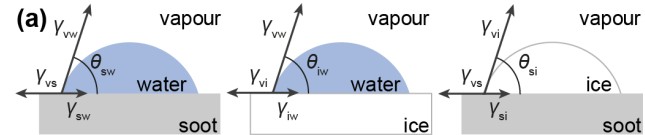

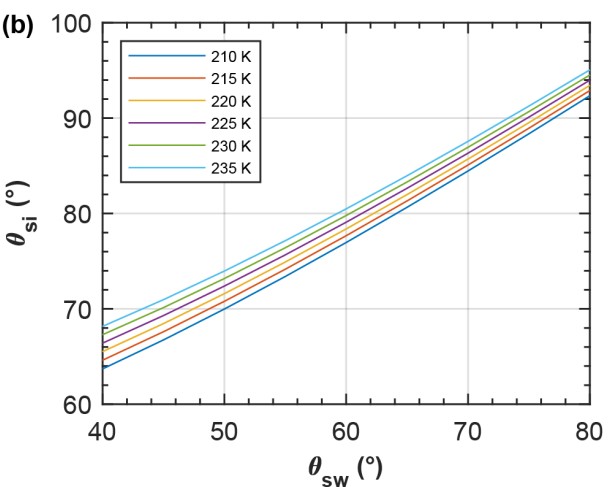

**Figure C1. (a)** Illustration of the contact angles of water on soot ($\theta_{sw}$), water on ice ($\theta_{iw}$), and ice on soot ($\theta_{si}$). **(b)** Soot–ice contact angle as a function of the soot–water contact angle for different temperatures, as indicated in the legend, calculated using Eq. (C6).

## Appendix D: Ice growth out of ring pores

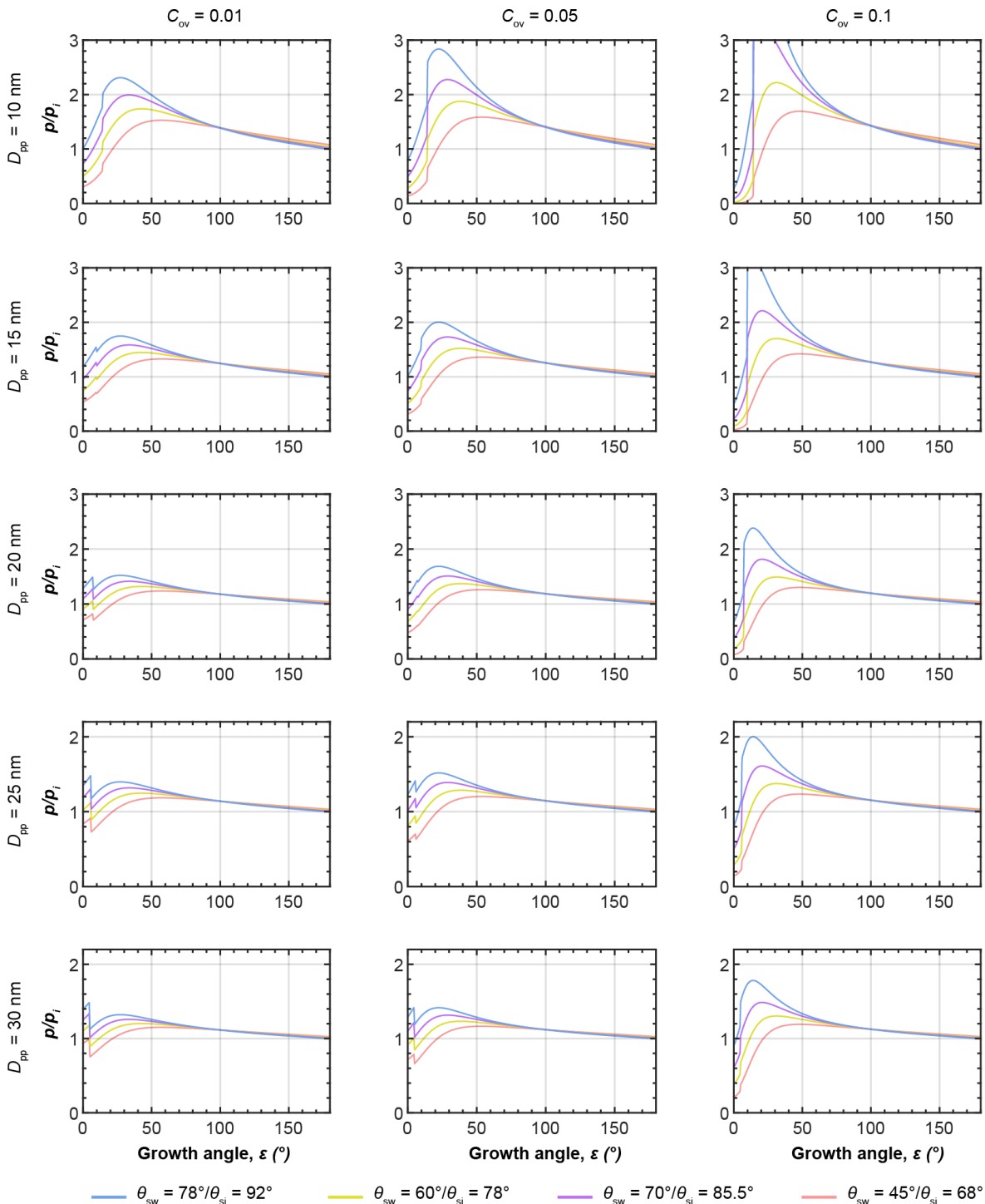

**Figure D1.** Saturation ratio with respect to ice ($S_i = p/p_i$) required for (1) filling of three-membered ring pores, followed by (2) growth of pore ice up to the top of the sphere and beyond the height of the three-membered ring pore structure, as detailed in Fig. 6 and described in the main text. The individual panels correspond to different values of the overlap coefficient ($C_{ov}$, columns) and primary particle diameters ($D_{pp}$, rows). Within each panel the curves correspond to different values of the contact angle between soot and water ($\theta_{sw}$) and soot and ice ($\theta_{si}$), as indicated in the legend. All calculations were performed for $T = 220$ K.

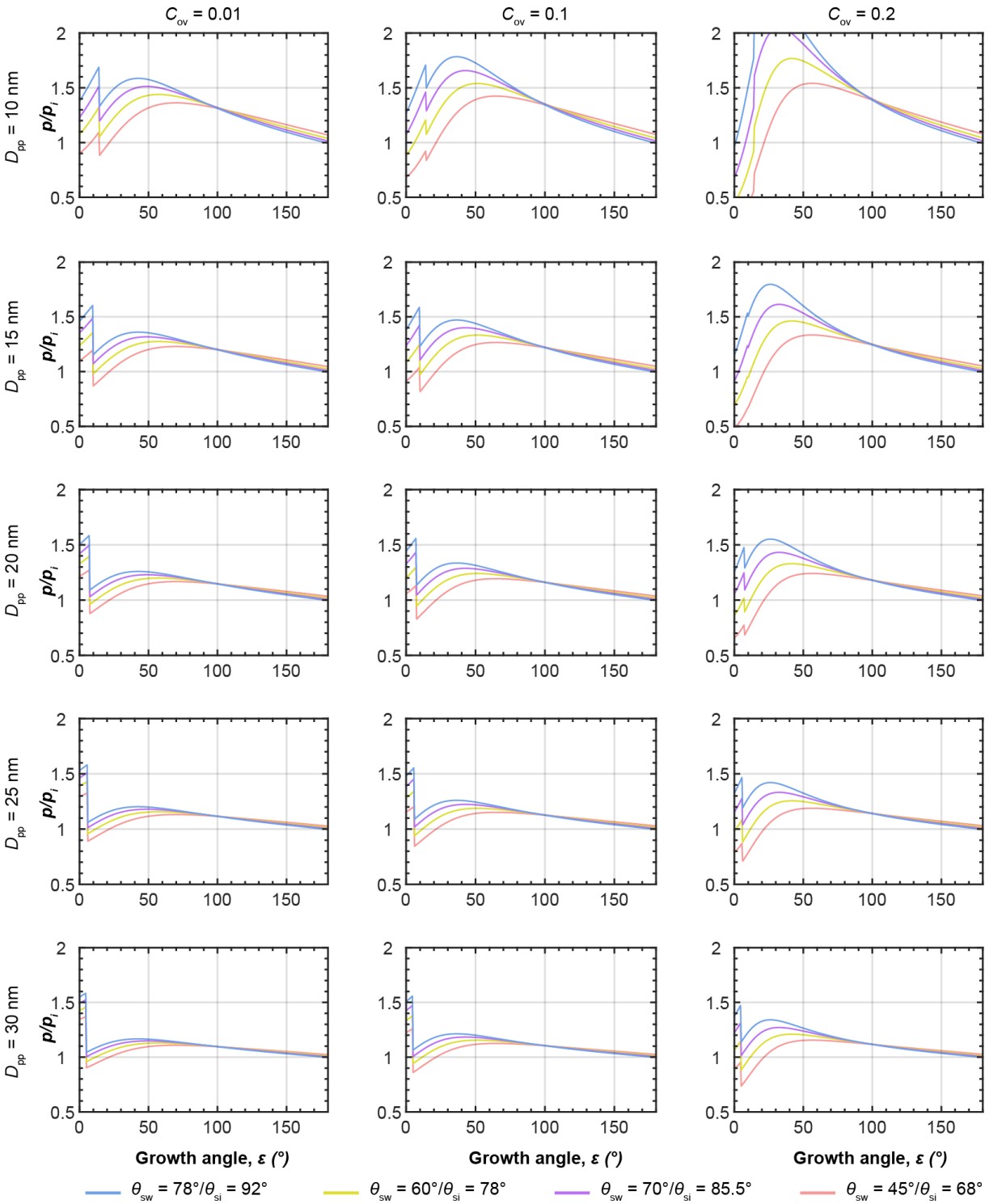

**Figure D2.** Saturation ratio with respect to ice ($S_i = p/p_i$) required for (1) filling of four-membered ring pores, followed by (2) growth of pore ice up to the top of the sphere and beyond the height of the four-membered ring pore structure, as detailed in Fig. 6 and described in the main text. The individual panels correspond to different values of the overlap coefficient ($C_{ov}$, columns) and primary particle diameters ($D_{pp}$, rows). Within each panel the curves correspond to different values of the contact angles between soot and water ($\theta_{sw}$) and soot and ice ($\theta_{si}$), as indicated in the legend. All calculations were performed for $T = 220$ K.

## Appendix E: Soot-PCF parameterization for ice activated fraction of soot aggregates

To predict soot PCF, we have developed an equation that quantifies the ice activated fraction as a function of RH based on the combined probability, $P_N(\text{RH})$, that $n$-membered ring pores nucleate ice at a given RH on the one hand and that a soot aggregate contains such a ring pore on the other hand. By splitting up the ice nucleation probability into two parts, we define a probability function $P_N(\text{RH})$ that depends on the properties of a single ring pore, namely the primary particle size, overlap, and contact angle, and apply it then to aggregates of different size, compaction, and number of (equally sized) primary particles. In other words, $P_N(\text{RH})$ defines the probability per (single) primary particle of it being part of a ring pore structure.

To calculate the probability that an aggregate contains a ring pore, we start by calculating the complementary probability, namely that an aggregate made of $N_p$ primary particles does not contain any primary particle belonging to a ring pore, i.e. $(1 - P_N(\text{RH}))^{N_p}$. However, it is important to consider that at least three (four) primary particles are needed to form a three-membered (four-membered) ring pore. Therefore, this probability needs to be decreased such that three and four primary particles just have one option to form a three-membered and four-membered ring pore, respectively, leading to $(1 - P_N(\text{RH}))^{N_p - n_m}$, where $n_m$ is 2 in the case of three-membered ring pores and $n_m$ is 3 in the case of four-membered ring pores. If both three-membered and four-membered ring pores are present on an aggregate, the use of $n_m = 2$ is advised, to avoid underrepresentation of ring pores in small aggregates.

This probability would be valid for ring pores occurring on a string or chain-like structure (1D) of primary particles. If instead primary particles cover an area (2D) or fill a volume (3D), a primary particle has more neighbouring primary particles with which it can form a ring pore. We account for these multiple options by potentiating the probability by the fractal dimension $D_f$, ranging from values of 1 for chain-like structures to 3 for perfectly spherical soot aggregates. Taken altogether, the activated fraction is then given as

$$\text{AF}(\text{RH}) = 1 - (1 - P_N(\text{RH}))^{\left((N_p - n_m)^{D_f}\right)}. \tag{E1}$$

Note that Eq. (E1) is identical to Eq. (9) of the main text. The fractal dimension determines the spacing between the AF curves of soot aggregates with different mobility diameters. Since experimental derivation of the true three-dimensional fractal dimension is challenging and associated with uncertainties or just derived from two-dimensional TEM images (see Sect. 2.3), $D_f$ can also be used as a fit parameter to adjust modelled AF curves to measurements for aggregates of different (mobility) sizes.

For soot PCF, as described in Eq. (E1), macroscopic ice formation is determined by the processes of pore filling or ice growth out of the pore, both of which occur deterministically at a critical RH. Only at temperatures of around 230 K, where the homogeneous ice nucleation rate may be critical, can the stochastic nature of ice nucleation be relevant. Therefore, vertical velocity, which is a determining factor for the number of homogeneously nucleated ice crystals (Hoyle et al., 2005; Kärcher and Lohmann, 2002; Sullivan et al., 2016), does not influence the number of ice crystals nucleated through soot PCF and can be neglected in the soot-PCF framework. It should further be noted that the function $P_N(\text{RH})$ can either be used to bring AF(RH) into agreement with an experimental dataset or be derived from soot properties by inspecting the different onset RH required to nucleate and grow ice out of ring pores, as shown in Appendix D. Note that within aggregates, there may be structures that are not fully closed to form ring pores. Such structures, which can be viewed as ring pores with negative overlap, may close to form effective ring pores during humidity cycles and cloud processing driven by capillary forces arising during condensation and evaporation (e.g. Huang et al., 1994; Ma et al., 2013). Such an increase in the pore number density would come in addition to the one expected by an increase in $D_f$ and require an adaptation of $P_N(\text{RH})$.

To calculate the number of primary particles, $N_p$, in soot aggregates, an input parameter required for Eq. (E1), we follow Sorensen (2011):

$$N_p = k_0 \left(\frac{2R_g}{D_{pp}}\right)^{D_f}. \tag{E2}$$

Here $k_0$ is the scaling pre-factor, $D_{pp}$ is the median primary particle size of an aggregate, and $D_f$ is the fractal dimension. Previous work has shown that soot aggregates formed through diffusion-limited cluster aggregation (Meakin, 1987; e.g. Witten and Sander, 1983) from different combustion sources exhibit $D_f \approx 1.78 \pm 0.1$ and $k_0 = 1.3 \pm 0.2$ (Sorensen, 2011, and references therein). $R_g$ denotes the radius of gyration of the soot aggregate. The radius of gyration can be related to the more readily measured mobility diameter, $D_m$ (Sorensen, 2011), by

$$\beta = \frac{D_m}{2R_g}, \tag{E3}$$

which in the single particle limit (Sorensen, 2011) is given by

$$\beta = \sqrt{\frac{5}{3}} = 1.29. \tag{E4}$$

The power law given in Eq. (E1) is still valid when the primary particles are sintered, i.e. overlap to some degree as demonstrated by Oh and Sorensen (1997). Furthermore, the factors $D_f \approx 1.78 \pm 0.1$ and $k_0 = 1.3 \pm 0.2$ lead to a good description of soot aggregates that formed through diffusion-limited cluster aggregation, when primary particles overlap,

although the values of $D_f$ and $k_0$ increase slightly with an increasing overlap coefficient (Oh and Sorensen, 1997).

For the parameterization of the experimental data by Mahrt et al. (2018, 2020b) shown in Fig. 7, we rely on the soot characterization presented in Mahrt et al. (2018), who determined primary particle sizes of $D_m = 31$ nm and $D_m = 22$ nm, for miniCAST black and FW200, respectively, and fractal dimensions of $D_f = 1.86$ for miniCAST black and $D_f = 2.35$ for FW200. With these values, the number of primary particles as a function of the mobility diameter of the soot aggregates can be determined using Eqs. (E2)–(E4). The thus-obtained values listed in Table E1 show that for the same aggregate size, the number of primary particles is higher for FW200 than for miniCAST black, owing to both the smaller primary particle size and the higher fractal dimension of FW200. The number of primary particles is required as a parameter in Eq. (9) together with the fractal dimension to determine the probability that a soot aggregate contains a ring pore. Assuming that three-membered ring pores contribute to PCF, we use $n_m = 2$. For FW200, we determined the active-site probability as a function of $RH_i$ because the experimental data acquired at 223 and 218 K overlap when plotted as a function of $RH_i$, which suggests that PCF is ice growth limited. We found good agreement with the experimental AF for all aggregate sizes by choosing the following active-site probability function:

$$P_N(RH_i) = 10^{(a - b RH_i)^{-1}}. \qquad (E5)$$

For miniCAST black, the experimental data acquired at 223 and 218 K overlap when plotted as a function of $RH_w$, indicating that PCF is capillary condensation limited. Therefore, we determined the active-site probability function for this soot as a function of $RH_w$:

$$P_N(RH_w) = 10^{(a - b RH_w)^{-1}} \qquad (E6)$$

The parameters used for the parameterizations are detailed in Table E2.

**Table E1.** Number of primary particles in soot aggregates as a function of soot aggregate mobility diameters $D_m$.

| | Number of primary particles, $N_p$ | |
| $D_m$ | FW200 | miniCAST black |
| --- | --- | --- |
| 400 nm | 501 | 94 |
| 300 nm | 255 | 55 |
| 200 nm | 98 | 26 |
| 100 nm | 19 | 7 |
| 80 nm | 11 | 5 |
| 60 nm | 6 | 3 |

**Table E2.** Parameters used in Eq. (9) to calculate activated fraction (AF).

| | $D_{pp}$ | $D_f$ | $k_0$ | $n_m$ | $a$ | $b$ |
| --- | --- | --- | --- | --- | --- | --- |
| FW200 | 22 nm | 2.35 | 1 | 2 | 0.1544 | 0.002208 |
| miniCAST black | 31 nm | 1.86 | 1.3 | 2 | 0.3374 | 0.006091 TS20 |

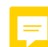

## Appendix F:  Summary of previous studies on ice nucleation with soot for $T <$ HNT

Table F1 summarizes the results of previous studies that investigated the ice nucleation activity of different soot types at $T <$ HNT. Depending on the particle generation method and conditions, the soot samples contained different amounts of organic carbon (OC). Moreover, the table discriminates the soot samples with respect to porosity, hydrophilicity/hydrophobicity, degree of oxidation, and (cloud) processing, whenever this information was provided in the original study. Most studies used size-selected soot aggregates with mobility diameters that exceeded the typical size range of atmospheric soot aggregates, thus overestimating the ice nucleation activity of typical tropospheric soot. Only a limited number of studies specified the primary particle size. GSG soot has been used in some studies such as the one by Möhler et al. (2005a). The primary particle diameter of this soot type is smaller than the typical range observed for atmospheric soot aggregates. Three-membered and four-membered ring pores made of such small primary particles are too small to be efficient structures for soot PCF. Nevertheless, this soot is among the soot types with the lowest ice nucleation onset $RH_i$, suggesting that cavities made of a larger number of primary particles are responsible for PCF. Some soot types consist of very large primary particles making a PCF mechanism unlikely. These soot types indeed showed ice nucleation onsets close to or even above the RH required for homogeneous freezing of solution droplets (Koop et al., 2000). This can for instance be seen in Fig. 1, where the $T$ and $RH_i$ conditions to reach an AF of 1 %, often termed ice onset conditions, are shown. The same figure reveals that the ice nucleation onsets were often observed well above ice saturation, reflecting the hydrophobicity of soot particles. These onset relative humidities show that the contact angle between soot and ice/water is highly relevant to the description of soot PCF.

Most studies shown in Fig. 1 and listed in Table F1 only report ice onset in terms of RH and $T$, without reporting the evolution of the ice activated fraction as a function of these parameters. Thus, the fraction of ice-active particles remains unknown for a large range of RH and $T$ conditions, limiting the use of these studies for an ice nucleation parameterization as presented here. Figure 1 clearly shows that the FW200 dataset that we parameterized in Sect. 8 is among the ones with the lowest onset RH. The contrail-processed mini-CAST black, as reported by Mahrt et al. (2020b) and used in Fig. 7, exhibits onsets clearly below the homogeneous freezing onsets of solution droplets, while most investigated samples show onsets coinciding with homogeneous freezing.

**Table F1.** Summary of studies that have investigated the ice nucleation ability of a variety of uncoated soot particles and different soot types at cirrus temperatures. For each study, the soot type; aggregate electric mobility diameter, $D_m$; average primary particle diameter, $D_{pp}$; and processing conditions are tabulated. The ice nucleation activity is listed in terms of the relative humidity with respect to ice, $RH_i$, at which an activated fraction (AF) of either 0.1–0.3 % or 1 % is reached at a given temperature. In addition, the AF in the conditions of homogeneous freezing of solution droplets is also reported. See primary references for abbreviations of soot types as used in the table.

| Study | Soot type or source | $D_m$ (nm) | $D_{pp}$ (nm) | Processing, properties | Ice formation | | | |
| --- | --- | --- | --- | --- | --- | --- | --- | --- |
| | | | | | $T$ (K) | $RH_i$ | | AF in hom. freezing conditions (%) |
| | | | | | | AF < 0.3 % | AF 1 % | |
| DeMott et al. (1999) | Degussa lamp black | 240 | | | 213 | | 156–161 | <1 |
| | | | | | 218–220 | | 153–156 | <1 |
| | | | | | 223–233 | | 149–152 | <1 |
| Möhler et al. (2005a) | GSG | 70–140 | 4–8 | | 239 | 132–134 | | |
| | | | | | 234 | 128–129 | | |
| | | | | | 208–224 | 110–119 | | |
| | | | | | 186–193 | 126–134 | | |
| Möhler et al. (2005b) | CAST | | 20–60 | 16 % OC | 207 | 145 | | |
| | | | | 40 % OC | 205 | 150–170 | | |
| Koehler et al. (2009) | TS | 200 | 260 | low porosity hydrophobic | 233.15 | 150 | 153 | 0.03 |
| | | | | | 221.65 | 144 | 174 | 0.1 |
| | GTS | 200 | 180 | zero porosity hydrophobic | 233.15 | 158 | 161 | <0.01 |
| | | | | | 221.65 | 170 | 177 | <0.01 |
| | TOS | 200 | 260 | low porosity hydrophilic | 233.15 | 146 | 147 | 1 |
| | | | | | 221.65 | 148 | 154 | 2 |
| | TC1 | 200 | 55 | high porosity hydrophilic | 233.15 | 150 | 155 | <0.01 |
| | | | | | 216.15 | 155 | 177 | 0.1 |
| | AEC | polydisp. 250 | 30–100 | low porosity hygroscopic | 233.15 | – | 145 | 4 |
| | | | | | 216.15 | 152 | 160 | 0.3 |
| Crawford et al. (2011) | CAST | 300 | 30 | OC 5 % | 226–228 | 110 | 122 | 25 |
| | | | | OC 30 % | 223–228 | 107–121 | 163–170 | <1 |
| | | | | OC 70 % | 224–225 | 148–152 | 154–167 | <1 |
| | miniCAST | 250 | $\sim$ 30 | OC 30 % | 224–225 | 132–142 | 136–147 | |
| | | 200 | | OC 80 % | 219 | >220 | – | |
| | | 100 | | OC 90 % | 220 | >160 | | |
| Kanji et al. (2011) | GSG | 20–450 mode: 150 | | OC 10 % | 230 | 140–145 | | |
| | | | | | 235 | 137–147 | | |
| Chou et al. (2013) | diesel (car) | 60–180 | | OC 10 % | 233.14 | 134 | 135 | |
| Kulkarni et al. (2016) | diesel soot | 120 | | unprocessed | 228–233 | | 142–143 | |
| | | | | | 223.15 | | 136 | |
| | | | | 80 % RH | 228–233 | | 139–142 | |
| | | | | | 223.15 | | 131–136 | |

| Study | Soot type or source | $D_m$ (nm) | $D_{pp}$ (nm) | Processing, properties | Ice formation | | | AF in |
|---|---|---|---|---|---|---|---|---|
| | | | | | $T$ (K) | RH$_i$ | | |
| | | | | | | AF < 0.3 % | AF 1 % | hom. freezing conditions (%) |
| Mahrt et al. (2018) | miniCAST black | 100 | 31 | | 218 | 169 | 170 | <0.1 |
| | | | | | 223 | 165 | 167 | <0.1 |
| | | | | | 228 | 160 | 164 | <0.1 |
| | | | | | 233 | 158 | 161 | <0.1 |
| | | 200 | 31 | | 218 | <157 | 157 | <0.1 |
| | | | | | 223 | 153 | 155 | <0.1 |
| | | | | | 228 | 150 | 153 | ∼ 0.1 |
| | | | | | 233 | 148 | 150 | ∼ 0.1 |
| | | 300 | 31 | | 218 | 161 | 166 | <0.1 |
| | | | | | 223 | 158 | 163 | <0.1 |
| | | | | | 228 | 154 | 160 | <0.1 |
| | | | | | 233 | 152 | 157 | <0.1 |
| | | 400 | 31 | | 218 | 155 | 158 | ∼ 0.1 |
| | | | | | 223 | 150 | 153 | ∼ 0.1 |
| | | | | | 228 | 151 | 153 | ∼ 0.1 |
| | | | | | 233 | 153 | >157 | ∼ 0.1 |
| | miniCAST brown | 100 | 21 | | 218 | 165 | 167 | <0.1 |
| | | | | | 223 | 159 | 162 | <0.1 |
| | | | | | 228 | 158 | 159 | <0.1 |
| | | | | | 233 | 154 | 156 | <0.1 |
| | | 200 | 21 | | 218 | ∼ 164 | 169 | ∼ 0.1 |
| | | | | | 223 | 159 | 161 | <0.1 |
| | | | | | 228 | 164 | >165 | <0.1 |
| | | | | | 233 | >158 | >158 | <0.1 |
| | | 300 | 21 | | 218 | 175 | >179 | <0.1 |
| | | | | | 223 | 170 | >171 | <0.1 |
| | | | | | 228 | ∼ 164 | >165 | <0.1 |
| | | | | | 233 | >162 | >162 | <0.1 |
| | | 400 | 21 | | 218 | 168 | 175 | <0.1 |
| | | | | | 223 | 167 | >168 | <0.1 |
| | | | | | 228 | >163 | >163 | <0.1 |
| | | | | | 233 | ∼ 159 | >159 | <0.1 |
| | FW200 | 100 | 22; 13 | | 218 | 157 | 162 | ∼ 0.1 |
| | | | | | 223 | 156 | 160 | ∼ 0.1 |
| | | | | | 228 | 160 | 163 | <0.1 |
| | | | | | 233 | 152 | 155 | <0.1 |
| | | 200 | 22; 13 | | 218 | 135 | 140 | 7 |
| | | | | | 223 | 131 | 136 | 9 |
| | | | | | 228 | 134 | 142 | 2 |
| | | | | | 233 | 145 | 148 | ∼ 0.3 |
| | | 300 | 22; 13 | | 218 | 123 | 126 | 13 |
| | | | | | 223 | 124 | 128 | 14 |
| | | | | | 228 | 122 | 126 | 20 |
| | | | | | 233 | 136 | 141 | 2 |
| | | 400 | 22; 13 | | 218 | 119 | 125 | 18 |
| | | | | | 223 | 121 | 124 | 11 |
| | | | | | 228 | 118 | 119 | 11 |
| | | | | | 233 | 133 | 140 | 2 |

https://doi.org/10.5194/acp-21-1-2021

| Study | Soot type or source | $D_m$ (nm) | $D_{pp}$ (nm) | Processing, properties | Ice formation | | | AF in |
|---|---|---|---|---|---|---|---|---|
| | | | | | $T$(K) | RH$_i$ | | |
| | | | | | | AF < 0.3% | AF 1% | hom. freezing conditions (%) |
| | LB_OEC | 100 | 43 | | 218 | 157 | 160 | 0.2 |
| | | | | | 223 | 155 | 157 | ~ 0.1 |
| | | | | | 228 | 154 | 158 | <0.1 |
| | | | | | 233 | 152 | 153 | <0.1 |
| | | 200 | 43 | | 218 | 155 | 160 | 0.2 |
| | | | | | 223 | 153 | 158 | 0.15 |
| | | | | | 228 | 144 | 147 | 0.15 |
| | | | | | 233 | 148 | 150 | 0.2 |
| | | 300 | 43 | | 218 | 150 | 157 | 0.2 |
| | | | | | 223 | 153 | 154 | 0.5 |
| | | | | | 228 | 150 | 154 | 0.3 |
| | | | | | 233 | 148 | 149 | 0.3 |
| | | 400 | 43 | | 218 | 149 | 154 | 0.5 |
| | | | | | 223 | 146 | 151 | 1 |
| | | | | | 228 | 145 | 148 | 1 |
| | | | | | 233 | 148 | 150 | ~ 0.3 |
| | LB_RC | 100 | 55.7 | | 218 | 158 | 162 | ~ 0.1 |
| | | | | | 223 | 156 | 158 | ~ 0.1 |
| | | | | | 228 | 154 | 156 | ~ 0.1 |
| | | | | | 233 | 151 | 153 | <0.1 |
| | | 200 | 55.7 | | 218 | 149 | 158 | 0.4 |
| | | | | | 223 | 147 | 155 | 0.4 |
| | | | | | 228 | 149 | 153 | 0.2 |
| | | | | | 233 | 150 | 151 | ~ 0.1 |
| | | 300 | 55.7 | | 218 | 131 | 141 | 2.4 |
| | | | | | 223 | 135 | 143 | 2.3 |
| | | | | | 228 | 135 | 146 | 1.5 |
| | | | | | 233 | 147 | 150 | 0.3 |
| | | 400 | 55.7 | | 218 | 143 | 150 | 1.3 |
| | | | | | 223 | 140 | 146 | 1.8 |
| | | | | | 228 | 141 | 147 | 1.3 |
| | | | | | 233 | 146 | 150 | 0.2 |
| | FS | 100 | 12.9 | | 218 | 172 | 174 | <0.1 |
| | | | | | 223 | 166 | 169 | <0.1 |
| | | | | | 228 | 162 | 165 | <0.1 |
| | | | | | 233 | 153 | 156 | <0.1 |
| | | 200 | 12.9 | | 218 | 157 | 162 | ~ 0.1 |
| | | | | | 223 | 152 | 158 | ~ 0.3 |
| | | | | | 228 | 151 | 156 | 0.2 |
| | | | | | 233 | 151 | 152 | 0.2 |
| | | 300 | 12.9 | | 218 | 147 | 154 | 0.9 |
| | | | | | 223 | 142 | 148 | 1.2 |
| | | | | | 228 | 143 | 149 | 1 |
| | | | | | 233 | 150 | 151 | 0.1 |
| | | 400 | 12.9 | | 218 | 147 | 152 | 1 |
| | | | | | 223 | 142 | 148 | 1.6 |
| | | | | | 228 | 143 | 148 | 1 |
| | | | | | 233 | 148 | 150 | 0.2 |

**Table F1.** Continued. TS21

| Study | Soot type or source | $D_m$ (nm) | $D_{pp}$ (nm) | Processing, properties | Ice formation | | | |
|---|---|---|---|---|---|---|---|---|
| | | | | | $T$ (K) | RH$_i$ | | AF in |
| | | | | | | AF < 0.3 % | AF 1 % | hom. freezing conditions (%) |
| Mahrt et al. (2020b) | miniCAST black | 400 | | unprocessed | 218 | 154 | 160 | $\sim 0.1$ |
| | | | | | 223 | 153 | 157 | $\sim 0.1$ |
| | | | | | 228 | 156 | 160 | <0.1 |
| | | | | | 233 | 156 | >157 | <0.1 |
| | | 400 | | contrail processed | 218 | 135 | 139 | 13 |
| | | | | | 223 | 132 | 136 | 17 |
| | | | | $T = 228$ | 228 | 137 | 142 | 4.4 |
| | | | | RH$_w$ = 104 % | 233 | 150 | 151 | <0.1 |
| | | 400 | | cirrus processed | 218 | 138 | 143 | 5.7 |
| | | | | | 223 | 136 | 141 | 6.1 |
| | | | | $T = 228$ | 228 | 136 | 143 | 2.4 |
| | | | | RH$_w$ = 96 % | 233 | 152 | 153 | <0.1 |
| | | 400 | | MPC processed | 218 | 138 | 141 | 10 |
| | | | | | 223 | 135 | 138 | 12 |
| | | | | $T = 243$ | 228 | 137 | 142 | 4 |
| | | | | RH$_w$ = 108 % | 233 | 152 | 153 | <0.1 |
| | | 400 | | pre-cooling MPC | 218 | 147 | 152 | 0.8 |
| | | | | | 223 | 142 | 147 | 2.1 |
| | | | | $T = 243$ | 228 | 148 | 151 | 0.3 |
| | | | | RH$_w$ = 96 % | 233 | 150 | 153 | <0.1 |
| | | 400 | | pre-cooling cirrus | 218 | 150 | 154 | 0.4 |
| | | | | | 223 | 146 | 149 | 1.1 |
| | | | | $T = 228$ | 228 | 149 | 153 | $\sim 0.1$ |
| | | | | RH$_w$ = 65 % | 233 | 153 | 154 | <0.1 |
| | | 100 | | contrail processed | 218 | 164 | 175 | $\sim 0.1$ |
| | | | | | 223 | 156 | 167 | $\sim 0.1$ |
| | | | | $T = 228$ | 228 | 157 | 165 | $\sim 0.1$ |
| | | | | RH$_w$ = 104 % | 233 | 157 | >159 | <0.1 |
| | miniCAST brown | 400 | | unprocessed | 218 | 176 | >177 | <0.1 |
| | | | | | 223 | 174 | >175 | <0.1 |
| | | | | | 228 | 165 | >166 | <0.1 |
| | | | | | 233 | 158 | >160 | <0.1 |
| | | 400 | | contrail processed | 218 | >176 | >176 | <0.1 |
| | | | | | 223 | >172 | >172 | <0.1 |
| | | | | $T = 228$ | 228 | >164 | >164 | <0.1 |
| | | | | RH$_w$ = 104 % | 233 | >158 | >158 | <0.1 |

| Study | Soot type or source | $D_m$ (nm) | $D_{pp}$ (nm) | Processing, properties | $T$ (K) | RHi AF < 0.3 % | AF 1 % | AF in hom. freezing conditions (%) |
|---|---|---|---|---|---|---|---|---|
| Mahrt et al. (2020a) | miniCAST black | 400 | | water aged | 218 | 125 | 130 | 57 |
| | | | | | 233 | 141 | 145 | 13 |
| | | | | H$_2$SO$_4$ aged | 218 | 127 | 132 | 27 |
| | | | | | 233 | 142 | 147 | 1 |
| | miniCAST brown | 400 | | unaged | 218 | 177 | >177 | <0.1 |
| | | | | | 233 | 158 | >160 | <0.1 |
| | | | | water aged | 218 | ∼ 130 | 137 | 5.7 |
| | | | | | 233 | >142 | 143 | ∼ 2 |
| | | | | H$_2$SO$_4$ aged | 218 | 137 | 141 | 2 |
| | | | | | 233 | >140 | 147 | 1 |
| Nichman et al. (2019) | Regal 330R | 800 | 45.4 | non-oxidized | 234 | | 141 | ∼ 1 |
| | | | | | 230–232 | | 137–144 | |
| | | | | | 228–229 | | 130–138 | |
| | Regal 400R | 800 | | oxidized | 232–234 | | 141–145 | ∼ 1 |
| | | | | | 230–232 | | 140–144 | ∼ 1 |
| | | | | | 225–229 | | 144–149 | ∼ 1 |
| | | | | | 219–220 | | 149–150 | ∼ 1 |
| | Monarch 880 | 800 | | non-oxidized | 232–234 | | 140–143 | ∼ 1 |
| | | | | | 230–232 | | 139–142 | ∼ 1 |
| | | 100 | | non-oxidized | 226–227 | | 149–151 | ∼ 1 |
| | Monarch 900 | 800 | | non-oxidized | 232–234 | | 136–144 | |
| | | | | | 230–232 | | 129–134 | |
| | | | | | 228–230 | | 127–136 | |
| | | | | | 226–228 | | 127–128 | |
| | Monarch 900 | 100 | | | 226–227 | | 149–151 | ∼ 1 |
| | R2500U | 800 | 35–42* | | 232–234 | | 133–140 | ∼ 1 |
| | | | | | 230–232 | | 125–130 | |
| | | | | | 228–230 | | 124–132 | |
| | | | | | 225–228 | | 121–131 | |
| | ethylene combustion soot | 800 | intermediate oxidation | | 230–231 | | 142–145 | ∼ 1 |
| | | | | | 229–230 | | 133 | |
| | | | | | 227–228 | | 128 | |
| | | | | | 225–227 | | 122–126 | |
| | | | | | 222–223 | | 122 | |
| | | | | | 219–220 | | 119 | |
| | ethylene combustion soot | 100 | | | 229–234 | | 146–151 | ∼ 1 |

| Study | Soot type or source | $D_m$ (nm) | $D_{pp}$ (nm) | Processing, properties | Ice formation $T$ (K) | RH$_i$ AF < 0.3 % | RH$_i$ AF 1 % | AF in hom. freezing conditions (%) |
|---|---|---|---|---|---|---|---|---|
| Zhang et al. (2020) | R2500U | 200 | 41.9 | | 229–233<br>226–229 | | 145–147<br>148–149 | ∼ 1<br>∼ 1 |
| | | 300 | 35.5 | | 226–234 | | 145–147 | ∼ 1 |
| | | 400 | 34.5 | | 227 | | 142 | 20–25 |
| | COJ300 | 100 | 34.2 | oxidized | 229–231<br>227–229 | | 144–145<br>138–142 | ∼ 1 |
| | | 200 | 34.2 | oxidized | 231–232<br>227–231 | | 138–140<br>130–132 | |
| | | 400 | 34.2 | oxidized | 232<br>230–231<br>229–230<br>226–228 | | 133<br>127–129<br>122<br>116–119 | 100 |
| | R330R | 100 | 45.4 | | 226–232 | | 146–148 | |
| | | 200 | 45.4 | | 228–233 | | 145–146 | |
| | | 400 | 45.4 | | 227–229<br>229–231 | | 137–139<br>143–144 | |
| Ikhenazene et al. (2020) | CAST 1 | Film on substrate, 0–40 μm thick 4 % OC | | | 228.15 | 114 | | |
| | CAST 3 | Film on substrate, 20–40 μm thick 87 % OC | | | 228.15 | 116 | | |

* Taken from Zhang et al. (2020).

## Appendix G: List of symbols and abbreviations

| Symbol | Unit | Description |
|---|---|---|
| AF | – | Activated fraction |
| $C_{ov}$ | – | Primary particle overlap coefficient |
| CFDC | – | Continuous-flow diffusion chamber |
| CNT | – | Classical nucleation theory |
| $d_{fl}$ | m | Width of the water slit in conical pores at the height of filling level ($h_{fl}$) |
| $D_f$ | – | Fractal dimension |
| $D_{fl}$ | m | Diameter of cross section of a spherical particle at the filling height $h_{fl}$ above the midplane |
| $D_{ij}$ | m | Distance between centres of two overlapping spheres |
| $D_m$ | m | Mobility diameter of soot aggregate |
| $D_p$ | m | Diameter of pore |
| $D_{pp}$ | m | Diameter of primary particle of soot |
| $h_{fl}$ | m | Height of filling level |
| $h_{fr}$ | m | Minimum height of capillary condensate in pore opening to allow for freezing to take place |
| $h_{te}$ | m | Height of regular tetrahedron with apices at the centres of four neighbouring spheres |
| $h_{tr}$ | m | Height of equilateral triangle formed between the centres of three spherical particles |
| $h_{ov}$ | m | Height of overlap between two spheres |
| HEFA | – | Hydro-processed esters and fatty acids biofuel |
| HNT | K | Homogeneous nucleation temperature of water |
| INP | – | Ice-nucleating particle |
| $k$ | $J\,K^{-1}$ | Boltzmann constant |
| $k_0$ | – | Scaling pre-factor |
| MPC | – | Mixed-phase cloud |
| $n_{it}$ | – | Exponent used to parameterize the interfacial tension between ice and water in the CNT parameterization proposed by Murray et al. (2010) |
| $n_m$ | – | Indicator for pore type, i.e. ring structure type; $n_m = 3$ (four-ring), $n_m = 2$ (three-ring) |
| $N_p$ | – | Number of (equally sized) primary particles in a soot aggregate |
| OC | – | Organic carbon |
| $p$ | Pa | Water vapour pressure |
| $p_{w/i}$ | Pa | Saturation vapour pressure with respect to water or ice |
| $P$ | MPa | Absolute pressure within liquid phase |
| $P_N$ | – | Active-site probability fit function |
| $P_0$ | MPa | Standard pressure (0.1 MPa) |
| $\Delta P$ | MPa | Pressure difference across water–water vapour interface of curved surface |
| PCF | – | Pore condensation and freezing |
| $r$ | m | Radius of spherical ice cluster |
| $r_c$ | m | Radius of curvature of the water surface |
| $r_{crit}$ | m | Radius of critical ice germ |
| $r_{c,fl,tr}$ | m | Principle radius of curvature of the meniscus at filling level ($h_{fl}$) in three-membered ring pore |
| $r_{c,fl,sq}$ | m | Principle radius of curvature of the meniscus at filling level ($h_{fl}$) in four-membered ring pore |
| $r_{c,tr}$ | m | Radius of circumcircle around three-membered ring pore |
| $r_{c,sq}$ | m | Radius of circumcircle around four-membered ring pore |
| $r_{i,sq}$ | m | Radius of incircle of square with apices at the centres of four neighbouring spheres (cubic packing arrangement) |
| $r_{i,te}$ | m | Radius of sphere within concave octahedron (tetrahedral packing arrangement) |
| $r_{i,tr}$ | m | Radius of incircle tangent to the sides of the equilateral triangle with corners at the centres of three nearest neighbouring spheres |
| $r_{o,cu}$ | m | Radius of sphere between eight spheres arranged on the corners of a cube |
| $r_{o,sq}$ | m | Radius of opening of four-membered ring pore |
| $r_{o,tr}$ | m | Radius of opening of three-membered ring pore |
| $r_{o,te}$ | m | Radius of sphere within concave octahedron (tetrahedral packing arrangement) |
| $r_{fl,tr}$ | m | Radius of three-membered ring pore at filling level |
| $r_{fl,sq}$ | m | Radius of four-membered ring pore at filling level |
| $r_{c1}, r_{c2}$ | m | Principal radii of curvature of curved water surface |
| $R_g$ | m | Radius of gyration |
| $RH_i$ | % | Relative humidity with respect to ice |
| $RH_w$ | % | Relative humidity with respect to water |

| Symbol | Unit | Description |
|---|---|---|
| $S_i$ | – | Saturation ratio with respect to ice |
| $S_w$ | – | Saturation ratio with respect to water |
| $T$ | K | Absolute temperature |
| TEM | – | Transmission electron microscopy |
| $v_i(T)$ | $m^3$ | Molecular volume of water in the ice phase |
| $v_w(T)$ | $m^3$ | Molecular volume of liquid water |
| $\beta$ | – | Ratio of mobility radius to radius of gyration of soot aggregate |
| $\gamma(T)$ | $N\,m^{-1}$ | General interfacial tension between two phases |
| $\gamma_{iw}(T)$ | $N\,m^{-1}$ | Interfacial tension between ice and water |
| $\gamma_{si}(T)$ | $N\,m^{-1}$ | Interfacial tension between soot and ice |
| $\gamma_{sw}(T)$ | $N\,m^{-1}$ | Interfacial tension between soot and water |
| $\gamma_{vi}(T)$ | $N\,m^{-1}$ | Interfacial tension between water vapour and ice |
| $\gamma_{vw}(T)$ | $N\,m^{-1}$ | Interfacial tension between water vapour and water |
| $\gamma_{vs}(T)$ | $N\,m^{-1}$ | Interfacial tension between water vapour and soot |
| $\delta$ | $^\circ$ | Pore-opening angle for conical pores |
| $\Delta G_{w,i}(T,P)$ | J | Change in Gibbs free energy of homogeneous ice nucleation within supercooled liquid water |
| $\varepsilon$ | $^\circ$ | Slit-filling angle/growth angle |
| $\varepsilon_{ov}$ | $^\circ$ | Angle of primary particle overlap, describing the angle between the centre line between the particle centres and the height of the overlap |
| $\theta$ | $^\circ$ | General contact angle |
| $\theta_{iw}$ | $^\circ$ | Contact angle between ice and water |
| $\theta_{si}$ | $^\circ$ | Contact angle between soot and ice |
| $\theta_{sw}$ | $^\circ$ | Contact angle between soot and water |
| $\mu_i(T,P)$ | J | Chemical potential of ice |
| $\mu_w(T,P)$ | J | Chemical potential of water |

*Data availability.* The data used for the parameterization of soot PCF can be found at https://doi.org/10.3929/ethz-b-000286409 (Mahrt et al., 2018b TS22) and at https://doi.org/10.3929/ethz-b-000340269 (Mahrt et al., 2019 TS23).

*Author contributions.* CM conceived the idea for the study. The theoretical soot-PCF framework was developed by CM and FM, who also prepared the figures and wrote the manuscript with contributions from BK. All authors were involved in discussions of results and data interpretation.

*Competing interests.* The authors declare that they have no conflict of interest.

*Acknowledgements.* We thank Kyeni S. Mbiti for constructing and rendering the 3D figures. We thank Franz Friebel and Zamin A. Kanji for reading the manuscript and for their helpful comments. We would like to thank the reviewers for helpful and constructive comments.

*Financial support.* This research has been supported by the European Union's Horizon 2020 research and innovation programme under the Marie Skłodowska-Curie grant (grant no. 890200).

*Review statement.* This paper was edited by Hinrich Grothe and reviewed by three anonymous referees.

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

## Remarks from the language copy-editor

## Remarks from the typesetter