# Peer review of "Soot-PCF: Pore condensation and freezing framework for soot aggregates"

_Atmospheric Chemistry and Physics, 2020_

## Referee Comment (RC1) · Anonymous Referee #2 · 17 Dec 2020

In this study, authors describe the ice formation dynamics within the soot aggregates. They describe and parameterize the relationship between various thermodynamic variables that control the nucleation of ice and soot morphology. The paper is well written, the analysis is original in such that they extend the previously PCF based theory to soot aggregates, and the development of a parameterization to predict AF is unique. I have few minor comments, and after addressing them I recommend the paper for publication.

Abstract: This section is bit long. To improve the readability, I would suggest reduce the length of the abstract. Some of the sentences can be moved to the main text.

Section 8: Soot particles in the atmosphere undergo aging, and the morphology of soot aggregates changes rapidly. The co-emitted gases (in biomass burning kind of

event) condense and modify the physical and chemical properties. Can the theoretical framework (equation 9) be applied in such case? The related question is how one can model if some pores are partially filled with the sulfates and organics. This will change or at least alter the ice formation mechanism. Any thoughts on applying this framework in a transient environment where the soot properties are evolving in time? There is vast literature on soot ice nucleation onset and AF at temperature less than -38 degC. How this previous data compares with the Fig 6? I wonder if it is possible to plot the Fig 6 data in RH-T space (RH on y-axis and Temperature on x-axis) to compare against literature data. AF can be colored with the marker size or colorbar. Can this parameterization (equation 9) be modified to take into account the vertical velocity (the information typically needed to model the cirrus clouds)?

Line 749: Discussion on aerosol-lamina is not clear. Is this means some particles were outside the lamina and did not had a chance to induce nucleation of ice? Does this artifact also affect all other measurements?

Line 752: what are other examples of previous parameterizations?

---

## Referee Comment (RC2) · Anonymous Referee #3 · 22 Dec 2020

The authors present an extensive theoretical framework to describe the heterogeneous ice nucleation ability of soot particles at cirrus conditions based on the pore condensation and freezing (PCF) mechanism. They provide a very careful analysis of the dependence of the individual steps of the PCF process, i.e., capillary condensation of water, homogeneous freezing of the "water pockets", and ice growth out of the pores, on the geometry of the pore types that can typically occur in soot aggregates and on the soot-water contact angle. The pore geometries were modelled/calculated based on primary particle size and particle overlap. Overall, the analysis provides some very nice insights into the pathway of forming macroscopic ice crystals on an "active site" on the soot surface, which can be limited, depending on the pore size and geometry, either by capillary condensation or ice growth out of the pore (see analysis of Fig. 5).

The approach is also useful to estimate the particles' susceptibility to pre-activation (see Sect. 7).

One point of criticism for me is the in some aspects somewhat limited discussion of the new parameterization introduced in Eq. (9). The authors present a rigorous mathematical treatment of the various steps in the PCF mechanism, but the motivation for choosing the specific formula to quantify the ice-active fraction in Eq. (9) is much less described. Also, the authors state that the parameterization can predict the ice activity of a soot sample (line 853) and that the active-site probability function can be derived from first principles (line 705), but in the end they just use in their current work some empirical formula to derive the active site probability function from fits to activated fractions measured for two soot types in a continuous flow diffusion chamber (Eq. E4 and E5). Furthermore, it is mentioned that the new approach has advantages over previously proposed parameterizations, but no comparison with other literature data is shown in Fig. 6. If these points are explained a little better, I will be very happy to accept the manuscript for publication in ACP.

Specific comments:

Line 75-77: Could you incorporate these literature data to Fig. 6 to compare them with the Mahrt et al. data?

Line 86: You specifically mention here "hydrophobic material" to suppress PCF. But wouldn't the same be true for water-soluble components like sulfates, because solutes would decrease the homogeneous freezing temperature compared to pure water?

Line 116: You state here that you "predict" the ice nucleation ability of soot particles which then "fits" experimental results – but in the end with Eq. E4 and E5 you use empirical formulas to derive the active site probability function from the experiments. So currently it is still more the development of a new framework, by which the experimental data should be fitted, than an a priori prediction of the ice nucleation ability of a soot sample.

Line 120: Given the length of the manuscript and the numerous aspects of the discussion, you might add here a short paragraph describing the general structure of the article.

Line 209: The heading of Sect. 2.3 "promises" some information on "compaction", but actually I couldn't find much of it in this section. Given that compaction is indeed an important process which affects the number of pores in a soot aggregate (line 141) and also appears as a parameter (via the fractal dimension) in Eq. 9, I would like to see some more discussion of this, see also my later comment regarding Eq. 9.

Line 264: Isn't is below that of bulk water?

Line 379: Can you briefly explain in the text what is the parameter "n"?

Line 590-593: You mention here the relatively large variability of the onset conditions in terms of S_ice with the chosen values for the contact angle, overlap factor, and primary particle size. It would be interesting, see comment above, to compare these onsets with existing literature data on ice nucleation by soot particles.

Line 703: Is there a "physical basis" behind the equation you used to represent the ice active fraction (Eq. 9), e.g. regarding the exponent defined as (Np-nm)ˆDf? Is there a quantitative relationship to describe the number of pores in a soot aggregate as a function of primary particle size and compaction which is reflected by this exponent or was this expression just found to be a suited formulation to represent that the AF increases with Np and Df? Lines 1175/1176 state that Np and Df determine the probability that a soot aggregate contains a ring pore, but line 703 states that the factor Pn(RH) also is the probability of a primary particle to be part of ring pore with given ice nucleating potential. I am a bit confused by this interpretation, is Pn(RH) not more of a factor that describes a kind of "averaged" ice nucleation ability of the pores present in the soot aggregate and the exponent (Np-nm)ˆDf a measure for the absolute number of pores in the aggregate? Maybe you could elaborate a bit more on the physical meaning of these parameters and clarify my potential misconception.

You mention in line 771-774 the very interesting finding from Mahrt et al. that the ice nucleation ability of compacted soot particles was found to be significantly higher compared to unprocessed, more fractal soot. Have you tried to model or even predict this behavior with your new parameterization? This would be in my opinion clear evidence of the advantages of your approach compared to previous parameterizations. Would compaction only affect the parameter Df so that one could use the same active-site probability function of a given soot type both before and after compaction? Or would compaction also alter Pn(RH) (a question which is somehow related to my comment above regarding the physical meaning of the parameters)? Also, is it immediately intuitive that compaction, i.e., a higher Df, leads to an increase in the AF? Would compaction not also lead to the situation that some potentially ice-active pores are no longer in the vicinity of the particle surface but somehow "shielded" in the inner part of the particle and can thus no longer contribute to macroscopic ice growth? This also relates to my question above how one can determine the number of pores that are actually accessible to the PCF depending on agglomerate size and compaction.

Line 705: You mention that Pn(RH) can be computed from first principles for a given soot type. But how would that look in practice – you would have to deal with a large parameter range with respect to primary particle size, overlap factor, and contact angle even for a single soot type, as reviewed in the first chapters of your manuscript, which then also depend on fuel type and operation conditions of the engine?

Line 751-753: You mention here the previously proposed parameterizations. Which are these exactly and can you provide some quantitative comparison with your new parameterization? I assume that in some of the previous parameterizations the activated fraction was normalized to the surface area of the aerosol (soot) particles to yield the so-called ice nucleation active surface site (INAS), whereas your approach specifically considers the number of active sites, i.e., pores in the soot aggregates, which in principle, no doubt, is the better physically-constrained approach. But how big would be the difference? I think it would be an important information for the modelling community to quantify the difference of your new theoretical framework to the widespread INAS approach, and to investigate with your size-resolved ice nucleation measurements and modelling simulations in Fig. 6 whether such an approach that only relies on the overall particle surface area is in accordance with the data or not. With a more quantitative comparison to previous parameterizations, you could really underline the advantages of your approach.

Line 855: I would rather say your parameterization increases the complexity of the parameters. You emphasize that you need different fit values for each soot type, and instead of just using the overall aerosol surface area as a parameter, you need additional values for the primary particles size and the degree of compaction. Depending on the soot type, you also discriminate between parameterizations with respect to either RHi or RHw (Eq. E4 and E5). This is by no means meant as a criticism of your approach – but my understanding of your study is rather that you need more parameters to characterize the actual ice active sites in the soot agglomerates using a physically-constrained approach so that you can properly describe their ice nucleation ability.

Line 871: "may vastly improve" – is a rather strong statement which also requires a more quantitative comparison to previous parameterizations.

Line 882: "have used available literature data" – yes, but there are many more previous measurements which you could include in Fig. 6, as suggested above.

Line 1004/1005: What is meant here with "example of a pore of 8 nm diameter"? Don't you calculate the saturation ratios for the whole range of filling levels?

Some technical corrections:

Line 38: Delete "of" before "ice formation".

Line 54: "major gaps exist".

Line 70: add "freezing", i.e. condensation/immersion freezing mode.

Line 187: Maybe add comma after "Overall".

Line 267: Delete comma after "graphite".

Line 324: Delete comma after "cell".

Line 403: "in the immersion freezing mode".

Line 546: please add: "growing ice phase out of a three-membered ring pore"

Line 568/569: You may briefly add here that the point for a filling level of 1.25 nm is indicated by "A" in Fig. 5.

Line 1077: "randomly" fluctuating

---

## Author Comment (AC1) · 4 Mar 2021

**Responses to Anonymous Referee #2**

*We thank the reviewer for his/her constructive comments that we address below point by point (responses are in italic).*

In this study, authors describe the ice formation dynamics within the soot aggregates. They describe and parameterize the relationship between various thermodynamic variables that control the nucleation of ice and soot morphology. The paper is well written, the analysis is original in such that they extend the previously PCF based theory to soot aggregates, and the development of a parameterization to predict AF is unique. I have few minor comments, and after addressing them I recommend the paper for publication.

1) Abstract: This section is bit long. To improve the readability, I would suggest reduce the length of the abstract. Some of the sentences can be moved to the main text.

*We have shortened the abstract.*

2) Section 8: Soot particles in the atmosphere undergo aging, and the morphology of soot aggregates changes rapidly. The co-emitted gases (in biomass burning kind of event) condense and modify the physical and chemical properties. Can the theoretical framework (equation 9) be applied in such case? The related question is how one can model if some pores are partially filled with the sulfates and organics. This will change or at least alter the ice formation mechanism. Any thoughts on applying this framework in a transient environment where the soot properties are evolving in time?

*This is a highly relevant question that cannot be fully answered within this study based on the currently available experimental data. We have addressed the different roles of aging occurring on soot particles acquiring coatings in Sect. 9, and extended the specific discussion in the revised manuscript to include the following points:*

*"In general, coatings of soot aggregates with semi-volatile material are expected to inhibit PCF. Yet, thin coatings with hydrophilic material should only have a minor effect. In general, they will (i) shift condensation of pore water to lower RH, (ii) lead to a freezing point depression, and (iii) may facilitate ice growth if diluted aqueous coatings wet the soot surface. Thus, the inhibiting effect of thin coatings should be most pronounced in the temperature range around 230 K where homogeneous ice nucleation rates are critical."*

*Thicker, more hydrophobic or viscous and glassy coatings will block water uptake by pores and inhibit PCF, while thick hydrophilic coatings will shift the freezing mode to immersion freezing or homogeneous freezing of solution drops depending on the ice activity of the specific soot in immersion mode. This is now addressed in the revised manuscript. Specifically, we have added the following:*

*"When sulfuric acid condensation leads to thick coatings, the freezing mode will shift to immersion or homogeneous freezing of solution droplets, depending on the ice activity of the specific soot in immersion mode. Whether a coating inhibits PCF depends on the amount of condensed material and on its distribution on the soot aggregate. Semi-volatile species may preferentially condense within the pores and block them for water condensation. Hydrophilic coatings do not spread on soot particles as an even coating but will form patches or droplets on soot aggregates, depending on the contact angle. A reasonable value to discriminate thin from thick coatings might be 20 % of the soot mass as chosen by Ullrich et al. (2017)."*

*Lastly, we acknowledge that the currently available experimental data on the impact of coating on the ice nucleation activity of soot particles is insufficient to make reliable and quantitative estimates about the amount of coating required to inhibit soot-PCF, which is addressed in a short outlook. Please see our answer to comment 17 of reviewer 3.*

3) There is vast literature on soot ice nucleation onset and AF at temperature less than -38 degC. How this previous data compares with the Fig 6? I wonder if it is possible to plot the

Fig 6 data in RH-T space (RH on y-axis and Temperature on x-axis) to compare against litera-ture data. AF can be colored with the marker size or colorbar.

*Ice nucleation studies often only report ice formation onsets in terms of relative humidity and temperature, for AF thresholds typically lying between 0.1 to 1 %. Based on these studies, many soot samples have been found to activate along the homogeneous freezing line of solution drop-lets (Koop et al., 2000), while some form ice well below these conditions, as can be seen e.g. in Fig. 1-7 of Kanji et al. (2017). We have added a table to Appendix F, that reports ice nucleation data together with physicochemical characterization reported in literature. Moreover, we discuss these data and set them into perspective with the soot-PCF framework presented here. To further acknowledge the available literature data on soot ice nucleation at cirrus conditions, we have added a new figure to our introduction that summarizes previous work.*

*Unfortunately, hardly any of the literature data is sufficiently detailed to develop an empirical soot-PCF parameterization, since most of them just report ice onset RH and T for one mobility diameter and/or do not report the physicochemical properties of the soot particles investigated. What is lacking in most cases is the whole AF curve up to homogeneous ice nucleation to obtain the fraction of particles that exhibits pores suited for soot-PCF and in many cases the primary particle size. We have specified this in the revised manuscript, as written in our answer to com-ment 17 of reviewer 3.*

4) Can this parameterization (equation 9) be modified to take into account the vertical velocity (the information typically needed to model the cirrus clouds)?

*Indeed, the vertical velocity is relevant for homogeneous ice nucleation, which is a stochastic pro-cess and should be parameterized as a temperature dependent rate. For soot-PCF, pore conden-sation and ice growth are deterministically determined by the critical ice activation RH. Therefore parameterizations should provide time-independent, ice active fractions similar to Eq. (9) instead of nucleation rates. Vertical velocity therefore does not need to be taken into account. We have clarified this by adding the following statement to Appendix E:*

*"For soot-PCF as described in Eq. (E1), macroscopic ice formation is determined by the processes of pore filling or ice growth out of the pore, which both occur deterministically at a critical RH. Only at temperatures around 230 K, where the homogeneous ice nucleation rates may be critical, the stochastic nature of ice nucleation can be relevant. Therefore, vertical velocity, which is a determining factor for the number of homogeneously nucleated ice crystals (Kärcher and Lohmann, 2002; Hoyle et al., 2005; Sullivan et al., 2016), does not influence the number of ice crystals nucleated through soot-PCF and can be neglected in the soot-PCF framework."*

5) Line 749: Discussion on aerosol-lamina is not clear. Is this means some particles were outside the lamina and did not had a chance to induce nucleation of ice? Does this artifact also affect all other measurements?

*The measurement of the ice nucleation activity of different soot types shown in Fig. 6 were per-formed with a continuous flow diffusion chamber (CFDC). The general working principle of a CFDC is detailed elsewhere (Rogers, 1988) and a more detailed characterization of the CFDC used to sample the data shown in Fig. 6, called the horizontal ice nucleation chamber (HINC), has been published in previous studies (Kanji and Abbatt, 2009; Lacher et al., 2017; Mahrt et al., 2018). In brief, in a CFDC a linear temperature gradient is established between two ice-coated walls, re-sulting in a parabolic water vapor supersaturation profile, due to the non-linear relationship be-tween temperature and saturation vapor pressure, as described by the Clausius-Clapeyron rela-tion. In an attempt to expose the aerosol particles in the CFDC to distinct temperature and relative humidity conditions, the aerosol flow in a CFDC is sandwiched by a particle free sheath flow. This sheath flow constrains the particles to a small region within the CFDC, called the aerosol-lamina (that largely coincides with the peak of the supersaturation profile), where the variation of RH and T is smallest. The ratio of the aerosol-to-sheath flow determines the width of the aerosol-lamina and hence the variation in RH and T that the aerosol particles are exposed to while trav-elling through the CFDC. A detailed discussion of the resulting uncertainties for the instrument used to sample the data in Fig. 6 can be found in Appendix D of Mahrt et al. (2018). The RH and*

*T conditions in this aerosol-lamina correspond to the ice nucleation conditions that are reported in most studies. While even across the aerosol lamina some variation in RH and T exists, the assumption of ideal instrument behavior, i.e. all particles remaining in the aerosol-lamina, is incorrect and to some extent also impacted by the difference in wall temperature, as demonstrated by previous studies (DeMott et al., 2015; Garimella et al., 2017). Specifically the comprehensive study by Garimella et al. (2017) performed experiments using two different CFDCs and demonstrated that some fraction of the injected aerosol particles leave the lamina, ultimately resulting in correction factors to the measured AF between 1.5 to 9.5. Both CFDCs tested by Garimella et al. (2017) had a vertical orientation. In this design, a buoyant flow of relatively warmer air close to the warmer CFDC wall causes a replacement of the aerosol-lamina (as a function of the wall temperature difference). Such a buoyant airflow is absent in CFDCs with horizontal orientation, where the warmer wall is above the colder wall. As a result, the AF correction factors for horizontally oriented CFDCs, such as the one used to measure the data shown in Fig. 6, should be different and likely lower. However, we are not aware of any systematic quantification of such a correction factor for horizontally oriented chambers, and to the data shown in Fig. 6 no correction factor has been applied.*

*In conclusion, particles leaving the aerosol-lamina are likely an inherent issue in all types of CFDCs. We have modified the relevant lines to direct the reader to the appropriate studies that focus on discussing this issue in more detail. Specifically, our revised paper reads:*

*"This suggests that the levelling off observed for the experimental data points at AF $\approx 0.2$ results from instrumental limitations. Specifically, for the data depicted in Fig. 6, that were sampled with a continuous flow diffusion chamber (CFDC), it could result from divergence of particles out of the region of highest RH conditions within the CFDC, the so-called aerosol-lamina, as has been demonstrated in previous studies (e.g. DeMott et al., 2015; Garimella et al., 2017)."*

6) Line 752: what are other examples of previous parameterizations?

*We have specified this by making the following additions to the revised manuscript in Sects. 8 and 9:*

*"The way current parameterizations in global climate models predict the ice nucleation activity of soot particles are hampered by a number of factors. Most importantly, they often do not include a size dependence, but simplistically assume a fixed percentage of ice active soot particles ranging from 0.1 to 100 % with ice activation occurring at one distinct $RH_i$ level (Gettelman et al., 2012; Gettelman and Chen, 2013; e.g. Hendricks et al., 2005; Penner et al., 2009; Wang and Penner, 2010; Zhou and Penner, 2014). Detailed cirrus models with spectral aerosol representations are capable of accommodating size- and supersaturation-dependent INP activity (Jensen et al., 2013, 2018; Kärcher, 2020; Kienast-Sjögren et al., 2015), leading to more realistic simulations of indirect aerosol effects on cirrus. While accounting for soot aggregate size is indispensable to correctly estimate the number of primary particles and pores present that ultimately determine the ice nucleation activity, explicitly measuring the size- and relative humidity dependent ice nucleation activity has only been the focus of a limited number of studies to date. Recently, Lohmann et al. (2020) used a more sophisticated parameterization derived from the ice nucleation activity of 400 nm miniCAST black soot aggregates, measured at 233 K and 218 K, with AF being a function of $RH_i$ but still independent of aggregate size. Moreover, Zhou and Penner (2014) and Lohmann et al. (2020) assumed that soot aggregates are rendered inactive over time upon acquiring a coating of three or one monolayer, respectively. Compared to these parameterizations the soot-PCF parameterization denotes a key step forward as it comprehensively reflects the strong size dependence of ice nucleation on soot aggregates and can easily be adapted to specific soot properties."*

*As another example, Ullrich et al. (2017) present an empirical active site parameterization relying on AIDA campaigns carried out with different types of soot. As this parameterization assumes an active site density per surface area, it includes a size-dependence, but it does not differentiate between more hydrophilic and hydrophobic soots. Moreover, its temperature dependence does not reflect the data well as it shows a continuous decrease with increasing temperature even*

*above the homogeneous nucleation threshold. Nevertheless, the parameterization by Ullrich et al. (2017) requires five fitting parameters and just describes the isolines of the ice nucleation activity, but does not have a deeper physical background. We further discuss this point in our answer to comment 14 of reviewer 3.*

**References:**

DeMott, P. J., Prenni, A. J., McMeeking, G. R., Sullivan, R. C., Petters, M. D., Tobo, Y., Niemand, M., Mohler, O., Snider, J. R., Wang, Z. and Kreidenweis, S. M.: Integrating laboratory and field data to quantify the immersion freezing ice nucleation activity of mineral dust particles, Atmospheric Chemistry and Physics, 15(1), 393–409, https://doi.org/10.5194/acp-15-393-2015, 2015.

Garimella, S., Rothenberg, D. A., Wolf, M. J., David, R. O., Kanji, Z. A., Wang, C., Roesch, M. and Cziczo, D. J.: Uncertainty in counting ice nucleating particles with continuous diffusion flow chambers, Atmos. Chem. Phys. Discuss., 2017, 1–28, https://doi.org/10.5194/acp-2016-1180, 2017.

Gettelman, A. and Chen, C.: The climate impact of aviation aerosols, Geophysical Research Letters, 40(11), 2785–2789, https://doi.org/10.1002/grl.50520, 2013.

Gettelman, A., Liu, X., Barahona, D., Lohmann, U. and Chen, C.: Climate impacts of ice nucleation, Journal of Geophysical Research-Atmospheres, 117, https://doi.org/10.1029/2012jd017950, 2012.

Hendricks, J., Kärcher, B., Lohmann, U. and Ponater, M.: Do aircraft black carbon emissions affect cirrus clouds on the global scale?, Geophysical Research Letters, 32(12), https://doi.org/10.1029/2005GL022740, 2005.

Hoyle, C. R., Luo, B. P. and Peter, T.: The Origin of High Ice Crystal Number Densities in Cirrus Clouds, J. Atmospheric Sci., 62(7), 2568–2579, https://doi.org/10.1175/JAS3487.1, 2005.

Jensen, E. J., Lawson, R. P., Bergman, J. W., Pfister, L., Bui, T. P. and Schmitt, C. G.: Physical processes controlling ice concentrations in synoptically forced, midlatitude cirrus, J. Geophys. Res. Atmospheres, 118(11), 5348–5360, https://doi.org/10.1002/jgrd.50421, 2013.

Kanji, Z. A. and Abbatt, J. P. D.: The University of Toronto Continuous Flow Diffusion Chamber (UT-CFDC): A Simple Design for Ice Nucleation Studies, Aerosol Science and Technology, 43(7), 730–738, https://doi.org/10.1080/02786820902889861, 2009.

Kanji, Z. A., Ladino, L. A., Wex, H., Boose, Y., Burkert-Kohn, M., Cziczo, D. J. and Krämer, M.: Overview of Ice Nucleating Particles, Meteorological Monographs, 58(0), 1.1-1.33, https://doi.org/10.1175/amsmonographs-d-16-0006.1, 2017.

Kärcher, B.: Process-Based Simulation of Aerosol-Cloud Interactions in a One-Dimensional Cirrus Model, J. Geophys. Res. Atmospheres, 125(6), e2019JD031847, https://doi.org/10.1029/2019JD031847, 2020.

Kärcher, B. and Lohmann, U.: A Parameterization of cirrus cloud formation: Homogeneous freezing including effects of aerosol size, J. Geophys. Res. Atmospheres, 107(D23), AAC 9-1-AAC 9-10, https://doi.org/10.1029/2001JD001429, 2002.

Kienast-Sjögren, E., Miltenberger, A. K., Luo, B. P. and Peter, T.: Sensitivities of Lagrangian modelling of mid-latitude cirrus clouds to trajectory data quality, Atmospheric Chem. Phys., 15(13), 7429–7447, https://doi.org/10.5194/acp-15-7429-2015, 2015.

Koop, T., Luo, B. P., Tsias, A. and Peter, T.: Water activity as the determinant for homogeneous ice nucleation in aqueous solutions, Nature, 406(6796), 611–614, https://doi.org/10.1038/35020537, 2000.

Lacher, L., Lohmann, U., Boose, Y., Zipori, A., Herrmann, E., Bukowiecki, N., Steinbacher, M. and Kanji, Z. A.: The Horizontal Ice Nucleation Chamber (HINC): INP measurements at conditions relevant for mixed-phase clouds at the High Altitude Research Station Jungfraujoch, Atmos Chem Phys, 17(24), 15199–15224, https://doi.org/10.5194/acp-17-15199-2017, 2017.

Lohmann, U., Friebel, F., Kanji, Z. A., Mahrt, F., Mensah, A. A. and Neubauer, D.: Future warming exacerbated by aged-soot effect on cloud formation, Nature Geoscience, 13(10), 674–680, https://doi.org/10.1038/s41561-020-0631-0, 2020.

Mahrt, F., Marcolli, C., David, R. O., Grönquist, P., Barthazy Meier, E. J., Lohmann, U. and Kanji, Z. A.: Ice nucleation abilities of soot particles determined with the Horizontal Ice Nucleation Chamber, Atmos. Chem. Phys., 18(18), 13363–13392, https://doi.org/10.5194/acp-18-13363-2018, 2018.

Marcolli, C., Gedamke, S., Peter, T. and Zobrist, B.: Efficiency of immersion mode ice nucleation on surrogates of mineral dust, Atmos. Chem. Phys., 7(19), 5081–5091, https://doi.org/10.5194/acp-7-5081-2007, 2007.

Penner, J. E., Chen, Y., Wang, M. and Liu, X.: Possible influence of anthropogenic aerosols on cirrus clouds and anthropogenic forcing, Atmospheric Chemistry and Physics, 9(3), 879–896, https://doi.org/10.5194/acp-9-879-2009, 2009.

Rogers, D. C.: Development of a continuous flow thermal gradient diffusion chamber for ice nucleation studies, Atmospheric Research, 22(2), 149–181, https://doi.org/10.1016/0169-8095(88)90005-1, 1988.

Sullivan, S. C., Lee, D., Oreopoulos, L. and Nenes, A.: Role of updraft velocity in temporal variability of global cloud hydrometeor number, Proc. Natl. Acad. Sci., 113(21), 5791–5796, https://doi.org/10.1073/pnas.1514039113, 2016.

Ullrich, R., Hoose, C., Möhler, O., Niemand, M., Wagner, R., Höhler, K., Hiranuma, N., Saathoff, H. and Leisner, T.: A New Ice Nucleation Active Site Parameterization for Desert Dust and Soot, Journal of the Atmospheric Sciences, 74(3), 699–717, https://doi.org/10.1175/jas-d-16-0074.1, 2017.

Vali, G.: Interpretation of freezing nucleation experiments: singular and stochastic; sites and surfaces, Atmospheric Chemistry and Physics, 14(11), 5271–5294, https://doi.org/10.5194/acp-14-5271-2014, 2014.

Wang, M. and Penner, J. E.: Cirrus clouds in a global climate model with a statistical cirrus cloud scheme, Atmospheric Chemistry and Physics, 10(12), 5449–5474, https://doi.org/10.5194/acp-10-5449-2010, 2010.

Zhou, C. and Penner, J. E.: Aircraft soot indirect effect on large-scale cirrus clouds: Is the indirect forcing by aircraft soot positive or negative?, Journal of Geophysical Research: Atmospheres, 119(19), 11,303-11,320, https://doi.org/10.1002/2014JD021914, 2014.

---

## Author Comment (AC2) · 4 Mar 2021

**Responses to Anonymous Referee #3**

The authors present an extensive theoretical framework to describe the heterogeneous ice nucleation ability of soot particles at cirrus conditions based on the pore condensation and freezing (PCF) mechanism. They provide a very careful analysis of the dependence of the individual steps of the PCF process, i.e., capillary condensation of water, homogeneous freezing of the "water pockets", and ice growth out of the pores, on the geometry of the pore types that can typically occur in soot aggregates and on the soot-water contact angle. The pore geometries were modelled/calculated based on primary particle size and particle overlap. Overall, the analysis provides some very nice insights into the pathway of forming macroscopic ice crystals on an "active site" on the soot surface, which can be limited, depending on the pore size and geometry, either by capillary condensation or ice growth out of the pore (see analysis of Fig. 5).

*We thank the reviewer for the detailed assessment of our manuscript and his/her constructive comments that we address below point by point (responses are blue and in italic).*

1) The approach is also useful to estimate the particles' susceptibility to pre-activation (see Sect. 7). One point of criticism for me is the in some aspects somewhat limited discussion of the new parameterization introduced in Eq. (9). The authors present a rigorous mathematical treatment of the various steps in the PCF mechanism, but the motivation for choosing the specific formula to quantify the ice-active fraction in Eq. (9) is much less described. Also, the authors state that the parameterization can predict the ice activity of a soot sample (line 853) and that the active-site probability function can be derived from first principles (line 705), but in the end they just use in their current work some empirical formula to derive the active site probability function from fits to activated fractions measured for two soot types in a continuous flow diffusion chamber (Eq. E4 and E5).

*In order to address this comment, we have significantly expanded the description and derivation of Eq. 9. Specifically, we have added the following text to Appendix E:*

*"To predict soot-PCF, we have developed an equation that quantifies the ice active fraction as a function of RH based on the combined probability $P_N(RH)$, that n-membered ring pores nucleate ice at a given RH on the one hand and the probability that a soot aggregate contains such a ring pore on the other hand . By splitting up the ice nucleation probability into two parts, we define a probability function $P_N(RH)$ that depends on the properties of a single ring pore, namely the primary particle size, overlap and contact angle, and apply it then to aggregates of different size, compaction and number of (equally-sized) primary particles. In other words, $P_N(RH)$ defines the probability per (single) primary particle to be part of a ring-pore structure.*
*To calculate the probability that an aggregate contains a ring pore, we start by calculating the complementary probability, namely that an aggregate made of $N_p$ primary particles does not contain any primary particle belonging to a ring pore, i.e. $(1 - P_N(RH))^{N_p}$. However, it is important to consider that at least three (four) primary particles are needed to form a three-membered (four-membered) ring pore. Therefore, this probability, needs to be decreased such that three and four primary particles just have one option to form a three-membered and four-membered ring pore, respectively, leading to $(1 - P_N(RH))^{N_p - n_m}$, where $n_m$ = 2 in the case of three-membered ring pores and $n_m$ = 3 in the case of four-membered ring pores. If both, three-membered and four-membered ring pores are present on an aggregate, the use of $n_m$ = 2 is advised, to avoid underrepresentation of ring pores in small aggregates.*
*This probability would be valid for ring pores occurring on a string or chain-like structure (1D) of primary particles. If instead primary particles cover an area (2D) or fill a volume (3D), a primary particle has more neighbouring primary particles with which it can form a ring pore. We account for these multiple options by potentiating the probability by the fractal dimension $D_f$, ranging from values of 1 for chain-like structures to 3 for perfectly spherical soot aggregates. Taken altogether, the activated fraction is then given as:*

$$AF(RH) = 1 - \left(1 - P_N(RH)\right)^{\left((N_p - n_m)^{D_f}\right)}. \qquad (E1)$$

*The fractal dimension determines the spacing between the AF-curves of soot aggregates with*

*different mobility diameters. Since experimental derivation of the true three-dimensional fractal dimension is challenging and associated with uncertainties or just derived from two dimensional TEM images (see Sect. 2.3), $D_f$ can also be used as a fit parameter to adjust modelled AF curves to measurements for aggregates of different (mobility) sizes.*

*For Soot-PCF as described in Eq. (E1), macroscopic ice formation is determined by the processes of pore filling or ice growth out of the pore, which both occur deterministically at a critical RH. Only at temperatures around 230 K, where the homogeneous ice nucleation rates may be critical, the stochastic nature of ice nucleation can be relevant. Therefore, vertical velocity, which is a determining factor for the number of homogeneously nucleated ice crystals* (Hoyle et al., 2005; Kärcher and Lohmann, 2002; Sullivan et al., 2016) *(Kärcher and Lohmann, 2002; Hoyle et al., 2005; Sullivan et al., 2016), does not influence the number of ice crystals nucleated through soot-PCF and can be neglected in the soot-PCF framework. It should further be noted that the function $P_N(RH)$ can either be used to bring AF(RH) in agreement with an experimental dataset or can be derived from soot properties by inspecting the different onset RH required to nucleate and grow ice out of ring pores, as shown in Appendix D. Note that within aggregates, there may be structures that are not fully closed to form ring pores. Such structures that can be viewed as ring pores with negative overlap may close to form effective ring pores during humidity cycles and cloud processing driven by capillary forces arising during condensation and evaporation (e.g. Huang et al., 1994; Ma et al., 2013). Such an increase in the pore number density would come in addition to the one expected by an increase in $D_f$ and require an adaptation of $P_N(RH)$."*

2) Furthermore, it is mentioned that the new approach has advantages over previously proposed parameterizations, but no comparison with other literature data is shown in Fig. 6.

*We believe that our soot-PCF parameterization is more elaborated than parameterizations commonly used in global models, which often treat ice nucleation on soot particles in a simplistic manner. To clarify this aspect, we added the following text to our manuscript in order to compare with ice nucleation parameterizations that are frequently used in global climate models as suggested by the reviewer:*

*"The way current parameterizations in global climate models predict the ice nucleation activity of soot particles are hampered by a number of factors. Most importantly, they often do not include a size dependence, but simplistically assume a fixed percentage of ice active soot particles ranging from 1 to 100 % with ice activation occurring at one distinct $RH_i$ level (Gettelman et al., 2012; Gettelman and Chen, 2013; e.g. Hendricks et al., 2005; Penner et al., 2009; Wang and Penner, 2010; Zhou and Penner, 2014). While accounting for soot aggregate size is indispensable to correctly estimate the number of primary particles and pores present that ultimately determine the ice nucleation activity, explicitly measuring the size- and relative humidity dependent ice nucleation activity as a function of $RH_i$ has only been the focus of a limited number of studies to date. Recently, Lohmann et al. (2020) used a more realistic parameterization derived from the ice nucleation activity of 400 nm miniCAST black soot aggregates, measured at 233 K and 218 K, with AF being a function of $RH_i$ but still independent of aggregate size. Moreover, Zhou and Penner (2014) and Lohmann et al. (2020) assumed that soot particles are rendered inactive over time upon acquiring a coating of three or one monolayer, respectively. Compared to these parameterizations soot-PCF denotes a key step forward as it comprehensively reflects the strong size and relative humidity dependence of ice nucleation on soot aggregates and can be adapted to specific soot properties."*

If these points are explained a little better, I will be very happy to accept the manuscript for publication in ACP.

Specific comments:

3) Line 75-77: Could you incorporate these literature data to Fig. 6 to compare them with the Mahrt et al. data?

*We have added a new appendix that summarizes previous literature on ice nucleation of soot*

*particles in the cirrus regime and have also included an additional overview figure into the introduction of our revised manuscript (see Fig. 1). Please directly see the new Appendix F. We do, however, not see a benefit in incorporating more data into Fig. 6, since the lines shown in Fig. 6 denote ice nucleation predicted by our soot-PCF parameterization (Eq. 9) considering the specific properties of the specific soot types shown in this figure.*

*Please note that in order to compare the ice nucleation activity predicted by the soot-PCF parametrization to that measured in previous studies, measured size-dependent ice activated fractions along with a characterization of the particle properties (e.g. primary particle size) would be desirable. Comparison is indeed limited by the lack of such data, as many studies only report onset conditions in terms of RH and T, for a certain AF that is reached. To address this issue, we have added the following statement:*

*"While accounting for soot aggregate size is indispensable to correctly estimate the number of primary particles and pores present that ultimately determine the ice nucleation activity, explicitly measuring the size- and relative humidity-dependent ice nucleation activity has only been the focus of a limited number of studies to date."*

*Furthermore, we address this issue in the outlook:*

*"Such studies should take great care in characterizing the physical and chemical properties of the soot particles, in particular those discussed herein, which will be essential to further validate the soot-PCF framework. We re-iterate that there is a clear need for dedicated ice nucleation studies on size-selected soot particles, also for aggregate sizes with (mobility) diameters below 100 nm, as well as studies exploring the impact of semi-volatile coatings on the ice nucleation ability of these particles. Lastly, we underscore the importance to report full ice activation spectra, rather than only ice nucleation onset conditions, to further improve our understanding of the physical principles determining the ice nucleation by soot particles and thereby better predict their effects on clouds and climate."*

4) Line 86: You specifically mention here "hydrophobic material" to suppress PCF. But wouldn't the same be true for water-soluble components like sulfates, because solutes would decrease the homogeneous freezing temperature compared to pure water?

*The reviewer raises a good point and is right that dissolved water-soluble components decrease the freezing temperature. This effect will be relevant in the temperature range close to the homogeneous ice nucleation threshold of aqueous solution droplets. Moreover, large amounts of hydrophilic material will shift the freezing mode from soot-PCF to immersion freezing or homogeneous ice nucleation of solution droplets. We discuss these aspects in more detail in the revised manuscript in Sect. 9. Please see our changes directly within Sect. 9 and our answer to comment 2 of reviewer 2.*

5) Line 116: You state here that you "predict" the ice nucleation ability of soot particles which then "fits" experimental results – but in the end with Eq. E4 and E5 you use empirical formulas to derive the active site probability function from the experiments. So currently it is still more the development of a new framework, by which the experimental data should be fitted, than an a priori prediction of the ice nucleation ability of a soot sample.

*Indeed, here we use $P_N(RH)$ to fit the experimental data. However, in the absence of experimental data the freezing onset for a given contact angle can be estimated based on the soot properties. We added a paragraph outlining how the parameterization can be derived based on soot properties to the revised manuscript:*

*"Soot-PCF parameterizations in the form of Eq. (9) can be derived from first principles using activation curves as the ones shown in Appendix D, presuming a contact angle combined with characteristic distributions of primary particle sizes and overlap coefficients. Weighting the ice activation RH for each combination of primary particle size and overlap coefficient with its occurrence probability leads to an ice activation probability distribution as a function of RH. If PCF is water*

*condensation limited, the parameterization should be formulated based on $RH_w$, if it is ice growth limited, it should be with respect to $RH_i$. Moreover, the temperature below which PCF becomes active needs to be defined. Assuming that soot is not ice active in immersion freezing mode, this threshold temperature depends on the increase of the homogeneous ice nucleation rate with decreasing temperature and should be around 230 K, given the small water volumes involved in forming ice via soot-PCF. Yet, more ice nucleation experiments are needed in this temperature range with ice nucleation active soot samples to better constrain this threshold temperature."*

*In conclusion, while we demonstrate the ability of the soot-PCF framework, by fitting experimental data using Eq. (9), the same framework can also be applied/transferred to other soot particles with different properties, where experimental data is absent. Once the parameters for the soot-PCF parametrization have been derived/estimated, the parametrization can be used within e.g. global climate models to predict the ice nucleation activity of soot particles with given physicochemical properties.*

6) Line 120: Given the length of the manuscript and the numerous aspects of the discussion, you might add here a short paragraph describing the general structure of the article.

*This is a good suggestion. We have added a paragraph summarizing the structure of the paper:*

*"The paper is structured as follows: in Sect. 2 we summarize the soot properties that are relevant for PCF and in Sect. 3 we define the pore structures that serve as the basis for the soot-PCF framework developed herein. In Sects. 4–6, we discuss the different steps of soot-PCF, namely ice nucleation (Sect. 4), which is preceded by pore water condensation (Sect. 5) and followed by ice growth out of the pores on the soot aggregates (Sect. 6). Section 7 discusses the relevance of preactivation for soot-PCF, while Sect. 8 presents a novel parameterization of ice nucleation by soot based on the soot-PCF framework. Atmospheric implications are discussed in Sect. 9 followed by a summary and conclusions in Sect. 10.*

*A detailed derivation of soot-PCF equations can be found in the appendices. Pore geometries are derived in Appendix A, pore filling and ice growth conditions in Appendix B. The derivation of the contact angle between ice and soot is given in Appendix C, while Appendix D features a compilation of different soot-PCF curves for relevant combinations of contact angle, primary particle diameter and overlap. A detailed derivation of the soot-PCF parameterization is given in Appendix E, followed by a compilation of ice nucleation data from previous studies (Appendix F) and a list of symbols (Appendix G)."*

7) Line 209: The heading of Sect. 2.3 "promises" some information on "compaction", but actually I couldn't find much of it in this section. Given that compaction is indeed an important process which affects the number of pores in a soot aggregate (line 141) and also appears as a parameter (via the fractal dimension) in Eq. 9, I would like to see some more discussion of this, see also my later comment regarding Eq. 9.

*Indeed, our discussion of compaction was incomplete. We therefore extended Sect. 2.3 to include a discussion of compaction and fractal dimension:*

[revised manuscript text omitted]

8) Line 264: Isn't is below that of bulk water?

*Thank you for spotting this. It is indeed below that of bulk water. We have corrected this in the revised version.*

9) Line 379: Can you briefly explain in the text what is the parameter "n"?

*We added "n" (as $n_{it}$) to the List of Symbols and Abbreviations (Appendix G) along with an explanation.*

10) Line 590-593: You mention here the relatively large variability of the onset conditions in terms of S_ice with the chosen values for the contact angle, overlap factor, and primary particle size. It would be interesting, see comment above, to compare these onsets with existing literature data on ice nucleation by soot particles.

*We added an additional appendix (Appendix F) containing a table and figure summarizing ice*

*nucleation onsets of soot reported in the literature and compare them with the onsets expected depending on contact angle.*

11) Line 703: Is there a "physical basis" behind the equation you used to represent the ice active fraction (Eq. 9), e.g. regarding the exponent defined as (Np-nm)^Df? Is there a quantitative relationship to describe the number of pores in a soot aggregate as a function of primary particle size and compaction which is reflected by this exponent or was this expression just found to be a suited formulation to represent that the AF increases with Np and Df? Lines 1175/1176 state that Np and Df determine the probability that a soot aggregate contains a ring pore, but line 703 states that the factor Pn(RH) also is the probability of a primary parti-cle to be part of ring pore with given ice nucleating potential. I am a bit confused by this interpretation, is Pn(RH) not more of a factor that describes a kind of "averaged" ice nuclea-tion ability of the pores present in the soot aggregate and the exponent (Np-nm)^Df a meas-ure for the absolute number of pores in the aggregate? Maybe you could elaborate a bit more on the physical meaning of these parameters and clarify my potential misconception.

*$P_N$(RH) defines the probability per (single) primary particle to be part of a ring-pore structure. The exponent, $\left(N_p - n_m\right)^{D_f}$, takes into account the number of primary particles available within an aggregate to form a ring pore, as well as their spatial/morphological arrangement, via the fractal dimension, $D_f$. To clarify the reviewer's concern, we have added a more detailed discussion of Eq. 9 to Appendix E. Please see our answer to your comment 1.*

12) You mention in line 771-774 the very interesting finding from Mahrt et al. that the ice nucle-ation ability of compacted soot particles was found to be significantly higher compared to unprocessed, more fractal soot. Have you tried to model or even predict this behavior with your new parameterization? This would be in my opinion clear evidence of the advantages of your approach compared to previous parameterizations. Would compaction only affect the parameter Df so that one could use the same active-site probability function of a given soot type both before and after compaction? Or would compaction also alter Pn(RH) (a ques-tion which is somehow related to my comment above regarding the physical meaning of the parameters)? Also, is it immediately intuitive that compaction, i.e., a higher Df, leads to an increase in the AF? Would compaction not also lead to the situation that some potentially ice-active pores are no longer in the vicinity of the particle surface but somehow "shielded" in the inner part of the particle and can thus no longer contribute to macroscopic ice growth? This also relates to my question above how one can determine the number of pores that are actually accessible to the PCF depending on agglomerate size and compaction.

*The pores that are responsible for soot-PCF are the narrowest within a soot aggregate. Therefore, we consider shielding (due to compaction) as irrelevant to render ring pores within soot particles ice inactive, at least for typical primary particle sizes and the degree of compaction of atmos-pheric soot particles. If the aggregate is sintered to a degree that shielding becomes relevant or when primary particles are approximately below 10 nm in diameter, the three-membered and four-membered ring pores, as described here, are unlikely to be the relevant pore structures for ice formation via soot-PCF since they require RH conditions for ice formation that are not relevant for typical upper tropospheric conditions (see our Figs. D1 and D2). For such pore structures, com-paction may render the openings of the inner cavities too narrow for ice to grow out of them.*

*Furthermore, we feel that compaction of soot aggregates will likely not only affect the fractal dimension, but also the active-site probability function $P_N$(RH), with the latter parameter domi-nating the impact on soot-PCF (please see our answer to your comments 1 and 11 above). For instance, for a soot particle with a fractal dimension well below 3 that contains open ring-pore structures, a compaction can lead to closure of these ring-pore structure, ultimately allowing such aggregates to become ice active via soot-PCF, while the fractal dimension might only be slightly increased and remains well below 3. As an example, please note that the miniCAST black data from Mahrt et al. (2018) shown in Fig. 6 use a fractal dimension of $D_f$ = 1.86, but are well described by our soot-PCF (see also Table E2). We emphasize that even in the absence of an experimentally determined fractal dimension, one can either use a fractal dimension of $D_f$ = 1.78, corresponding*

*to the typical fractal dimension of soot particles formed through diffusion limited cluster aggregation (see Appendix E), or use $D_f$ as a (free) fitting parameter (within the bounds 1–3).*

13) Line 705: You mention that Pn(RH) can be computed from first principles for a given soot type. But how would that look in practice – you would have to deal with a large parameter range with respect to primary particle size, overlap factor, and contact angle even for a single soot type, as reviewed in the first chapters of your manuscript, which then also depend on fuel type and operation conditions of the engine?

*We have added the following text to Sect. 8 to make clear how soot-PCF parameterizations can be derived from first principles. Please also see our answer to your comment 5 above.*

*Using this procedure, we have developed a soot-PCF parameterization to describe the ice nucleation activity in a separate study* (Kärcher et al., in prep.)*. In this case, we assumed a fixed, characteristic contact angle and distributions for the overlap coefficient and measured distribution of the primary particle sizes (e.g. Fig. 3 in Liati et al.* (2014)*). To calculate the number of primary particles within an aggregate of given (electrical mobility) size, we assumed the average primary particle diameter of the measured primary particle size distribution.*

14) Line 751-753: You mention here the previously proposed parameterizations. Which are these exactly and can you provide some quantitative comparison with your new parameterization? I assume that in some of the previous parameterizations the activated fraction was normalized to the surface area of the aerosol (soot) particles to yield the so-called ice nucleation active surface site (INAS), whereas your approach specifically considers the number of active sites, i.e., pores in the soot aggregates, which in principle, no doubt, is the better physically-constrained approach. But how big would be the difference? I think it would be an important information for the modelling community to quantify the difference of your new theoretical framework to the widespread INAS approach, and to investigate with your size-resolved ice nucleation measurements and modelling simulations in Fig. 6 whether such an approach that only relies on the overall particle surface area is in accordance with the data or not. With a more quantitative comparison to previous parameterizations, you could really underline the advantages of your approach.

*The reviewer raises a fair question. INAS-based parameterizations are empirical and hence completely rely on the specific datasets that were chosen for a specific parameterization, which may not always be representative for the INP categories (e.g. soot or mineral dust) that should be covered. Therefore, new datasets often require new INAS parameterizations. Our soot-PCF parameterization, on the contrary, only relies on general soot particle properties (overlap, primary particle diameter, contact angle), and can either be adjusted to an experimental dataset, or can be chosen to broadly cover characteristic properties of atmospheric soot particles in the absence of experimental data on ice nucleation.*

*Therefore, we feel that our soot-PCF parameterization is superior to INAS-based parameterizations and indeed has predictive power for atmospheric relevant soot properties.*

*We are aware of just one INAS parameterization of soot in the cirrus regime, namely the one by Ullrich et al.* (2017)*. This parameterization relies on ice nucleation onsets from AIDA campaigns performed with only a few different soot types. It is therefore very specific for the soot samples chosen for the AIDA campaigns and does not necessarily cover the wide variety of atmospheric soot types and their ice nucleation activities. The equation used by Ullrich et al.* (2017) *for the soot parameterization requires five fitting parameters (see their Eq. 7 and Table 7) that just describe the isolines of the ice nucleation activity, but do not have a deeper physical background. By contrast, the soot-PCF parametrization requires a single fit function, $P_N(RH)$, along with information about the (measured) physicochemical particle properties to describe the ice nucleation activity of a soot sample as a function of relative humidity and soot aggregate size as shown in our Fig. 6.*

*We added the following text to specify the advantages of the soot-PCF parameterization compared with INAS parameterizations.*

*"The physical basis of the soot-PCF parameterization permits easy and fast adaptation to different types of atmospherically relevant soot, while parameterizations based on ice nucleation active surface site (INAS) densities are mostly restricted to the data underlying their parameterization. Consequently, INAS-based parametrizations cannot account for the wide diversity of soot types and their ice nucleation activities. Moreover, INAS-based parameterizations, such as the one by Ullrich et al. (2017), assume a scaling of ice nucleation sites with the particle (aggregate) surface area. Yet, the size-resolved measurements displayed in Fig. 6 exemplify that the ice-active particle fraction declines much more strongly with particle diameter than the surface area does. This is even more important considering that laboratory studies often concentrate on soot aggregates with mobility diameters much larger than typically found in the atmosphere (see Sect. 2.3), likely resulting in an overestimation of the effect soot has on cirrus formation. Conversely, the soot-PCF parameterization has a physically constrained size dependence and more importantly assumes pores and not the particle surface area to determine the ice nucleation activity of soot aerosols."*

15) Line 855: I would rather say your parameterization increases the complexity of the parameters. You emphasize that you need different fit values for each soot type, and in- stead of just using the overall aerosol surface area as a parameter, you need additional values for the primary particles size and the degree of compaction. Depending on the soot type, you also discriminate between parameterizations with respect to either RHi or RHw (Eq. E4 and E5). This is by no means meant as a criticism of your approach – but my understanding of your study is rather that you need more parameters to characterize the actual ice active sites in the soot agglomerates using a physically-constrained approach so that you can properly describe their ice nucleation ability.

*Please note that the complexity of our soot-PCF parameterization reflects the complexity of soot and allows to explain why different soot types show widely different ice nucleation activity. Having just one soot parameterization for all soot types – while desirable – would neglect this diversity. Furthermore, we highlight that the soot-PCF parameterization just requires two fitting parameters with physical significance compared with e.g. five purely empirical fitting parameters used by the Ullrich et al. (2017) parameterization.*

*However, we acknowledge that future soot ice nucleation studies should more carefully characterize and quantify the physiochemical properties of the soot types investigated, such as primary particle size and fractal dimension. This will ultimately allow to further test the validity of the soot-PCF framework presented here and eventually allow to derive a set of general "average parameters" that can be used within the soot-PCF framework.*

16) Line 871: "may vastly improve" – is a rather strong statement which also requires a more quantitative comparison to previous parameterizations.

*We indeed view the increased physical understanding of ice nucleation by soot thanks to the soot-PCF framework as a vast improvement. Please also see our answers to your comments above. This framework provides guidance in the exploration of the multidimensional parameter space spanned by soot. This allows a more focused planning of future ice nucleation experiments of soot and faster progress of the community to better understand the impacts and consequences of aerosol-cloud interaction of soot particles.*

*Furthermore, application of INAS-based parameterizations within a bulk microphysical scheme requires drastic assumptions and simplifications to derive ice-active soot particle fractions from the active site surface density. Conversely, the soot-PCF framework directly predicts ice-active fractions from measured mobility diameters, thus eliminating conversion bias. Thus, the physically based description of particle size and RH dependence of soot-PCF allows accurate simulation of soot-cirrus interactions in detailed cloud models* (Kärcher et al., in prep.).

17) Line 882: "have used available literature data" – yes, but there are many more previous measurements which you could include in Fig. 6, as suggested above.

*We have chosen the two datasets from Mahrt et al. (2018, 2020) because they cover the whole*

*ice activation curve and include more than one particle size. Moreover, the soot types used in these studies have been carefully characterized with respect to fractal dimension and primary particle size, two parameters that are key for the soot-PCF framework. To emphasize the need of carefully characterizing the soot particle properties in future soot ice nucleation experiments, we have added the following:*

*"While we have used available literature data to constrain our framework, further experimental exploration is required, in particular on ambient soot particles. Such studies should take great care in characterizing the physical and chemical properties of the soot particles, in particular those discussed herein, which will be essential to further validate the soot-PCF framework. We re-iterate that there is a clear need for dedicated ice nucleation studies on size-selected soot particles, also for aggregate sizes with (mobility) diameters below 100 nm, as well as studies exploring the impact of semi-volatile coatings on the ice nucleation ability of these particles. Lastly, we underscore the importance to report full ice activation spectra, rather than only ice nucleation onset conditions, to further improve our understanding of the physical principles determining the ice nucleation by soot particles and thereby better predict their effects on clouds and climate."*

*Furthermore, we added an additional table in Appendix F of the revised manuscript that tabulates previous studies of ice nucleation by soot in the cirrus regime.*

18) Line 1004/1005: What is meant here with "example of a pore of 8 nm diameter"? Don't you calculate the saturation ratios for the whole range of filling levels?

*This statement is erroneous. We correct it in the revised version. Thank you for pointing this out.*

Some technical corrections:

*Thank you for the technical corrections.*

Line 38: Delete "of" before "ice formation". *Done*

Line 54: "major gaps exist". *Done*

Line 70: add "freezing", i.e. condensation/immersion freezing mode. *Done*

Line 187: Maybe add comma after "Overall". *Done*

Line 267: Delete comma after "graphite". *Done*

Line 324: Delete comma after "cell". *Done*

Line 403: "in the immersion freezing mode". *Done*

Line 546: please add: "growing ice phase out of a three-membered ring pore" *Done*

Line 568/569: You may briefly add here that the point for a filling level of 1.25 nm is indicated by "A" in Fig. 5. *Done*

Line 1077: "randomly" fluctuating. *Done*

[revised manuscript text omitted]